# Accuracies of field $CO_2$−$H_2O$ data from open-path eddy-covariance flux systems: Assessment based on atmospheric physics and biological environment

Xinhua Zhou[1,2], Tian Gao[1,3], Ning Zheng[1,4], Bai Yang[1,2], Yanlei Li[1,2], Fengyuan Yu[1,3], Tala Awada[5], Jiaojun Zhu[1,3]

[1] Ker Research and Development, CAS Key Laboratory of Forest Ecology and Management, Institute of Applied Ecology, Chinese Academy of Sciences, Shenyang 110016, China
[2] Campbell Scientific Inc., Logan, UT 84321, USA
[3] Qingyuan Forest CERN, National Observation and Research Station, Liaoning Province, Shenyang 110016, China
[4] Beijing Servirst Technology Limited, Beijing 100085, China
[5] School of Natural Resources, University of Nebraska, Lincoln, NE 68583, USA

*Correspondence to*: Tian Gao (tiangao@iae.ac.cn) and Ning Zheng (ning.zheng@servirst.com)

**Abstract.** Ecosystem $CO_2$−$H_2O$ data measured by infrared gas analyzers in open-path eddy-covariance (OPEC) systems have numerous applications, such as estimations of $CO_2$ and $H_2O$ fluxes in the atmospheric boundary layer. To assess the applicability of the data for these estimations, data uncertainties from analyzer measurements are needed. The uncertainties are sourced from the analyzers in zero drift, gain drift, cross-sensitivity, and precision variability. These four uncertainty sources are individually specified for analyzer performance, but no methodology exists yet to combine these individual sources into a composite uncertainty for the specification of an overall accuracy, which is ultimately needed. Using the methodology for close-path eddy-covariance systems, this overall accuracy for OPEC systems is determined from all individual uncertainties via an accuracy model and further formulated into $CO_2$ and $H_2O$ accuracy equations. Based on atmospheric physics and the biological environment, for EC150 infrared $CO_2$−$H_2O$ analyzers, these equations are used to evaluate $CO_2$ accuracy ($\pm1.22$ mg$CO_2$ m$^{-3}$, relatively $\pm0.19\%$) and $H_2O$ accuracy ($\pm0.10$ g$H_2O$ m$^{-3}$, relatively $\pm0.18\%$ in saturated air at 35 °C and 101.325 kPa). Both accuracies are applied to conceptual models addressing their roles in uncertainty analyses for $CO_2$ and $H_2O$ fluxes. For the high-frequency air temperature derived from $H_2O$ density along with sonic temperature and atmospheric pressure, the role of $H_2O$ accuracy in its uncertainty is similarly addressed. Among the four uncertainty sources, cross-sensitivity and precision variability are minor, although unavoidable, uncertainties whereas zero drift and gain drift are major uncertainties but are minimizable via corresponding zero and span procedures during field maintenance. The accuracy equations provide rationales to assess and guide the procedures. For the atmospheric background $CO_2$ concentration, $CO_2$ zero and $CO_2$ span procedures can narrow the $CO_2$ accuracy range by 40%, from $\pm1.22$ to $\pm0.72$ mg$CO_2$ m$^{-3}$. In hot and humid weather, $H_2O$ gain drift potentially adds more to the $H_2O$ measurement uncertainty, which requires more attention. If $H_2O$ zero and $H_2O$ span procedures can be performed practically from 5 to 35 ℃, the $H_2O$ accuracy can be improved by 30% at minimum, from $\pm0.10$ to $\pm0.07$ g$H_2O$ m$^{-3}$. Under freezing conditions, the $H_2O$ span

procedure is impractical but can be neglected because of its trivial contributions to the overall uncertainty. However, the zero procedure for $H_2O$, along with $CO_2$, is imperative as an operational and efficient option under these conditions to minimize $H_2O$ measurement uncertainty.

## 1 Introduction

Open-path eddy-covariance (OPEC) systems are used most in quantity to measure boundary-layer $CO_2$, $H_2O$, heat, and momentum fluxes between ecosystems and the atmosphere (Lee and Massman, 2011). For flux measurements, an OPEC system is equipped with a fast-response three-dimensional (3-D) sonic anemometer, to measure 3-D wind velocities and sonic temperature ($T_s$), and a fast-response infrared $CO_2$−$H_2O$ analyzer (hereafter referred to as an infrared analyzer or analyzer) to measure $CO_2$ and $H_2O$ concentrations or densities (Fig. 1). In this system, the analyzer is adjacent to the sonic measurement volume. Both anemometer and analyzer together provide synchronized high-frequency (e.g., 10 to 20 Hz) measurements, which are used to compute the fluxes at a location represented by the measurement volume (Aubinet et al., 2012). Given that the measurement conditions, which are spatially homogenous in flux sources/sinks and temporally steady in turbulent flows without advection, satisfy the underlying theory for eddy-covariance flux techniques (Katul et al., 2004; Finnigan, 2008), the quality of each flux data primarily depends on the exactness of field measurements of the variables, such as $CO_2$, $H_2O$, $T_s$, and 3-D wind, at the sensor sensing scales (Foken et al., 2012; Richardson et al., 2012), although the quality may also be degraded by other biases if not fully corrected. In an OPEC system, other biases are commonly sourced from the tilt of vertical axis of the sonic anemometer away from the vertical vector of natural wind (Kaimal and Haugen, 1969), the spatial separation between the anemometer and the analyzer (Laubach and McNaughton, 1998), the line and/or volume averaging of measurements (Wyngaard, 1971; Andreas, 1981), the response delay of sensors to fluctuations in measured variables (Horst, 2000), the air density fluctuations due to heat and water vapor transfer (Webb et al., 1980), and the filtering in data processing (Rannik and Vesala, 1999). These biases are theoretically correctable through coordinate rotation corrections for the tilt (Tanner and Thurtell, 1960; Wilczak, 2001), covariance lag maximization for the separation (Moncrieff et al., 1997; Ibrom et al., 2007), low- and high-frequency corrections for the data filtering, line and/or volume averaging, and response delay (Moore, 1986; Lenschow et al., 1994; Massman, 2000; van Dijk, 2002), and Webb-Pearman-Leuning (WPL) corrections for the air density fluctuations (Webb et al., 1980). Even though these corrections are thorough for corresponding biases, errors in the ultimate flux data still exist due to uncertainties related to measurement exactness at the sensor sensing scales (Fratini et al., 2014; Zhou et al., 2018). These uncertainties are not only unavoidable because of actual or apparent instrumental drifts due to the thermal sensitivity of sensor path lengths, long-term aging of sensor detection components, and unexpected factors in field operations (Fratini et al., 2014), but they are also not mathematically correctable because their sign and magnitude are unknown (Richardson et al., 2012). The overall measurement exactness related to these uncertainties would be a valuable addition to flux data analysis (Goulden et al., 1996; Anthoni et al., 2004).

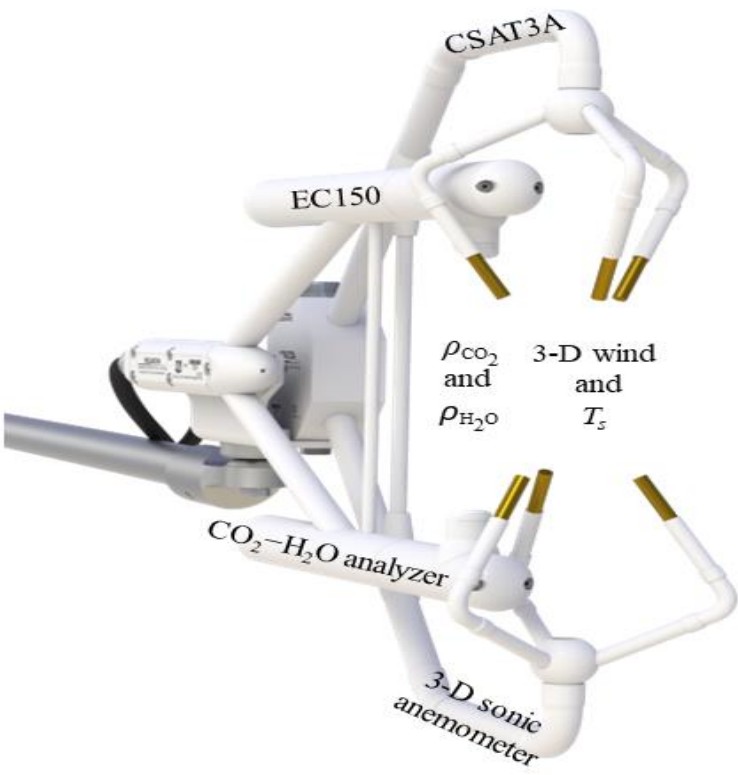

**Figure 1.** Integration of an EC150 infrared $CO_2-H_2O$ analyzer for $CO_2$ density ($\rho_{CO2}$) and $H_2O$ density ($\rho_{H2O}$) with a CSAT3A sonic anemometer for three-dimensional (3-D) wind velocities and sonic temperature ($T_s$) in an open-path eddy-covariance flux system (Campbell Scientific Inc., UT, USA).

In addition to flux computations, the data for individual variables from these field measurements can be important in numerous applications. Knowledge of measurement exactness is also required for an accurate assessment of data applicability (Csavina et al., 2017; Hill et al., 2017). The infrared analyzer in an OPEC system output $CO_2$ density ($\rho_{CO2}$ in $mgCO_2$ $m^{-3}$) and $H_2O$ density ($\rho_{H2O}$ in $gH_2O$ $m^{-3}$). For instance, $\rho_{H2O}$, along with $T_s$ and atmospheric pressure ($P$), can be used to derive ambient high-frequency air temperature ($T_a$) (Swiatek, 2018). In this case, given an exact equation of $T_a$ in terms of the three independent variables $\rho_{H2O}$, $T_s$, and $P$, the applicability of this equation to the OPEC systems for $T_a$ depends wholly on the measurement exactness of the three independent variables. The higher the degree of exactness, the less uncertain the $T_a$. The assessment on the applicability necessitates the knowledge of the measurement exactness. In reality, to the best of our knowledge, neither the overall measurement exactness of $\rho_{H2O}$ from the infrared analyzers nor the exactness of $T_s$ from the sonic anemometers (personal commnication: Larry Jecobsen, 2022) is available. This study defines

and estimates the measurement exactness of $\rho_{H_2O}$ including $\rho_{CO_2}$ from infrared analyzers through the consolidation of the measurement uncertainties, which are not practically avoidable or mathematically correctable although they can be minimized through analzyer maintenance.

As comprehensively reviewed by Richardson et al. (2012), numerous previous studies including Goulden et al. (1996), Lee et al. (1999), Anthoni et al. (1999, 2004), and Flanagan and Johonson (2005) have quantified various sources of flux measurement erorrs and have attempted to attach confidence intervals to the annual sums of net ecosystem exchange. These sources include measurement methods (e.g., sensor separation and site homogeneity (Munger et al., 2012)), data processing algorithms (e.g., data filtering (Rannik and Vesala, 1999) and data gap filling (Richardson and Hollinger, 2007)),

measurement conditions (e.g., advection (Finnigan, 2008)), energy closure (Foken, 2008), and sensor body heating effects (Burba et al., 2008). Instead of quantifying the flux errors, Foken et al. (2004, 2012) assessed the flux data into nine grades (1 to 9) based on steady state, turbulence conditions, and wind direction in the sonic anemometer coordinate system. The lower the grade, the smaller error in flux data (i.e., higher flux data quality); the higher grade, the greater error in flux data (i.e., lower flux data quality). This grade matrix (Foken et al., 2004, 2012) has been adopted by AmeriFlux (2018) for their

flux data quality assessments. To correct the measurement biases from infrared analyzers, Burba et al. (2008) developed a correction method for a sensor body heating effects on $CO_2$ and $H_2O$ fluxes, whereas Fratini et al. (2014) developed a method for correcting the raw high-frequency $CO_2$ and $H_2O$ data using the interpolated zero and span coeffcients of an infrared analyzer from the analyzer maintenance such as zero and span procedures under the same conditions, but at the beginning and ending of each maintenance period. The corrected data were then used to re-estimate the fluxes. Nevertheless,

no study has addressed the overall measurement exactness of $\rho_{H_2O}$ or $\rho_{CO_2}$, which are related to the unavoidable and uncorrectable measurement uncertainties in the $CO_2$ and $H_2O$ data from the infrared anlyzers in OPEC systems even though this overall measurement exactness is fundamental for data analysis in applications (Richardson et al., 2012). Therefore, instead for the overall exactness of an individual field $CO_2$ or $H_2O$ measurement, the infrared analyzers are specified only for their individual $CO_2$ and $H_2O$ measurement uncertainties sourced from their zero and gain drifts, cross-sensitivity to

background $H_2O/CO_2$, and measurement precision variability (LI−COR Biosciences, 2021c; Campbell Scientific Inc., 2021b).

     For any sensor, the measurement exactness depends on its performances as commonly specified in terms of accuracy, precision, and other uncertainty descriptors such as sensor hysteresis. Conventionally, accuracy is defined as a systematic uncertainty, while precision is defined as a random measurement error (ISO, 2012, where ISO is the acronym of

110 International Organization for Standardization). Other uncertainty descriptors are also defined for specific reliabilities in instrumental performance. For example, $CO_2$ zero drift is one of the descriptors specified for the performance of infrared analyzers in $CO_2$ measurements (Campbell Scientific Inc., 2021b). Both accuracy and precision are universally applicable to any sensor for the specification of its performance in measurement exactness. Other uncertainty descriptors are more sensor-

specific (e.g., cross-sensitivity to $CO_2$/$H_2O$ is used for infrared analyzers in OPEC and CPEC systems, where CPEC is an acronym for closed-path eddy-covariance).

Conventionally, sensor accuracy is the degree of closeness to which its measurements are to the true value in the measured variable; sensor precision, related to repeatability, is the degree to which repeated measurements under unchanged conditions produce the same values (Joint Committee for Guides in Metrology, 2008). Another definition advanced by the ISO (2012), revising the conventional definition of accuracy as trueness originally representing only systematic uncertainty, specifies accuracy as a combination of both trueness and precision. An advantage of this definition for accuracy is the consolidation of all measurement uncertainties. According to this definition, the accuracy is the range of composited uncertainty from all uncertainty sources in field measurements. For CPEC systems, Zhou et al. (2021) developed a method and derived a model to assess the accuracy of $CO_2$/$H_2O$ mixing ratio measurements of infrared analyzers. Their model was further formulated as a set of equations to evaluate the defined accuracies for $CO_2$ and $H_2O$ mixing rato data from CPEC systems. Although the CPEC systems are very different from OPEC systems in their structural designs (e.g., measurements take place inside a closed cuvette *vs.* in an open space) and in output variables (e.g., $CO_2$/$H_2O$ mixing ratio *vs.* $CO_2$/$H_2O$ density), similarities exist between the two systems in measurement uncertainties as specified by their manufacturers (Campbell Scientific Inc., 2021a,; 2021b) because the infrared analyzers in both systems use the same physics theories and similar optical techinques for their measurements (LI−COR Biosciences, 2021b; 2021c). Accordingly, the method developed by Zhou et al. (2021) for CPEC systems can be reasonably applied to their OPEC counterparts with rederivation of model and reformulation of equations. Following the methdology of Zhou et al. (2021) and using the specifications of EC150 infrared analyzers in OPEC systems as an example (Campbell Scientific Inc., 2021b), we can derive the model and formulate equations to assess the accuracies of $CO_2$ and $H_2O$ measurements by infrared analyzers in OPEC systems; discuss the usage of accuracies in flux analysis, data applications, and analyzer field maintenance; and ultimately provide a reference for the flux measurement community in order to specify the overall accuracy of field $CO_2$/$H_2O$ measurements by infrared analyzers in OPEC systems.

## 2 Specification implications

An OPEC system for this study includes, but is not limited to, a CSAT3A sonic anemometer and an EC150 infrared analyzer (Fig. 1). The system operates in a $T_a$ range from –30 to 50 °C and in a $P$ range from 70 to 106 kPa. Within these operational ranges, the specifications for $CO_2$ and $H_2O$ measurements (Campbell Scientific Inc., 2021b) are given in Table 1.

**Table 1.** Measurement specifications for EC150 infrared $CO_2-H_2O$ analyzers (Campbell Scientific Inc., UT, USA)

| | CO₂ | | | H₂O | | | Note |
|---|---|---|---|---|---|---|---|
| | notation | value | unit | notation | value | unit | |
| Calibration range | | $0-1,553$ | $mgCO_2\ m^{-3}$ | | $0-44$ | $gH_2O\ m^{-3}$ | For $CO_2$ up to 4,500 $mgCO_2\ m^{-3}$ if specially needed. Zero/gain drift is the possible maximum range within the system operational ranges in ambient air temperature ($T_a$) and atmospheric pressure. The actual drift depends more on $T_a$. |
| Zero drift | $d_{cz}$ | $\pm0.55$ | $mgCO_2\ m^{-3}$ | $d_{wz}$ | $\pm0.04$ | $gH_2O\ m^{-3}$ | |
| Gain drift | $d_{cg}$ | $\pm0.10\%$ [a/] true $\rho_{CO2}$ | $mgCO_2\ m^{-3}$ | $d_{wg}$ | $\pm0.30\%$ [b/] true $\rho_{H2O}$ | $gH_2O\ m^{-3}$ | |
| Cross-sensitivity to H₂O | $s_{H2O}$ | $\pm2.69\times10^{-7}$ | $mgCO_2\ m^{-3}$ $(gH_2O\ m^{-3})^{-1}$ | | N/A | | |
| Cross-sensitivity to CO₂ | | N/A | | $s_{CO2}$ | $\pm4.09\times10^{-5}$ | $gH_2O\ m^{-3}$ $(mgCO_2\ m^{-3})^{-1}$ | |
| Precision | $\sigma_{CO2}$ | 0.200 | $mgCO_2\ m^{-3}$ | $\sigma_{H2O}$ | 0.004 | $gH_2O\ m^{-3}$ | |

[a] 0.10% is the $CO_2$ gain drift percentage denoted by $\delta_{CO2\_g}$ in text, and $\rho_{CO2}$ is $CO_2$ density.

[b] 0.30% is the $H_2O$ gain drift percentage denoted by $\delta_{H2O\_g}$ in text, and $\rho_{H2O}$ is $H_2O$ density.

In Table 1, the top limit of 1,553 $mgCO_2\ m^{-3}$ in the calibration range for $CO_2$ density in dry air is more than double the atmospheric background $CO_2$ density of 767 $mgCO_2\ m^{-3}$, or 419 $\mu molCO_2\ mol^{-1}$, where mol is the unit for dry air, reported by Global Monitoring Laboratory (2022) with a $T_a$ of 20 °C under a $P$ of 101.325 kPa (i.e., normal temperature and
pressure - Wright et al. (2003)). The top limit of 44 $gH_2O\ m^{-3}$ in the calibration range for $H_2O$ density is equivalent to a 37 °C dew point, higher than the highest 35 °C dew point ever recorded under natural conditions on the Earth (National Weather Service, 2022).

The measurement uncertainties of infrared analyzers for $CO_2$ and $H_2O$ in Table 1 are specified by individual uncertainty components along with their magnitudes: zero drift, gain drift, cross-sensitivity to $CO_2/H_2O$, and precision
variability. Zero drift uncertainty is an analyzer non-zero response to zero air/gas (i.e., air/gas free of $CO_2$ and $H_2O$). Gain drift uncertainty is an analyzer trend-deviation response to a measured gas species away from its true value in proportion (Campbell Scientific Inc., 2021b). Cross-sensitivity is an analyzer background response to either $CO_2$ if $H_2O$ is measured, or $H_2O$ if $CO_2$ is measured. Precision variability is an analyzer random response to minor unexpected factors. For $CO_2$ and $H_2O$, respectively, these four components should be composited as an overall uncertainty in order to evaluate the accuracy, which
is ultimately needed in practice.

Precision variability is a random error, and the other specifications can be considered as trueness. Zero drifts are primarily impacted by $T_a$, and so are gain drifts (see the note column in Table 1 and also Fratini et al, (2014)). Additionally, each gain drift is also positively proportional to the true magnitude of $CO_2$/$H_2O$ density (i.e., true $\rho_{CO_2}$ or true $\rho_{H_2O}$) under measurements. Lastly, cross-sensitivity to $H_2O$/$CO_2$ is related to the background amount of $H_2O$/$CO_2$ as indicated by its units, $mgCO_2\ m^{-3}\,(gH_2O\ m^{-3})^{-1}$ for $CO_2$ measurements, and $gH_2O\ m^{-3}\,(mgCO_2\ m^{-3})^{-1}$ for $H_2O$ measurements.

Accordingly, beyond statistical analysis, the accuracy of $CO_2$/$H_2O$ measurements should be evaluated over a $T_a$ range of $-30$ to $50\ °C$, a $\rho_{H_2O}$ range of up to $44\ gH_2O\ m^{-3}$, and a $\rho_{CO_2}$ range of up to $1{,}553\ mgCO_2\ m^{-3}$.

## 3 Accuracy model

The measurement accuracy of infrared analyzers is the possible maximum range of overall measurement uncertainty from the four uncertainty sources as specified in Table 1: zero drift, gain drift, cross-sensitivity, and precision variability. The four uncertainties interactionally or independently contribute to the overall uncertainty of a measured value. Given the true $\alpha$ density ($\rho_{\alpha T}$, where subscript $\alpha$ can be either $CO_2$ or $H_2O$) and measured $\alpha$ density ($\rho_\alpha$), the difference between the true and measured $\alpha$ densities ($\Delta\rho_\alpha$) is given by

$$\Delta\rho_\alpha = \rho_\alpha - \rho_{\alpha T}. \tag{1}$$

The analyzer overestimates the true value if $\Delta\rho_\alpha > 0$, exactly estimates the true value if $\Delta\rho_\alpha = 0$, and underestimates the true value if $\Delta\rho_\alpha < 0$. The measurement accuracy is the maximum range of $\Delta\rho_\alpha$ (i.e., an accuracy range). According to the analyses of Zhou et al. (2021) for CPEC infrared analyzers, as mathematically shown in Appendix A, this range is interactionally contributed by the zero drift uncertainty $(\Delta\rho_\alpha^z)$, gain drift uncertainty $(\Delta\rho_\alpha^g)$, and cross-sensitivity uncertainty $(\Delta\rho_\alpha^s)$ along with an independent additon from the precision uncertainty $(\Delta\rho_\alpha^s)$. However, any interactional contribution from a pair of uncertainties is three orders smaller in magnitude than each individual contribution in the pair. The contribution to the accuracy range due to interactions can be reasonably neglected. Therefore, the accuracy range can be simply modeled as a sum of the absolute values of the four component uncertainties. From Eq. (A7) in Appendix A, the measurement accuracy of $\alpha$ density from OPEC systems by infrared analyzers is defined in an accuracy model as

$$\Delta\rho_\alpha \equiv \pm\left(\left|\Delta\rho_\alpha^z\right| + \left|\Delta\rho_\alpha^g\right| + \left|\Delta\rho_\alpha^s\right| + \left|\Delta\rho_\alpha^p\right|\right). \tag{2}$$

Assessment of the accuracy of field $CO_2$ or $H_2O$ measurements is, by use of known and/or estimable variables, the formulation and evaluation of the four terms on the right side of this accuracy model.

## 4 Accuracy of $CO_2$ density measurements

Based on accuracy Model (2), we define the accuracy of field $CO_2$ measurements from OPEC systems by infrared analyzers ($\Delta\rho_{CO_2}$) as

$\quad \Delta\rho_{CO_2} \equiv \pm\left(\left|\Delta\rho_{CO_2}^{z}\right| + \left|\Delta\rho_{CO_2}^{g}\right| + \left|\Delta\rho_{CO_2}^{s}\right| + \left|\Delta\rho_{CO_2}^{p}\right|\right),$ $\hspace{5cm}$ (3)

where $\Delta\rho_{CO_2}^{z}$ is $CO_2$ zero drift uncertainty, $\Delta\rho_{CO_2}^{g}$ is $CO_2$ gain drift uncertainty, $\Delta\rho_{CO_2}^{s}$ is cross-sensitivity-to-$H_2O$ uncertainty, and $\Delta\rho_{CO_2}^{p}$ is $CO_2$ precision uncertainty.

$CO_2$ precision ($\sigma_{CO_2}$) is the standard deviation of $\rho_{CO_2}$ random errors among repeated measurements under the same conditions (Joint Committee for Guides in Metrology, 2008). The random errors generally have a normal statistical

distribution (Hoel, 1984). Therefore, using this deviation, the precision uncertainty for an individual $CO_2$ measurement at a 95% confidence interval (P-value of 0.05) can be statistically formulated as

$$\Delta\rho_{CO_2}^{p} = \pm 1.96 \times \sigma_{CO_2}. \hspace{5cm} (4)$$

The other uncertainties, due to $CO_2$ zero drift, $CO_2$ gain drift, and cross-sensitivity-to-$H_2O$, are caused by the inability of the working equation inside the analyzer operating system (OS) to adapt the changes in analyzer-internal and

205 ambient environmental conditions, such as internal housing $CO_2$ and/or $H_2O$ levels and ambient air temperature. From the derivations in the Theory and operation section in LI−COR Biosciences (2001; 2021b; 2021c), a general model of the working equation for $\rho_{CO_2}$ is given by

$$\rho_{CO_2} = P \sum_{i=1}^{5} a_{ci} \left\{ 1 - \left[ \frac{A_c}{A_{cs}} + S_w \left( 1 - \frac{A_w}{A_{ws}} \right) \right] Z_c \right\}^{i} \left\{ \frac{G_c}{P} \right\}^{i}, \hspace{3cm} (5)$$

where subscripts $c$ and $w$ indicate $CO_2$ and $H_2O$, respectively; $a_{ci}$ ($i$ = 1, 2, 3, 4, or 5) is a coefficient of the five-order

polynomial for the terms inside curly brackets; $A_{cs}$ and $A_{ws}$ are the power values of analyzer source lights at the chosen wavelengths for $CO_2$ and $H_2O$ measurements, respectively; $A_c$ and $A_w$ are their respective remaining power values after the source lights pass through the measured air sample; $S_w$ is cross-sensitivity of the detector to $H_2O$, while detecting $CO_2$, at the wavelength for $CO_2$ measurements (hereafter referred to as sensitivity-to-$H_2O$); $Z_c$ is the $CO_2$ zero adjustment (i.e., $CO_2$ zero coefficient); and $G_c$ is the $CO_2$ gain adjustment (i.e., commonly known as the $CO_2$ span coefficient). For an individual

analyzer, the parameters $a_{ci}$, $Z_c$, $G_c$, and $S_w$ in Model (5) are statistically estimated in the production calibration against a series of standard $CO_2$ gases at different concentration levels over the ranges of $\rho_{H_2O}$ and $P$ (hereafter referred to as calibration). Since the estimated parameters are specific for the analyzer, Model (5) with these estimated parameters becomes an analyzer-specific $CO_2$ working equation. The working equation is used internally by the infrared analyzer to compute $\rho_{CO_2}$ as the closet proxy for true $\rho_{CO_2}$ from field measurements of $A_c$, $A_{cs}$, $A_w$, $A_{ws}$, and $P$.

The analyzer-specific working equation is deemed to be accurate immediately after the calibration through estimations of $a_{ci}$, $Z_c$, $G_c$, and $S_w$ in production while $Z_c$ and $G_c$ can be re-estimated in the field (LI−COR Biosciences, 2021c). However, as used internally by an optical instrument under changing environments vastly different from its calibration conditions in its

manufacturer, the working equation may not be fully adaptable to the changes, which might be reflected through $CO_2$ zero and/or gain drifts of the deployed infrared analyzer. In the working equation for $\rho_{CO_2}$ from Model (5), the parameter $Z_c$ is related to $CO_2$ zero drift; $G_c$, to $CO_2$ gain drift; and $S_w$, to sensitivity-to-$H_2O$. Therefore, the analyses of $Z_c$ and $G_c$, along with $S_w$, are an approach to understand the causes of $CO_2$ zero drift, $CO_2$ gain drift, and sensitivity-to-$H_2O$. Such understanding is necessary to formulate $\Delta\rho_{CO_2}^z$, $\Delta\rho_{CO_2}^g$, and $\Delta\rho_{CO_2}^s$ in Model (3).

## 4.1 $Z_c$ and $\Delta\rho_{CO_2}^z$ ($CO_2$ zero drift uncertainty)

In production, an infrared analyzer is calibrated for zero air/gas to report zero $\rho_{CO_2}$ plus an unaviodable random error. However, when using the analyzer in measurement environments that are different from calibration conditions, the analyzer often reports this zero $\rho_{CO_2}$, while exposed to zero air, as a value that migrates gradually away from zero and possibly beyond $\pm\Delta\rho_{CO_2}^p$, which is known as $CO_2$ zero drift. This drift is primarily affected by a combination of the three factors: i) the temperature surrounding the analyzer away from the calibration temperature, ii) traceable $CO_2$ and $H_2O$ accumulations, such as during use, inside the analyzer light housing due to an inevitable, although little, leaking exchange of housing air with the ambient air (hereafter referred to as housing $CO_2$−$H_2O$ accumulation), and iii) aging of analyzer components (Richardson et al., 2012).

Firstly, the dependency of analyzer $CO_2$ zero drift on ambient air temperature arises due to a thermal expansion/contraction of analyzer components that slightly changes the analyzer geometry (Fratini et al., 2014). This change in geometry can deviate the light path length for measurement a little away from the length under manufacturer calibration, contributing to the drift. Additionally, inside an analyzer, the performance of the light source and absorption detector for measurement, as well as the electronic components for measurement control, can vary slightly with temperature. In production, an analyzer is calibrated to compensate for the ensemble of such dependencies as assessed in a calibration chamber. The compensation algorithms are implemented in the analyzer OS, which is kept as proprietary by the analyzer manufacturer. However, the response of an analyzer to a temperature varies as conditions change over time (Fratini et al., 2014). Therefore, manufacturers typically specify an expected range of typical or maximal drift per ℃ (see Table 1 and also see the section for analyzer specifications in Campbell Scientific Inc. (2021b)). Secondly, the housing $CO_2$−$H_2O$ accumulation is caused by unavoidable little leaks in the light housing of an infrared analyzer. The housing is technically sealed to keep housing air close to zero air by implementing scrubber chemicals into the housing to absorb any $CO_2$ and $H_2O$ that may sneak into the housing through an exchange with any ambient air (LI−COR Biosciences, 2021c). Over time, the scrubber chemicals may be saturated by $CO_2$ and/or $H_2O$ or lose their active absorbing effectiveness, which can result in housing $CO_2$−$H_2O$ accumulations. Thirdly, as optical components, the light source may gradually become dim, and the absorption detector may gradually become less sensitive. The accumulation and aging develop less obviously and slowly in the relative long term (e.g., months or longer), whereas the dependencies of drift on ambient air temperature occur obviously

and quickly as soon as an analyzer is deployed in the field (Richardson et al., 2012). Apparently, the drift with ambient air temperature is a major concern if an analyzer is maintained as scheduled by its manufacturer for the replacement of scrubber chemicals (Campbell Scientific Inc., 2021b).

Due to the $CO_2$ zero drift, the working equation needs to be adjusted through its parameter re-estimation to adapt the ambient air temperature near which the system is running, housing $CO_2-H_2O$ accumulation, and analyzer component aging. This adjustment technique is the zero procedure, which brings the $\rho_{CO_2}$ and $\rho_{H_2O}$ in zero air/gas measurement back to zero as closely as possible. In this section, our discussion focuses on $CO_2$, and the same application to $H_2O$ will be described in following sections. In the field, the zero procedure should be feasibly operational using one air/gas benchmark to re-estimate one parameter in the working equation. This parameter must be adjustable to output zero $\rho_{CO_2}$ from the zero air/gas benchmark. By setting the left side of Model (5) to zero and re-arranging it, it is clear that $Z_c$ is such a parameter that can be adjusted to result in a zero $\rho_{CO_2}$ value for zero air/gas,

$$ Z_c = \left[ \frac{A_{c0}}{A_{cs}} + S_w \left( 1 - \frac{A_{w0}}{A_{ws}} \right) \right]^{-1}, \tag{6} $$

where $A_{c0}$ and $A_{w0}$ are the counterparts of $A_c$ and $A_w$ for zero air/gas, respectively. For an analyzer, the zero procedure for $CO_2$ is thus to re-estimate $Z_c$ in balance of Eq. (6).

If $Z_c$ could continually balance Eq. (6) after the zero procedure, the $CO_2$ zero drift would not be significant; however, this is not the case. Similar to its performance after the manufacturer calibration, an analyzer may still drift after the zero procedures due to frequent changes in ambient air temperature, housing $CO_2-H_2O$ accumulation, and/or analyzer component age. Nevertheless, the $Z_c$ value needed for an analyzer to be adaptable for these changes is unpredictable because these changes are not foreseeable. Assuming on-schedule maintenance (i.e., the scrubber chemicals inside the analyzer light housing is replaced following the manufacturer's guidelines), the housing $CO_2-H_2O$ accumulation should not be a concern. While the ambient temperature surrounding the infrared analyzer is not controlled, the $CO_2$ zero drift is therefore mainly influenced by $T_a$ and can be $\pm 0.55$ mg$CO_2$ m$^{-3}$ at the most within the operational ranges in $T_a$ and $P$ for the EC150 infrared analyzers in OPEC systems (Table 1).

Given that an analyzer performs best almost without zero drift at the ambient air temperature for the calibration/zeroing procedure ($T_c$), and that it possibly drifts while $T_a$ gradually changes away from $T_c$, then the further away $T_a$ is from $T_c$, the more it possibly drifts in the $CO_2$ zero. Over the operational range in $P$ of EC150 infrared ananlyzers used for OPEC systems, this drift is more proportional to the difference between $T_a$ and $T_c$ but is still within the specifications (Campbell Scientific Inc., 2021b). Accordingly, $CO_2$ zero drift uncertainty at $T_a$ can be formulated as

$$ \Delta \rho_{CO_2}^z = \frac{d_{cz}}{T_{rh} - T_{rl}} \times \begin{cases} T_a - T_c & T_c < T_a < T_{rh} \\ T_c - T_a & T_c > T_a > T_{rl} \end{cases}, \tag{7} $$

where, over the operational range in $T_a$ of EC150 infrared analyzers used for OPEC systems, $T_{rh}$ is the highest-end value (50 °C) and $T_{rl}$ is the lowest-end value (–30 °C, Table 1). $\Delta\rho_{CO_2}^z$ from this equation has the maximum range, as specified in Table 1, equal to $d_{cz}$ in magnitude as if $T_a$ and $T_c$ were separately at the two ends of operational range in $T_a$ of OPEC systems.

## 4.2 $G_c$ and $\Delta\rho_{CO_2}^g$ (CO₂ gain drift uncertainty)

An infrared analyzer was also calibrated against a series of standard CO₂ gases. The calibration sets the working equation from Model (5) to closely follow the gain trend of change in $\rho_{CO2}$. As was determined with the zero drift, the analyzer, with changes in housing CO₂−H₂O accumulation, ambient conditions, and age during its deployment, could report CO₂ gradually drifting away from the real gain trend of the change in $\rho_{CO2}$, which is specifically termed CO₂ gain drift. This drift is affected by almost the same factors as the CO₂ zero drift (Richardson et al., 2012; Fratini et al., 2014; LI−COR Biosciences, 2021c).

Due to the gain drift, the infrared analyzer needs to be further adjusted, after the zero procedure, to tune its working equation back to the real gain trend in $\rho_{CO2}$ of measured air as close as possible. This is done with the CO₂ span procedure. This procedure can be performed through use of either one or two span gases (LI−COR Biosciences, 2021c). If two are used, one span gas is slightly below the ambient CO₂ density and the other is at a much higher density to fully cover the CO₂ density range by the working equation. However, commonly, like the zero procedure, this procedure is simplified by the use of one CO₂ span gas, as a benchmark, with a known CO₂ density ($\tilde{\rho}_{CO_2}$) around the typical CO₂ density values in the measurement environment. While one CO₂ span gas is used, only one parameter in the working equation is available for adjustment. Weighing the gain of the working equation more than any other parameter, this parameter is the CO₂ span coefficient ($G_c$) (see Model 5). The CO₂ span gas is used to re-estimate $G_c$ to satisfy the following equation (for details, see LI−COR Biosciences, 2021c)

$$\left|\tilde{\rho}_{CO_2} - \rho_{CO_2}\left(G_c\right)\right| \leq \min\left|\tilde{\rho}_{CO_2} - \rho_{CO_2}\right|. \tag{8}$$

Similar to the zero drift, the CO₂ gain drift continues after the CO₂ span procedure. Based on a similar consideration for the specifications of CO₂ zero drift, the CO₂ gain drift is specified by the maximum CO₂ gain drift percentage ($\delta_{CO2\_g}$ = 0.10%) associated with $\rho_{CO2}$ as ±0.10%×(true $\rho_{CO2}$) (Table 1). This specification is the maximum range of CO₂ measurement uncertainty due to the CO₂ gain drift within the operational ranges in $T_a$ and $P$ of OPEC systems.

Given that an analyzer performs best, almost without gain drift, at the ambient air temperature for calibration/span procedure (also denoted by $T_c$, because zero and span procedures should be performed under similar ambient air temperature conditions) but also drifts while $T_a$ gradually changes away from $T_c$, then the further away $T_a$ is from $T_c$, the greater potential the drift has. Accordingly, the same approach to the formulation of CO₂ zero drift uncertainty can be applied to the formulation of approximate equation for CO₂ gain drift uncertainty at $T_a$ as

$$\Delta\rho^g_{CO_2} = \pm \frac{\delta_{CO_{2\_g}}\rho_{CO_2T}}{T_{rh} - T_{rl}} \times \begin{cases} T_a - T_c & T_c < T_a < T_{rh}, \\ T_c - T_a & T_c > T_a > T_{rl} \end{cases} \tag{9}$$

where $\rho_{CO_2T}$ is true $CO_2$ density unknown in measurement. Given that the measured value of $CO_2$ density is represented by $\rho_{CO_2}$, by referencing Eq. (1), $\rho_{CO_2T}$ can be expressed as

$$\rho_{CO_2T} = \rho_{CO_2} - (\Delta\rho^z_{CO_2} + \Delta\rho^g_{CO_2} + \Delta\rho^s_{CO_2} + \Delta\rho^p_{CO_2}). \tag{10}$$

The terms inside the parentheses in this equation are the measurement uncertainties for $\rho_{CO_2T}$ that are smaller in magnitude, by at least two orders, than $\rho_{CO_2T}$, whose magnitude in atmospheric background under the normal temperature and pressure as used by Wright et al. (2003) is 767 mgCO2 m$^{-3}$ (Global Monitoring Laboratory, 2022). Therefore, $\rho_{CO_2}$ in Eq. (10) is the best alternative, with the most likelihood, to $\rho_{CO_2T}$ for the application of Eq. (9). As such, $\rho_{CO_2T}$ in Eq. (9) can be reasonably approximated by $\rho_{CO_2}$ for equation applications. Using this approximation, Eq. (9) becomes

$$\Delta\rho^g_{CO_2} = \pm \frac{\delta_{CO_{2\_g}}\rho_{CO_2}}{T_{rh} - T_{rl}} \times \begin{cases} T_a - T_c & T_c < T_a < T_{rh} \\ T_c - T_a & T_c > T_a > T_{rl} \end{cases}. \tag{11}$$

With $\rho_{CO_2}$ being measured, this equation is applicable in estimating the $CO_2$ gain drift uncertainty. The gain drift uncertainty ($\Delta\rho^g_{CO_2}$) from this equation has the maximum range of $\pm\delta_{CO_{2\_g}}\,\rho_{CO_2}$, as if $T_a$ and $T_c$ were separately at the two ends of operational range in $T_a$ of OPEC systems. With the most likelihood, this maximum range is the closest to $\pm\delta_{CO_{2\_g}}\times$(true $\rho_{CO_2}$) as specified in Table 1.

### 4.3 $S_w$ and $\Delta\rho^s_{CO_2}$ (sensitivity-to-H$_2$O uncertainty)

The infrared wavelength of 4.3 μm for $CO_2$ measurements is minorly absorbed by H$_2$O (LI−COR Biosciences, 2021c; Campbell Scientific Inc., 2021b). This minor absorption slightly interferes with the absorption by $CO_2$ in the wavelength (McDermitt et al., 1993). The power of the same measurement light (i.e., $A_{cs}$ as a steady value in the $CO_2$ working equation from Model 5) through several gas samples with the same $CO_2$ density, but different backgrounds of H$_2$O densities, is detected with different values of $A_c$ into the working equation from Model (5). Without parameter $S_w$ and its joined term in the working equation, different $A_c$ values must result in significantly different $\rho_{CO_2}$ values, although they are actually the same. To report the same $\rho_{CO_2}$ for air flows with the same $CO_2$ density under different H$_2$O backgrounds, the different values of $A_c$ in such a case to report similar $\rho_{CO_2}$ are accounted for by $S_w$ associated with $A_w$ and $A_{ws}$ in the working equation from Model (5). Similar to $Z_c$ and $G_c$ in the $CO_2$ working equation, $S_w$ is not perfectly accurate and can have uncertainty in the determination of $\rho_{CO_2}$. This uncertainty for EC150 infrared analyzers is specified by sensitivity-to-H$_2$O ($s_{H2O}$) as $\pm2.69\times10^{-7}$ mgCO2 m$^{-3}$ (gH$_2$O m$^{-3}$)$^{-1}$ (Table 1). As indicated by its unit, this uncertainty is linearly related to $\rho_{H2O}$. Assuming the analyzer for $CO_2$ works best, without this uncertainty, in dry air, $\Delta\rho^s_{CO_2}$ could be formulated as

$$\Delta\rho^s_{CO_2} = s_{H_2O}\rho_{H_2O} \quad 0 \le \rho_{H_2O} \le 44 \, \text{gH}_2\text{On m}^{-3}, \tag{12}$$

where 44 $gH_2O\ m^{-3}$, as addressed in section 2, is the top limit of $H_2O$ density measurements. Accordingly, $\Delta\rho_{CO_2}^s$ can be in a range of

$$\Delta\rho_{CO_2}^s \leq 44|s_{H_2O}|. \tag{13}$$

### 4.4 $\Delta\rho_{CO_2}$ (CO₂ measurement accuracy)

Substituting Eqs. (4), (7), (11), and (13) into Model (3), $\Delta\rho_{CO_2}$ for an individual $CO_2$ measurement from OPEC systems can be expressed as

$$\Delta\rho_{CO_2} = \pm\left[ 1.96\sigma_{CO_2} + 44|s_{H_2O}| + \frac{|d_{cz}| + \delta_{CO_2\_g}\rho_{CO_2}}{T_{rh} - T_{rl}} \times \begin{cases} T_a - T_c & T_c < T_a < T_{rh} \\ T_c - T_a & T_c > T_a > T_{rl} \end{cases} \right]. \tag{14}$$

This is the $CO_2$ accuracy equation for EC150 infrared analyzers within OPEC systems. It expresses the accuracy of a field $CO_2$ measurement from the OPEC systems in terms of its specifications $\sigma_{CO_2}$, $s_{H_2O}$, $d_{cz}$, $\delta_{CO_2\_g}$, and the OPEC system operational range in $T_a$ as indicated by $T_{rh}$ and $T_{rl}$; measured variables $\rho_{CO_2}$ and $T_a$; and a known variable $T_c$. Given the specifications and the known variable, this equation can be used to evaluate the $CO_2$ accuracy as a range in relation to $T_a$ and $\rho_{CO_2}$.

### 4.5 Evaluation of $\Delta\rho_{CO_2}$

Given the analyzer specifications, the accuracy of field $CO_2$ measurements from an infrared analyzer after calibration, zero, and/or span at $T_c$ can be evaluated using the $CO_2$ accuracy equation (14) over a domain of $T_a$ and $\rho_{CO_2}$. To visualize the relationship of accuracy with $T_a$ and $\rho_{CO_2}$, the accuracy is presented better as the ordinate along the abscissa of $T_a$ for $\rho_{CO_2}$ at different levels and must be evaluated within possible maximum ranges of $T_a$ and $\rho_{CO_2}$ in ecosystems. In evaluation, the $T_a$ is limited to the $-30$ to 50 °C range within which EC150 infrared analyzers used for OPEC systems operate, $T_c$ can be assumed to be 20 °C (i.e., standard air temperature as used by Wright et al. (2003)), and $\rho_{CO_2}$ can be ranged acoording to its variation in ecosystems.

### 4.5.1 $\rho_{CO_2}$ range

Upper measurement limit of $CO_2$ density by the infrared analyzers can reach up to 1,553 $mgCO_2\ m^{-3}$. In the atmosphere, its $CO_2$ background mixing ratio currently is 419 $\mu molCO_2\ mol^{-1}$ (Global Monitoring Laboratory, 2022). Under the normal temperature and pressure conditions (Wright et al., 2003), this background mixing ratio is equivalent to 767 $mgCO_2\ m^{-3}$ in dry air. $CO_2$ density in ecosystems commonly ranges from 650 to 1,500 $mgCO_2\ m^{-3}$ (LI−COR Biosciences, 2021c), depending on biological processes (Wang et al., 2016), aerodynamic regimes (Yang et al., 2007), and thermodynamic states (Ohkubo et al., 2008). In this study, this range is extended from 600 to 1,600 $mgCO_2\ m^{-3}$ as a common range within which $\Delta\rho_{CO_2}$ is evaluated. Because of the dependence of $\Delta\rho_{CO_2}$ on $\rho_{CO_2}$ (Eq. 14), to show the accuracy at different $CO_2$ levels, the

range is further divided into five grades of 600, 767 (atmospheric background), 1000, 1300, and 1600 $mgCO_2$ $m^{-3}$ for evaluation presentations as in Fig. 2.

According to a brief review by Zhou et al. (2021) on the plant physiological threshold in air temperature for growth and development and the soil temperature dynamic related to $CO_2$ from microorganism respiration and/or wildlife activities in terrestrial ecosystems, $\rho_{CO_2}$ at any grade of 1,000, 1300, or 1600 $mgCO_2$ $m^{-3}$ should start, at 5 ℃, to converge asymptotically to the atmospheric $CO_2$ background (767 $mgCO_2$ $m^{-3}$ at –30 ℃, Fig. 2). Without the asymptotical function for the convergence curve, conservatively assuming the convergence has a simple linear trend with $T_a$ from 5 to –30 ℃,

$\Delta\rho_{CO_2}$ is evaluated up to the magnitude of $\rho_{CO_2}$ along the trend (Fig. 2).

### 4.5.2 $\Delta\rho_{CO_2}$ range

At $T_a = T_c$, the $CO_2$ accuracy is best at its narrowest range to be the sum of precision and sensitivity-to-$H_2O$ uncertainties (±0.39 $mgCO_2$ $m^{-3}$). However, away from $T_c$, its range near-linearly becomes wider. The $\Delta\rho_{CO_2}$ range can be summarized as ±0.40 − ±1.22 $mgCO_2$ $m^{-3}$ over the domain of $T_a$ and $\rho_{CO_2}$ (Fig. 2a and $CO_2$ columns in Table 2). The maximum $CO_2$ relative

accuracy at the different levels of $\rho_{CO_2}$ is in a range of ±0.07% at 1,600 $mgCO_2$ $m^{-3}$ to 0.19% at 600 $mgCO_2$ $m^{-3}$ (from data for Fig. 2b).

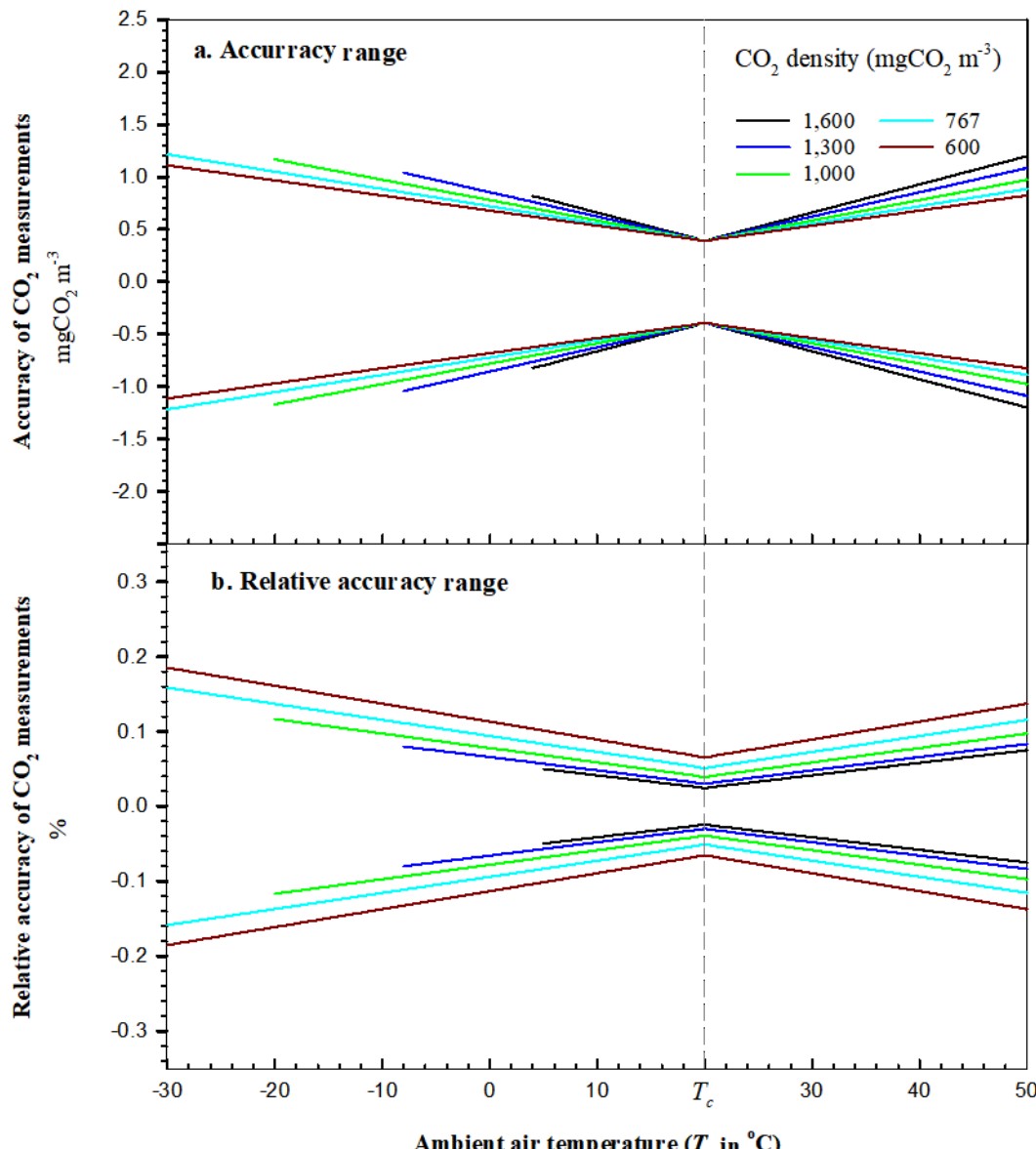

**Figure 2.** Accuracy of field $CO_2$ measurements from open-path eddy-covariance flux systems by EC150 infrared $CO_2$–$H_2O$ analyzers (Campbell Scientific Inc., UT, USA) over their operational range in $T_a$ at atmospheric pressure of 101.325 kPa. The vertical dashed line represents ambient temperature $T_c$ at which an analyzer was calibrated, zeroed, and/or spanned. Above 5 °C, accuracy is evaluated up to the possible maximum $CO_2$ density in ecosystems (black curve). Assume that this maximum $CO_2$ density starts linearly decreasing at 5 °C to the atmospheric $CO_2$ background value 767 $mgCO_2$ $m^{-3}$ at –30 °C. Accordingly, below 5 °C, the accuracy for $CO_2$ density at a level above the background value (green, blue, or black curve) is evaluated up to this decreasing trend of $CO_2$ densities. Relative accuracy of $CO_2$ measurements is the ratio of $CO_2$ accuracy to $CO_2$ density.

**Table 2.** Accuracies of field $CO_2$ and $H_2O$ measurements from open-path eddy-covariance systems by EC150 infrared $CO_2-H_2O$ analyzers (Campbell Scientific Inc., UT, USA) on the major values of background ambient air temperature, $CO_2$, and $H_2O$ in ecosystems. (Atmospheric pressure: 101.325 kPa. Calibration ambient air temperature: 20 ℃.)

| Ambient air temperature | $CO_2$ | | | | $H_2O$ | | | |
|---|---|---|---|---|---|---|---|---|
| | 767 mgCO$_2$ m$^{-3}$ [a/] | | 1,600 mgCO$_2$ m$^{-3}$ [b/] | | 60% Relative humidity | | Saturated | |
| | Accuracy ± | Relative accuracy ± | Accuracy ± | Relative accuracy ± | Accuracy ± | Relative accuracy ± | Accuracy ± | Relative accuracy ± |
| ℃ | mgCO$_2$ m$^{-3}$ | % | mgCO$_2$ m$^{-3}$ | % | gH$_2$O m$^{-3}$ | % | gH$_2$O m$^{-3}$ | % |
| −30 | 1.215 | 0.16 | | | 0.065 | 32.00 | 0.066 | 19.27 |
| −25 | 1.133 | 0.15 | | | 0.063 | 18.92 | 0.063 | 11.42 |
| −20 | 1.051 | 0.14 | | | 0.061 | 11.41 | 0.061 | 6.90 |
| −15 | 0.968 | 0.13 | N/A[c/] | | 0.059 | 7.00 | 0.059 | 4.26 |
| −10 | 0.886 | 0.12 | | | 0.056 | 4.38 | 0.057 | 2.67 |
| −5 | 0.804 | 0.10 | | | 0.054 | 2.78 | 0.056 | 1.70 |
| 0 | 0.721 | 0.09 | | | 0.052 | 1.78 | 0.054 | 1.10 |
| 5 | 0.639 | 0.08 | 0.795 | 0.05 | 0.049 | 1.22 | 0.051 | 0.75 |
| 10 | 0.557 | 0.07 | 0.661 | 0.04 | 0.047 | 0.83 | 0.049 | 0.51 |
| 15 | 0.474 | 0.06 | 0.526 | 0.03 | 0.044 | 0.57 | 0.045 | 0.35 |
| 20 | 0.392 | 0.05 | 0.392 | 0.02 | 0.040 | 0.38 | 0.040 | 0.23 |
| 25 | 0.474 | 0.06 | 0.526 | 0.03 | 0.045 | 0.33 | 0.047 | 0.20 |
| 30 | 0.557 | 0.07 | 0.661 | 0.04 | 0.052 | 0.28 | 0.056 | 0.19 |
| 35 | 0.639 | 0.08 | 0.795 | 0.05 | 0.061 | 0.26 | 0.070 | 0.18 |
| 37 | 0.672 | 0.09 | 0.849 | 0.05 | 0.065 | 0.25 | 0.077 | 0.17 |
| 40 | 0.721 | 0.09 | 0.930 | 0.06 | 0.073 | 0.24 | | |
| 45 | 0.804 | 0.10 | 1.064 | 0.07 | 0.089 | 0.23 | N/A[d/] | |
| 48 | 0.853 | 0.11 | 1.145 | 0.07 | 0.099 | 0.23 | | |
| 50 | 0.886 | 0.12 | 1.198 | 0.07 | N/A[e/] | | | |

[a] 767 mgCO$_2$ m$^{-3}$ is the atmospheric background $CO_2$ density (Global Monitoring Laboratory, 2022).

[b] 1,600 mgCO$_2$ m$^{-3}$ is assumed to be the maximum $CO_2$ density in ecosystems.

[c] $CO_2$ density in ecosystems is assumed to be lower than 1,600 mgCO$_2$ m$^{-3}$ when ambient air temperatures is below 5 ℃.

[d] $H_2O$ density in saturated air above 37 ℃ is out of the measurement range of EC150 infrared $CO_2-H_2O$ analyzers (0 − 44 gH$_2$O m$^{-3}$).

[e] $H_2O$ density in air of 60% relative humidity above 48 ℃ is out of the measurement range of EC150 infrared $CO_2-H_2O$ analyzers (0 − 44 gH$_2$O m$^{-3}$).

## 5 Accuracy of H$_2$O density measurements

Model (2) defines the accuracy of field $H_2O$ measurements from OPEC systems by infrared analyzers ($\Delta\rho_{H_2O}$) as

$$\Delta\rho_{H_2O} \equiv \pm\left(\left|\Delta\rho_{H_2O}^z\right| + \left|\Delta\rho_{H_2O}^g\right| + \left|\Delta\rho_{H_2O}^s\right| + \left|\Delta\rho_{H_2O}^p\right|\right), \tag{15}$$

where $\Delta\rho_{H_2O}^z$ is $H_2O$ zero drift uncertainty, $\Delta\rho_{H_2O}^g$ is $H_2O$ gain drift uncertainty, $\Delta\rho_{H_2O}^s$ is cross-sensitivity-to-$CO_2$ uncertainty, and $\Delta\rho_{H_2O}^p$ is $H_2O$ precision uncertainty. Using the same approach as for $\Delta\rho_{CO_2}^p$, $\Delta\rho_{H_2O}^p$ is formulated as

$$\Delta\rho_{H_2O}^P = \pm1.96 \times \sigma_{H_2O}, \tag{16}$$

where $\sigma_{H_2O}$, as defined in Table 1, is the precision of EC150 analyzers for $H_2O$ measurements. The other uncertainty terms in Model (15) can be understood and formulated using the similar approach for their counterparts in Model (3).

## 5.1 $\Delta\rho_{H_2O}^z$ ($H_2O$ zero drift uncertainty) and $\Delta\rho_{H_2O}^g$ ($H_2O$ gain drift uncertainty)

The model of the analyzer working equation for $\rho_{H_2O}$ is similar to Model (5) for $\rho_{CO_2}$ in formulation, given also by the derivations in the Theory and operation section in LI−COR Biosciences (2001; 2021b; 2021c)

$$\rho_{H_2O} = P\sum_{i=1}^{3} a_{wi}\left\{1 - \left[\frac{A_w}{A_{ws}} + S_c\left(1 - \frac{A_c}{A_{cs}}\right)\right]Z_w\right\}^i \left\{\frac{G_w}{P}\right\}^i, \tag{17}$$

where $a_{wi}$ (i = 1, 2, or 3) is a coefficient of the three-order polynomial in the terms inside curly brackets; $S_c$ is the cross-sensitivity of a detector to $CO_2$, while detecting $H_2O$, at the wavelength for $H_2O$ measurements (hereafter referred to as sensitivity-to-$CO_2$); $Z_w$ is the $H_2O$ zero adjustment (i.e., $H_2O$ zero coefficient); $G_w$ is the $H_2O$ gain adjustment (i.e., commonly referred as to $H_2O$ span coefficient); and $A_w$, $A_{ws}$, $A_c$, and $A_{cs}$ represent the same as in Model (5). The parameters of $a_{wi}$, $Z_w$ $G_w$, and $S_c$ in Model (17) are statistically estimated to establish an $H_2O$ working equation in the production calibration against a series of air standards with different $H_2O$ contents under ranges of $\rho_{CO_2}$ and $P$ (i.e., calibration). The $H_2O$ working equation (i.e., Model 17 with estimated parameters) is used inside the analyzer OS to compute $\rho_{H_2O}$ as the closest proxy for true $\rho_{H_2O}$ from field measurements of $A_w$, $A_{ws}$, $A_c$, $A_{cs}$, and $P$.

Because of the similarities in model principles and parameter implications between Models (5) and (17), following the same analyses and rationales as for $\Delta\rho_{CO_2}^z$ and $\Delta\rho_{CO_2}^g$, $\Delta\rho_{H_2O}^z$ is formulated as

$$\Delta\rho_{H_2O}^z = \frac{d_{wz}}{T_{rh} - T_{rl}} \times \begin{cases} T_a - T_c & T_c < T_a < T_{rh} \\ T_c - T_a & T_c > T_a > T_{rl} \end{cases}, \tag{18}$$

and $\Delta\rho_{H_2O}^g$ is formulated as

$$\Delta\rho_{H_2O}^g = \pm\frac{\delta_{H_2O\_g}\rho_{H_2O}}{T_{rh} - T_{rl}} \times \begin{cases} T_a - T_c & T_c < T_a < T_{rh} \\ T_c - T_a & T_c > T_a > T_{rl} \end{cases}. \tag{19}$$

## 5.2 $\Delta\rho_{H_2O}^s$ (sensitivity-to-CO₂ uncertainty)

The infrared light at wavelength of 2.7 μm for $H_2O$ measurement is traceably absorbed by $CO_2$ (see Fig. 4.7 in Wallace and Hobbs, 2006). This absorption interferes slightly with the $H_2O$ absorption at this wavelength (McDermitt et al., 1993). As such, the power of identical measurement lights (i.e., $A_{ws}$ as a steady value in the $H_2O$ working equation from Model 17) through several air standards with the same $H_2O$ density but different backgrounds of $CO_2$ amounts would result in different values of $A_w$ into the $H_2O$ working equation from Model (17). In this equation, without parameter $S_c$ and its joined term, different $A_w$ values will result in significantly different $\rho_{H2O}$ values, although $\rho_{H2O}$ is essentially the same. To report the same $\rho_{H2O}$ for air flows with the same $H_2O$ amount under different $CO_2$ backgrounds, different values of $A_w$ in such a case to report the same $\rho_{H2O}$ are accounted for by $S_c$ associated with $A_c$ and $A_{cs}$ in the $H_2O$ working equation from Model (17). However, $S_c$ is not perfectly accurate, either, having uncertainty in the determination of $\rho_{H2O}$. This uncertainty in the EC150 infrared analyzer is specified by the sensitivity-to-CO₂ ($s_{CO2}$) value as the maximum range of $\pm 4.09 \times 10^{-5}$ gH₂O m⁻³ (mgCO₂ m⁻³)⁻¹ (Table 1). Assuming the infrared analyzers for $H_2O$ have the lowest sensitivity-to-CO₂ uncertainty for air flow with an atmospheric background $CO_2$ amount (i.e., 767 mgCO₂ m⁻³), $\Delta\rho_{H_2O}^s$ could be formulated as

$$\Delta\rho_{H_2O}^s = s_{CO_2}\left(\rho_{CO_2} - 767\right) \qquad \rho_{CO_2} \leq 1,553 \text{ mgCO}_2\text{n m}^{-3}. \tag{20}$$

Accordingly, $\Delta\rho_{H_2O}^s$ can be reasonably expressed as

$$\left|\Delta\rho_{H_2O}^s\right| \leq 786 s_{CO_2}. \tag{21}$$

## 5.3 $\Delta\rho_{H2O}$ (H₂O measurement accuracy)

Substituting Eqs. (16), (18), (19) and (21) into Model (15), $\Delta\rho_{H2O}$ for an individual $H_2O$ measurement from OPEC systems can be expressed as

$$\Delta\rho_{H_2O} = \pm\left[1.96\sigma_{H_2O} + 786\left|s_{CO_2}\right| + \frac{\left|d_{wz}\right| + \delta_{H_2O\_g}\rho_{H_2O}}{T_{rh} - T_{rl}} \times \begin{cases} T_a - T_c & T_c < T_a < T_{rh} \\ T_c - T_a & T_c > T_a > T_{rl} \end{cases}\right]. \tag{22}$$

This equation is the $H_2O$ accuracy equation for the OPEC systems with infrared analyzers. It expresses the accuracy of $H_2O$ measurements from the OPEC systems in terms of the specifications $\sigma_{H2O}$, $s_{CO2}$, $d_{wz}$, $\delta_{H2O\_g}$, $T_{rh}$, and $T_{rl}$; measured variables $\rho_{H2O}$ and $T_a$; and a known variable $T_c$. Using this equation and the specification values as in Table 1 for EC150 infrared analyzers, the accuracy of field $H_2O$ measurements can be evaluated as a range for OPEC systems with such anlyzers. For an OPEC system with another model of open-path infrared anlyzer, such as the LI−7500 series (LI−COR Biosciences, NE, USA) or IRGASON (Campbell Scientific Inc., UT, USA), its corresponding specification values are used.

### 5.4 Evaluation of $\Delta\rho_{H_2O}$

$H_2O$ accuracy ($\Delta\rho_{H_2O}$) can be evaluated using the $H_2O$ accuracy equation over a domain of $T_a$ and $\rho_{H_2O}$. Similar to the $CO_2$ accuracy equation in Fig. 2, $\Delta\rho_{H_2O}$ is presented as the ordinate along the abscissa of $T_a$ at different $\rho_{H_2O}$ levels within the ranges of $T_a$ and $\rho_{H_2O}$ in ecosystems (Fig. 3). As with the evaluation of $\Delta\rho_{CO_2}$, $T_a$ is limited from –30 to 50 °C and $T_c$ can be assumed to be 20 °C. The range of $\rho_{H_2O}$ at $T_a$ needs to be determined using atmospheric physics (Buck, 1981).

### 5.4.1 $\rho_{H_2O}$ range

The EC150 analyzers were calibrated for $H_2O$ density from 0 to 44 $gH_2O\ m^{-3}$ due to the reason addressed in Sect. 2. The highest limit of measurement range for $H_2O$ density by other models of analyzers also should be near 44 $gH_2O\ m^{-3}$. However, due to the positive exponential dependence of air water vapor saturation on $T_a$ (Wallace and Hobbs, 2006), $\rho_{H_2O}$ has a range that is wider at higher $T_a$ and narrower at lower $T_a$. Below 37 °C at 101.325 kPa, $\rho_{H_2O}$ is lower than 44 $gH_2O\ m^{-3}$, and its range becomes narrower and narrower, reaching 0.34 $gH_2O\ m^{-3}$ at –30 °C. To determine the $H_2O$ accuracy over the same relative range of air moisture, even at different $T_a$, the water vapor saturation density is used to scale air moisture to 20, 40, 60, 80 and 100% (i.e., relative humidity, or RH). For each scaled RH value, $\rho_{H_2O}$ can be calculated at different $T_a$ and $P$ (Appendix B) for use in the $H_2O$ accuracy equation. In this way, over the range of $T_a$, $H_2O$ accuracy can be shown as curves, along each of which RH is equal (Fig. 3).

### 5.4.2 $\Delta\rho_{H_2O}$ range

In the same way as with $CO_2$ accuracy, the $H_2O$ accuracy at $T_a = T_c$ is best at its narrowest as the sum of precision and sensitivity-to-$CO_2$ uncertainties (<0.040 $gH_2O\ m^{-3}$ in magnitude). However, away from $T_c$, its non-linear range becomes wider, very gradually below this $T_c$ value but more abruptly above, because, as $T_a$ increases, $\rho_{H_2O}$ at the same RH increases exponentially (Eqs. B1 and B2 in Appendix B) while $\Delta\rho_{H_2O}$ increases linearly with $\rho_{H_2O}$ in the $H_2O$ accuracy equation (22). This non-linear range can be summarized as the widest at 48 °C to be ±0.099 $gH_2O\ m^{-3}$ for air with 60% RH (Fig. 3a and $H_2O$ columns in Table 2). The number can be rounded up to ±0.10 $gH_2O\ m^{-3}$ for the overall accuracy of field $H_2O$ measurements from OPEC systems by the EC150 infrared analyzers.

Fig. 3b shows an interesting trend of $H_2O$ relative accuracy with $T_a$. Given the RH range shown in Fig. 3b, the relative accuracy diverges with a $T_a$ decrease and converges with a $T_a$ increase. The $H_2O$ relative accuracy varies from 0.17% for saturated air at 37 °C to 96% for 20% RH air at –30 °C (data for Fig. 3b) and, at this low $T_a$, can be much greater if RH goes further lower. The $H_2O$ relative accuracy in magnitude is < 1% while $\rho_{H_2O} > 5.00\ gH_2O\ m^{-3}$, < 5% while $\rho_{H_2O} > 1.20$ $gH_2O\ m^{-3}$, and >10% while $\rho_{H_2O} < 0.60\ gH_2O\ m^{-3}$.

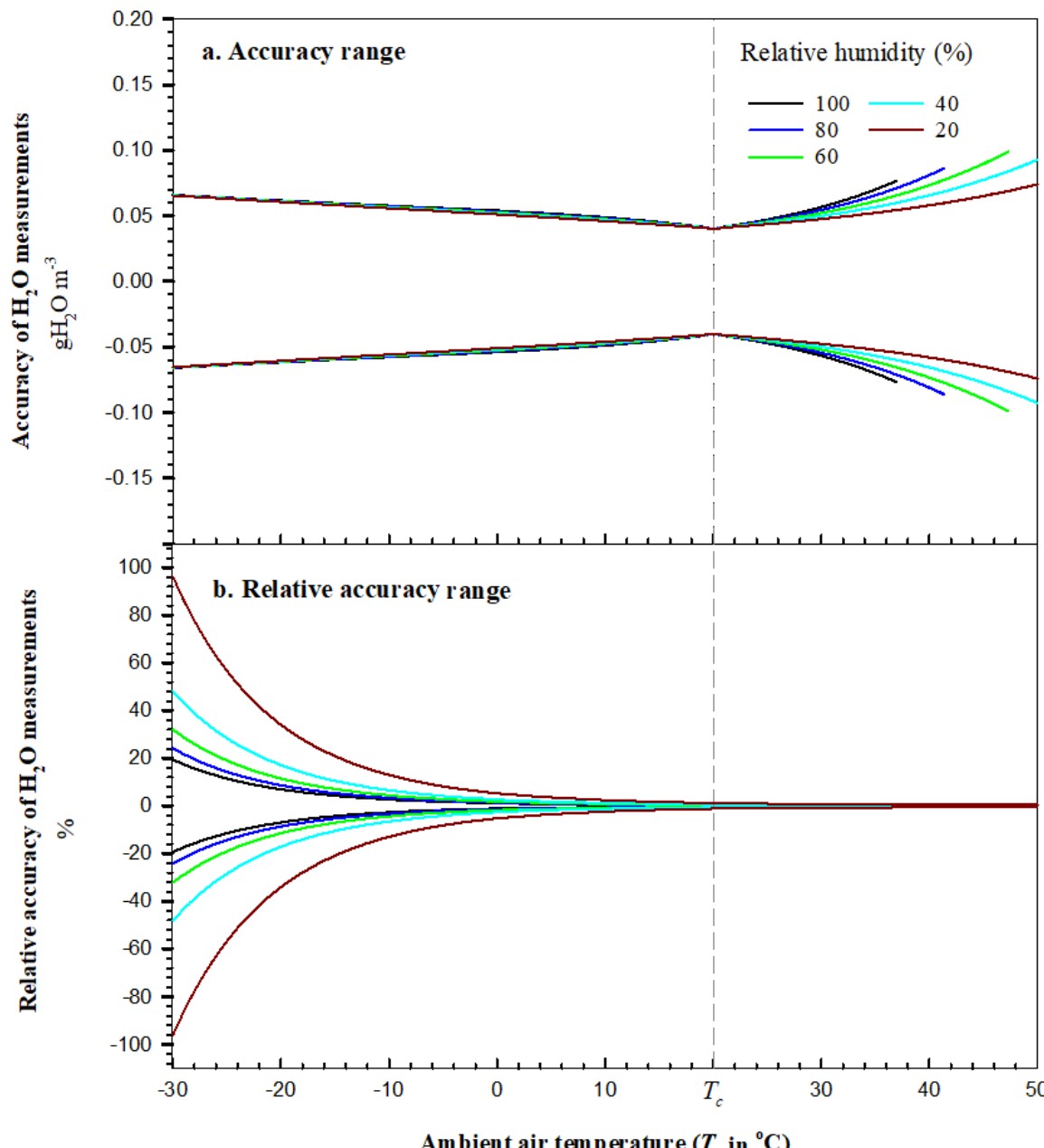

**Figure 3.** Accuracy of field $H_2O$ measurements from open-path eddy-covariance systems by EC150 infrared $CO_2$−$H_2O$ analyzers (Campbell Scientific Inc., UT, USA) over their operational range in $T_a$ under atmospheric pressure of 101.325 kPa. The vertical dashed line represents the ambient air temperature ($T_c$) at which an analyzer was calibrated, zeroed, and/or spanned. Relative accuracy of $H_2O$ measurements is the ratio of $H_2O$ accuracy to $H_2O$ density.

## 6 Application

The primary objective of this study is to develop an assessment methodology to evaluate the overall accuracies of field $CO_2$ and $H_2O$ measurements from the infrared analyzers in OPEC systems by compositing their individual measurement uncertainties as specified with four uncertainty descriptors: zero drift, gain drift, sensitivity-to-$CO_2$/$H_2O$, and precision variability (Table 1). Ultimately, the overall accuracies (i.e., $\Delta\rho_{CO2}$ and $\Delta\rho_{H2O}$) make uncertainty analyses possible for the various applications of $CO_2$ and $H_2O$ data and the composited accuracy equations (i.e., Eqs. 14 and 22) make the field maintenance rationale for infrared analyzers.

### 6.1 Application of $\Delta\rho_{CO2}$ and $\Delta\rho_{H2O}$ to the uncertainty analyses for $CO_2$ and $H_2O$ flux data

As discussed in Introduction, the uncertainty of each flux data is contributed by numerous sub-uncertainties in the processes of measurements and computations, among which $\Delta\rho_{CO2}$ and $\Delta\rho_{H2O}$ are two fundamental uncertainties of the measurements from infrared analyzers. For this study topic, assuming 3-D wind speeds are accurately measured by a sonic anemometer, Appendix C demonstrates that neither $\Delta\rho_{CO2}$ nor $\Delta\rho_{H2O}$ brings an uncertainty into the covariance of vertical wind speed ($w$) with $\rho_{CO2}$, $\rho_{H2O}$, or $T_a$ even after coordinate rotations, lag maximization, and low- and high-frequency corrections, given by Eqs. (C8) and (C9) in the appendix as

$$\overline{\left(w'\rho'_{CO_2}\right)}_{rmf} = \overline{\left(w'\rho'_{CO_2T}\right)}_{rmf}$$
$$\overline{\left(w'\rho'_{H_2O}\right)}_{rmf} = \overline{\left(w'\rho'_{H_2OT}\right)}_{rmf} \tag{23}$$
$$\overline{\left(w'T'_a\right)}_{rmf} = \overline{\left(w'T'_{aT}\right)}_{rmf}$$

where the overbar is a Reynolds' averaging operator, prime denotes the fluctuations of a variable away from its mean (e.g., $w'_i = w_i - \overline{w}$), subscript $T$ indicates "true" value (see Appendix C for the implication of "true" value), and subscript $rmf$ indicates the covariance was corrected through coordinate rotations ($r$), lag maximization ($m$), and low- and high-frequency corrections ($f$). The three equalities in Eq. (23) that are proved in Appendix C prove that the measured covariance of $w$ with $\rho_{CO2}$, $\rho_{H2O}$, or $T_a$ is not affected by corresponding $\Delta\rho_{CO2}$, $\Delta\rho_{H2O}$ or $\Delta T_a$ (i.e., accuracy of $T_a$), being equal to the true covariance. Further, through WPL corrections, the three terms on the left side of Eq. (23) can be used to derive an analytical equation for measured $CO_2$ or $H_2O$ flux whereas the three terms on the right side of this equation can be used to derive an analytical equation for true $CO_2$ or $H_2O$ flux. The comparison of both analytical equations can demonstrate the partial effects of $\Delta\rho_{CO2}$ and $\Delta\rho_{H2O}$ on the uncertainty of $CO_2$ or $H_2O$ flux data.

## 6.1.1 Roles of $\Delta\rho_{CO_2}$ and $\Delta\rho_{H_2O}$ in the uncertainty of CO2 flux data

Using the terms on the left side of Eq. (23), through the WPL corrections for CO2 flux from $\overline{\left(w'\rho'_{CO_2}\right)}_{rmf}$ (Webb et al., 1980), the measured CO2 flux ($F_{CO_2}$) is given by

$$F_{CO_2} = \overline{\left(w'\rho'_{CO_2}\right)}_{rmf} + \left[ \mu \frac{\overline{\rho}_{CO_2}}{\overline{\rho}_d} \overline{\left(w'\rho'_{H_2O}\right)}_{rmf} + \left(1 + \mu \frac{\overline{\rho}_{H_2O}}{\overline{\rho}_d}\right) \frac{\overline{\rho}_{CO_2}}{\overline{T}_{aK}} \overline{\left(w'T'_a\right)}_{rmf} \right], \tag{24}$$

where $\mu$ is the ratio of dry air to water molecular weight, $\rho_d$ is dry air density, and $T_{aK}$ is air temperature in Kalvin.

According to Eqs. (C1) and (23), this equation can be written as

$$F_{CO_2} = \overline{\left(w'\rho'_{CO_2T}\right)}_{rmf} +$$
$$\left[ \mu \frac{\overline{\rho}_{CO_2T} + \Delta\overline{\rho}_{CO_2}}{\overline{\rho}_{dT} - \Delta\overline{\rho}_{H_2O}} \overline{\left(w'\rho'_{H_2O}\right)}_{rmf} + \left(1 + \mu \frac{\overline{\rho}_{H_2OT} + \Delta\overline{\rho}_{H_2O}}{\overline{\rho}_{dT} - \Delta\overline{\rho}_{H_2O}}\right) \frac{\overline{\rho}_{CO_2T} + \Delta\overline{\rho}_{CO_2}}{\overline{T}_{aKT} + \Delta\overline{T}_a} \overline{\left(w'T'_a\right)}_{rmf} \right], \tag{25}$$

where $\Delta\overline{T}_a$ is the accuracy of $\overline{T}_{aK}$. $\Delta\overline{T}_a$ is well defined as ±0.20 K in compliance with the WMO standard (WMO, 2018).

According to Eqs. (23) and (24), from $\overline{\left(w'\rho'_{CO_2T}\right)}_{rmf}$, the nominal true CO2 flux ($F_{CO_2T}$) can be given by

$$F_{CO_2T} = \overline{\left(w'\rho'_{CO_2T}\right)}_{rmf} + \left[ \mu \frac{\overline{\rho}_{CO_2T}}{\overline{\rho}_{dT}} \overline{\left(w'\rho'_{H_2O}\right)}_{rmf} + \left(1 + \mu \frac{\overline{\rho}_{H_2OT}}{\overline{\rho}_{dT}}\right) \frac{\overline{\rho}_{CO_2T}}{\overline{T}_{aKT}} \overline{\left(w'T'_a\right)}_{rmf} \right]. \tag{26}$$

From Eqs (25) and (26), the uncertainty of CO2 flux ($\Delta F_{CO_2}$) can be expressed as

$$\Delta F_{CO_2} = F_{CO_2} - F_{CO_2T}$$
$$= \mu \left( \frac{\overline{\rho}_{CO_2T} + \Delta\overline{\rho}_{CO_2}}{\overline{\rho}_{dT} - \Delta\overline{\rho}_{H_2O}} - \frac{\overline{\rho}_{CO_2T}}{\overline{\rho}_{dT}} \right) \overline{\left(w'\rho'_{H_2O}\right)}_{rmf} +$$
$$\left[ \left(1 + \mu \frac{\overline{\rho}_{H_2OT} + \Delta\overline{\rho}_{H_2O}}{\overline{\rho}_{dT} - \Delta\overline{\rho}_{H_2O}}\right) \frac{\overline{\rho}_{CO_2T} + \Delta\overline{\rho}_{CO_2}}{\overline{T}_{aKT} + \Delta\overline{T}_a} - \left(1 + \mu \frac{\overline{\rho}_{H_2OT}}{\overline{\rho}_{dT}}\right) \frac{\overline{\rho}_{CO_2T}}{\overline{T}_{aKT}} \right] \overline{\left(w'T'_a\right)}_{rmf} \tag{27}$$

This derivation provides a conceptual model for the partial effects of $\Delta\rho_{CO_2}$ and $\Delta\rho_{H_2O}$ on the uncertainty of CO2 flux data. This uncertainty is added by $\Delta\rho_{CO_2}$ and $\Delta\rho_{H_2O}$ interactively with the density effect due to H2O flux (i.e., the term with $\overline{\left(w'\rho'_{H_2O}\right)}_{rmf}$ in Eq. 27) and temperature flux (i.e., the term with $\overline{\left(w'T'_a\right)}_{rmf}$ in Eq. 27).

## 6.1.2 $\Delta\rho_{H_2O}$ on uncertainty of H2O flux data

Using the same approach to Eq. (27), the uncertainty of H2O flux ($\Delta F_{H_2O}$) can be expressed as

$$\Delta F_{H_2O} = \mu \left( \frac{\bar{\rho}_{H_2OT} + \Delta\bar{\rho}_{H_2O}}{\bar{\rho}_{dT} - \Delta\bar{\rho}_{H_2O}} - \frac{\bar{\rho}_{H_2OT}}{\bar{\rho}_{dT}} \right) \overline{\left(w'\rho'_{H_2O}\right)}_{rmf} +$$

$$\left[ \left( 1 + \mu \frac{\bar{\rho}_{H_2OT} + \Delta\bar{\rho}_{H_2O}}{\bar{\rho}_{dT} - \Delta\bar{\rho}_{H_2O}} \right) \frac{\bar{\rho}_{H_2OT} + \Delta\bar{\rho}_{H_2O}}{\bar{T}_{aKT} + \Delta\bar{T}_a} - \left( 1 + \mu \frac{\bar{\rho}_{H_2OT}}{\bar{\rho}_{dT}} \right) \frac{\bar{\rho}_{H_2OT}}{\bar{T}_{aT}} \right] \overline{\left(w'T'_a\right)}_{rmf} \tag{28}$$

This formulation provides a conceptual model for the partial effects of $\Delta\rho_{H2O}$ on the uncertainty of $H_2O$ flux data. This uncertainty is added only by $\Delta\rho_{H2O}$ also interactively with the density effect due to $H_2O$ flux (i.e., the term with $\overline{\left(w'\rho'_{H_2O}\right)}_{rmf}$

in Eq. 28) and temperature flux (the term with $\overline{\left(w'T'_a\right)}_{rmf}$ in Eq. 28). Further analysis and more discussion about Eqs. (27) and (28) go beyond the scope of this study.

### 6.2 Application of $\Delta\rho_{H2O}$ to the uncertainty analysis for high-frequency air temperature

The measured variables $\rho_{H2O}$, along with $T_s$ and $P$ can be used to compute high-frequency $T_a$ in OPEC systems (Swiatek, 2018). If $T_a(\rho_{H_2O}, T_s, P)$ were an exact function from the theoretical principles, it would not have any error itself. However,

in our applications, variables $\rho_{H2O}$, $T_s$, and $P$ are measured from the OPEC systems experiencing seasonal climates. As addressed in this study, the measured values of these variables have measurement uncertainty in $\rho_{H2O}$ ($\Delta\rho_{H2O}$, i.e., accuracy of field $H_2O$ measurement); in $T_s$ ($\Delta T_s$, i.e., accuracy of field $T_s$ measurement); and in $P$ ($\Delta P$, i.e., accuracy of field $P$ measurement). The uncertainties from the measurements propagate to the computed $T_a$ as an uncertainty ($\Delta T_a$, i.e., accuracy of $T_a(\rho_{H_2O}, T_s, P)$). This accuracy is a reference by any application of $T_a$. It should be specified through the relationship of

$\Delta T_a$ to $\Delta\rho_{H2O}, \Delta T_s,$ and $\Delta P$.

As field measurement uncertainties, $\Delta\rho_{H2O}$, $\Delta T_s$, or $\Delta P$ are reasonably small increments in numerical analysis (Burden et al., 2016). As such, depending on all the small increments, $\Delta T_a$ is a total differential of $T_a(\rho_{H_2O}, T_s, P)$ with respect to $\rho_{H2O}, T_s,$ and $P$, which are measured independently by three sensors, given by

$$\Delta T_a = \frac{\partial T_a}{\partial \rho_{H_2O}} \Delta\rho_{H_2O} + \frac{\partial T_a}{\partial T_s} \Delta T_s + \frac{\partial T_a}{\partial P} \Delta P. \tag{29}$$

In this equation, $\Delta\rho_{H2O}$ from the application of Eq. (22) is a necessary term to acquire $\Delta T_a$, $\Delta T_s$ can be acquired from the specifications for 3-D sonic anemometers (Zhou et al., 2018), $\Delta P$ can be acquired from the specifications for the barometer used in the OPEC systems (Vaisala, 2020), and the three partial derivatives can be derived from the explicit function $T_a(\rho_{H_2O}, T_s, P)$. With $\Delta\rho_{H2O}, \Delta T_s, \Delta P$, and the three partial derivatives, $\Delta T_a$ can be ranged as a function of $\rho_{H2O}, T_s,$ and $P$.

### 6.3 Application of accuracy equations in analyzer field maintenance

An infrared analyzer performs better if the field environment is near its manufacturing conditions (e.g., $T_a$ at 20 °C), which is demonstrated in Figs. 2a and 3a for measurement accuracies associated with $T_c$. As indicated by the accuracies in both figures, the closer to $T_c$ at 20 °C while $T_a$ is, the better analyzers perform. However, the analyzers are used in OPEC systems mostly for long-term field campaigns through four-seasonal climates vastly different from those in the manufacturing processes. Over time, an analyzer gradually drifts in some ways and needs field maintenance although within its

specifications.

The field maintenance cannot improve the sensitivity-to-$CO_2$/$H_2O$ uncertainty and precision variability, but both are minor (their sum $< 0.392$ mg$CO_2$ m$^{-3}$ for $CO_2$, Eqs. 4 and 13; $< 0.045$ g$H_2O$ m$^{-3}$ for $H_2O$, Eqs. 16 and 21) as compared to the zero or gain drift uncertainties. However, the zero and gain drift uncertainties are major in determination of field $CO_2$/$H_2O$ measurement accuracy (Figs. 2 to 4 and Eqs. 14 and 22), but adjustable, through the zero and/or span procedures, to be

minimized. Therefore, manufacturers of infrared analyzers have provided software and hardware tools for the procedures (Campbell Scientific Inc., 2021b) and scheduled the procedures using those tools (LI−COR Biosciences, 2021c). Fratini et al. (2014) provided a technique implemented into the EddyPro® Eddy Covariance Software (LI−COR Biosciences, 2021a) to correct the drift biases from a raw time series of $CO_2$ and $H_2O$ data through post-processing. This study provides rationales how to assess, schedule, and perform the zero and span procedures (Figs. 2a, 3a, and 4).

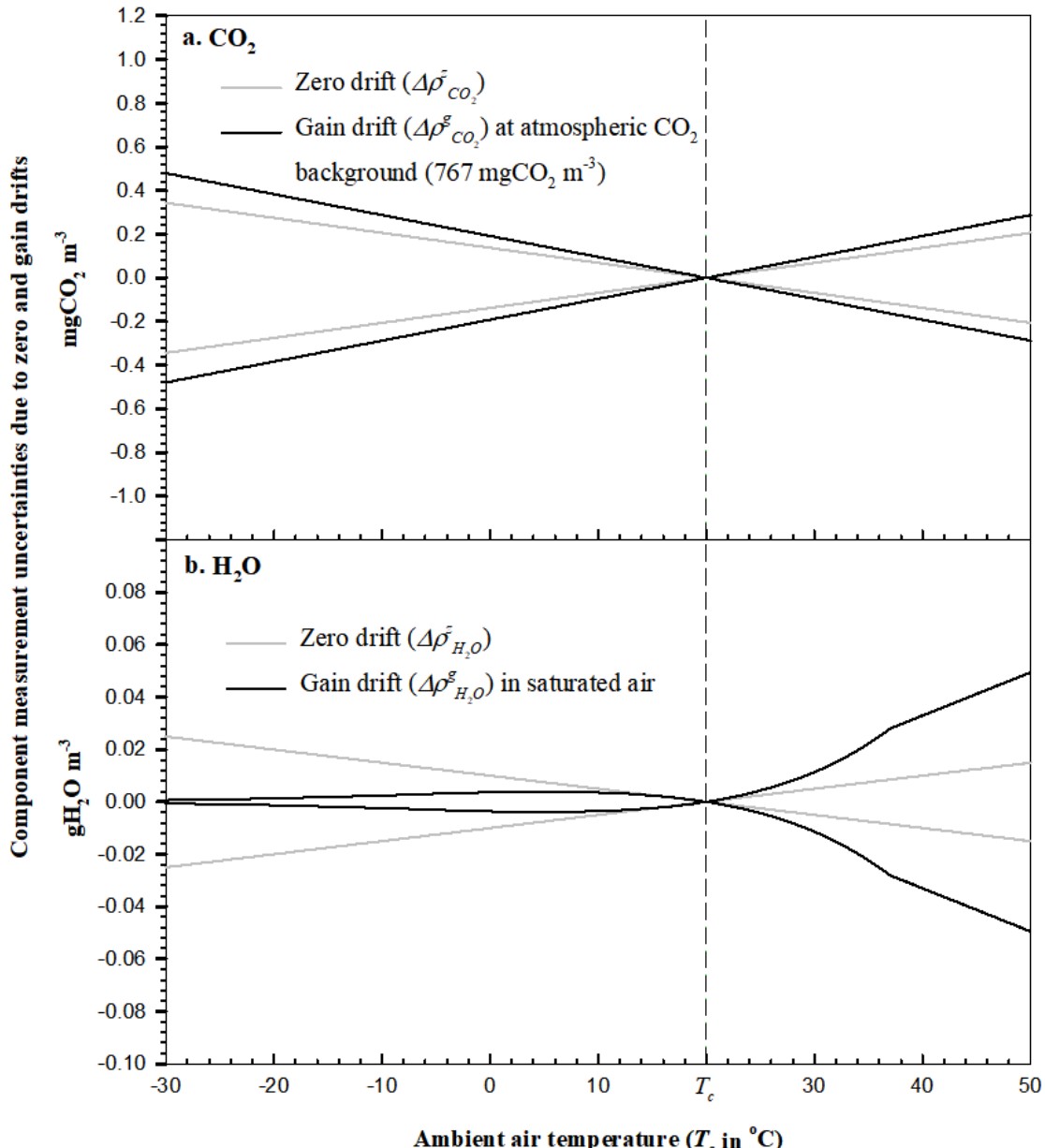

**Figure 4.** Component measurement uncertainties due to the zero and gain drifts of EC150 infrared $CO_2$–$H_2O$ analyzers (Campbell Scientific Inc, UT, USA) in open-path eddy-covariance flux systems over their operational range in $T_a$ under an atmospheric pressure of 101.325 kPa. The vertical dashed line represents the ambient air temperature ($T_c$) at which an analyzer was calibrated, zeroed, and/or spanned.

### 6.3.1 $CO_2$ zero and span procedures

Figure 4a shows that the $CO_2$ zero drift uncertainty linearly increases with $T_a$ away from $T_c$ over the full $T_a$ range within which OPEC systems operate; so, too, does $CO_2$ gain drift uncertainty increase for a given $CO_2$ concentration. As suggested by Zhou et al. (2021), both drifts should be adjusted near the $T_a$ value around which the system runs. The zero and gain drifts should be adjusted, through zero and span procedures, at a $T_a$ close to its daily mean around which the system runs. Based on the range of $T_a$ daily cycle, the procedures are set at a moderate, instead of the highest or lowest, moment in $T_a$. Given the daily cycle range is much narrower than 40 °C, an OPEC system could run at $T_a$ within ±20 of $T_c$ if the procedures are performed at a right moment of $T_a$. For our study case on atmospheric $CO_2$ background (left $CO_2$ column in Table 2), the procedures can narrow the widest possible range of ±1.22 $mgCO_2$ m$^{-3}$ for field $CO_2$ measurement at least 40% to ±0.72 $mgCO_2$ m$^{-3}$ (i.e., accuracy at 0 or 40 °C when $T_c$ = 20 °C), which would be a significant improvement to ensure field $CO_2$ measurement accuracy through $CO_2$ zero and span procedures.

### 6.3.2 $H_2O$ zero and span procedures

Figure 4b shows that the $H_2O$ zero drift uncertainty increases as $T_a$ moves away from $T_c$ in the same trend as $CO_2$ zero drift uncertainty. Therefore, an $H_2O$ zero procedure can be performed in the same technique as for $CO_2$ zero procedure. $H_2O$ gain drift uncertainty has a different trend. It exponentially diverges, as $T_a$ increases away from $T_c$, to ±5.0 × 10$^{-2}$ g$H_2O$ m$^{-3}$ near 50 °C, and gradually converges by two orders smaller, as $T_a$ decreases away from $T_c$, to ±6.38 × 10$^{-4}$ g$H_2O$ m$^{-3}$ at −30 °C (data for Fig. 4b). The exponential divergence results from the linear relationship of $H_2O$ gain drift uncertainty (Eq. 19) with $\rho_{H2O}$, which exponentially increases (Eq. B1) with a $T_a$ increase away from $T_c$ for the same RH (Buck, 1981). The convergence results from the linear relationship offset by the exponential decrease in $\rho_{H2O}$ with a $T_a$ decrease for the same RH. This trend of $H_2O$ gain drift uncertainty with $T_a$ is a rationale to guide the $H_2O$ span procedure, which adjusts the $H_2O$ gain drift.

The $H_2O$ span procedure needs standard moist air with known $H_2O$ density from a dew point generator. The generator is not operational near or below freezing conditions (LI−COR Biosciences, 2004), which limits the span procedure to be performed only under non-freezing conditions. This condition, from 5 to 35 °C, may be considered for the generator to be conveniently operational in the field. Accordingly, the zero and span procedures for $H_2O$ should be discussed separately for a $T_a$ above and below 5 °C.

#### 6.3.2.1 $T_a$ above 5 °C

Looking at the right portion with $T_a$ above 5 °C in Fig. 4b, $H_2O$ gain drift has a more obvious impact on measurement uncertainty in a higher $T_a$ range (e.g., above $T_c$), within which the $H_2O$ span procedure is most needed. In this range, the maximum accuracy range of ±0.10 g$H_2O$ m$^{-3}$ can be narrowed by 30% to ±0.07 (assessed from data for Fig 3a) if the zero and span procedures for $H_2O$ can be sequentially performed as necessary in a $T_a$ range from 5 to 35 °C.

### 6.3.2.2 $T_a$ below 5 ºC

Looking at the left portion with $T_a$ below 5 ºC in Fig 4b, $H_2O$ gain drift has a less obvious contribution to the measurement uncertainty in a lower $T_a$ range (e.g., below 5 ºC), within which the $H_2O$ span procedure may be unnecessary. An $H_2O$ gain drift uncertainty at 5 ºC is 50% of the $H_2O$ zero drift uncertainty (dotted curve in Fig. 5). This percentage decreases to 3% at –30 ºC. On average, this percentage over a range of –30 to 5 ºC is 18% (assessed from data for dotted curve in Fig. 5). Thus, for $H_2O$ measurements over the lower $T_a$ range, it can be concluded that $H_2O$ zero drift is a major uncertainty source, and $H_2O$ gain drift is a minor uncertainty source.

A close examination of the other curves in Fig. 5 for the portion in the accuracy range from $H_2O$ zero/gain drift makes this conclusion more convincing. Given $T_c$ = 20, in accuracy range, the portion from $H_2O$ zero drift uncertainty is much greater (maximum 38% at –30 ºC) than that from $H_2O$ gain drift uncertainty (maximum only 7% at 5 ºC). On average over the lower $T_a$ range, the former is 27% and the latter only 4%. Further, given $T_c$ = 5 ºC, in the accuracy range, the portion from $H_2O$ gain drift uncertainty is even smaller (maximum only 3% at –5 ºC); in contrast, the portion from zero drift uncertainty is more major (one order higher, 30% at –30 ºC). On average over the lower $T_a$ range, the minor gain drift uncertainty is 1.7%, and the major zero drift uncertainty is 17%. Both percentages underscore that the $H_2O$ span procedure is reasonably unnecessary under cold/dry conditions, and, under such conditions, the $H_2O$ zero procedure is the only necessary option to efficiently minimize $H_2O$ measurement uncertainty in OPEC systems. This finding gives confidence in $H_2O$ measurement accuracy to users who are worried about $H_2O$ span procedures for infrared analyzers in the cold seasons when a dew point generator is not operational in the field (LI−COR Biosciences, 2004).

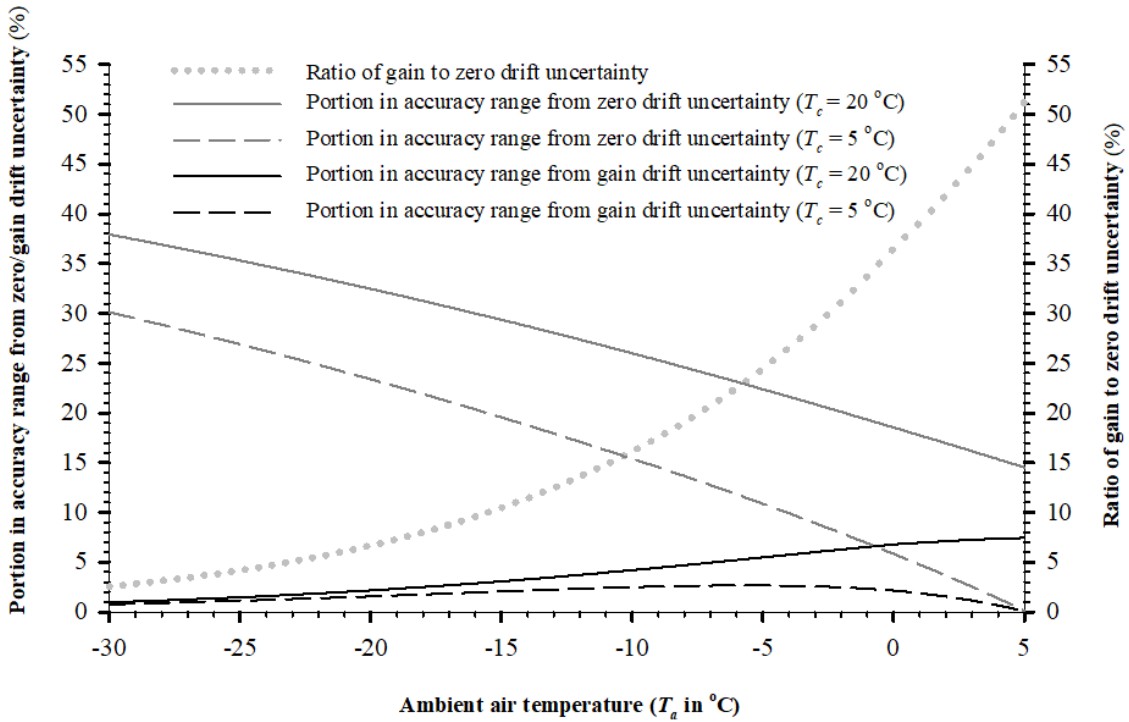

**Figure 5.** For a range of low $T_a$, the portion in the accuracy range from zero/gain drift uncertainty (left ordinate) and the ratio of gain to zero drift uncertainty (right ordinate). The curves are evaluated by Eqs. (18), (19), and (22) from measurement specifications for EC150 infrared $CO_2$–$H_2O$ analyzers (Campbell Scientific Inc, UT, USA) in open-path eddy-covariance flux systems over the $T_a$ range from –30 to 5 °C under atmospheric pressure of 101.325 kPa. $T_c$ is the ambient air temperature at which an analyzer was calibrated, zeroed, and/or spanned.

### 6.3.3 $H_2O$ zero procedure in cold and/or dry environments

In cold environments, although the non-operational $H_2O$ span procedure is unnecessary, the $H_2O$ zero procedure is asserted to be a prominently important option for minimizing the $H_2O$ measurement uncertainty in OPEC systems. This procedure, although operational under freezing conditions, is still inconvenient for users when weather is very cold (e.g., when $T_a$ is below –15 °C). If the field $H_2O$ zero procedure is performed as needed above this $T_a$ value, an OPEC system can be assumed to run at $T_a$ with ±20 °C of $T_c$. Under this assumption, the poorest $H_2O$ accuracy of ±0.066 g$H_2O$ m$^{-3}$ below 5 °C in Table 2 can be narrowed, through the $H_2O$ zero procedure, by at least 22% to 0.051 g$H_2O$ m$^{-3}$ (assessed from data for Fig. 3a). Correspondingly, the relative accuracy range can be narrowed by the same percentage. The $H_2O$ zero procedure can ensure both accuracy and relative accuracy of $H_2O$ measurements in a cold environment (Fratini et al., 2014). In a dry environment,

it plays the same role as in a cold environment, but it would be more convenient for users to perform the zero procedure if warmer.

In a cold and/or dry environment, $H_2O$ zero procedures that are undergone on a regular schedule would best minimize the impact of zero drifts on measurements. Under such an environment, the automatic zero procedure for $CO_2$ and $H_2O$ together in CPEC systems is an operational and efficient option to ensure and improve field $CO_2$ and $H_2O$ measurement
accuracies (Campbell Scientific Inc., 2021a; Zhou et al., 2021).

## 7 Discussion

An assessment methodology to evaluate the overall accuracies of field $CO_2$ and $H_2O$ measurements from the infrared analyzers in OPEC systems is developed using analyzer individual measurement uncertainties as specified using four uncertainty descriptors: zero drift, gain drift, sensitivity-to-$CO_2$/$H_2O$, and precision variability (Table 1). For the evaluation,
these uncertainty descriptors are comprehensively composited into the accuracy model (2) and then formulated as a $CO_2$ accuracy equation (14) and an $H_2O$ accuracy equation (22) (Sects. 3 to 5 and Appendix A). The assessment methodology, along with the model and the equations, presents our development for the objective (Sects. 4.5 and 5.4).

### 7.1 Accuracy model

Accuracy model (2) composites the four measurement uncertainties (zero drift, gain drift, sensitivity-to-$CO_2$/$H_2O$, and
precision variability) specified for analyzer performance as an accuracy range. This range is modeled as a simple addition of the four uncertainties. The simple addition is derived from our analysis assertion that the four measurement uncertainties interactionally or independently contribute to the accuracy range, but the contributions from the interactions inside any pair of uncertainties are negligible since they are three orders smaller in magnitude than an individual contribution in the pair (Appendix A). This derived model is simple and applicable, paving an approach to the formulation of accuracy equations
that are computable for evaluating the overall accuracies of field $CO_2$ and $H_2O$ measurements from infrared analyzers in OPEC systems.

Additionally, included in the accuracy model, the four types of measurement uncertainty sources (i.e., zero drift, gain drift, sensitivity-to-$CO_2$/$H_2O$, and precision variability) to specify the performance of infrared $CO_2$−$H_2O$ analyzers for OPEC systems have been consistently used over last two decades (LI−COR Biosciences, 2001; 2021b; 2021c; Campbell
Scientific Inc., 2021). With the advancement of optical technologies, the number of these uncertainty sources for analyzer specifications is not expected to increase rather some current uncertainty sources could be eliminated from the current specification list, even if not in the near future. If eliminated, in Models (3) and (15) and Eqs. (14) and (22), the parameters and variables related to the eliminated uncertainty sources could be easily removed for adoption of the new set of specifications for infrared $CO_2$−$H_2O$ analyzers.

## 7.2 Formulation of uncertainty terms in Model (2) for accuracy equations

In Sects. 4 and 5, each of the four uncertainty terms in accuracy model (2) is formulated as a computable sub-equation for $CO_2$ (Eqs. 4, 7, 11, and 13) and $H_2O$ (Eqs. 16, 18, 19, and 21), respectively. The accuracy model, whose terms are replaced with the formulated sub-equations for $CO_2$, becomes a $CO_2$ accuracy equation and, for $H_2O$, becomes an $H_2O$ accuracy equation. In the formulation, approximation is used for zero drift, gain drift, and sensitivity-to-$CO_2$/$H_2O$, while statistics are applied for precision variability.

For the zero/gain drift, although it is well known that the drift is influenced more by $T_a$ if housing $CO_2$−$H_2O$ accumulation is assumed to be minimized as insignificant under normal field maintenance (LI−COR Biosciences, 2021c; Campbell Scientific Inc., 2021b), the exact relationship of drift to $T_a$ does not exist. Alternatively, the zero/gain drift uncertainty is formulated by an approximation of drifts away from $T_c$ linearly in proportion to the difference between $T_a$ and $T_c$ but within its maximum range over the operational range in $T_a$ of OPEC systems (Eqs. 7, 11, 18, and 19). A drift uncertainty equation formulated through such an approximation is not an exact relationship of drift to $T_a$, but it does represent the drift trend, as influenced by $T_a$, to be understood by users. The accuracy from this equation at a given $T_a$ is not exact either, but the maximum range over the full range, which is the most likelihood estimation, is most needed by users.

In fact, the $H_2O$ accuracy as influenced by the linear trend of zero and gain drifts with the difference between $T_a$ and $T_c$ is shadowed by the exponential trend of saturated $H_2O$ density with $T_a$ (Fig. 4b). Similarly, the $CO_2$ accuracy as influenced by the linear trend of zero and gain drifts with this difference is dominated by the $CO_2$ density of the ecosystem background with $T_a$, particularly in the low temperature range (Fig. 2). Ultimately, the assumed linear trend does not play a dominant role in the accuracy trends of $CO_2$ and $H_2O$, which shows the merits of our methodology in the uses of atmospheric physics and biological environment principles for the field data.

The sensitivity-to-$CO_2$/$H_2O$ uncertainty can be formally formulated as Eq. (20) or (12), but, if directly used, this formulation would add an additional variable to the $CO_2$/$H_2O$ accuracy equation. Equation (12) would add $H_2O$ density ($\rho_{H_2O}$) to the $CO_2$ accuracy equation, and Eq. (20) would add $CO_2$ density ($\rho_{CO_2}$) to the $H_2O$ accuracy equation. For either accuracy equation, the additional variable would complicate the uncertainty analysis. According to the ecosystem environment background, the maximum range of sensitivity-to-$CO_2$/$H_2O$ uncertainty is known and, as compared to the major uncertainty of zero/gain drift (Table 1), this range is narrow (Table 1 and Eqs. 13 and 21). Therefore, the sensitivity-to-$CO_2$/$H_2O$ uncertainty is approximated as Eq. (21) or (13). This approximation widens the accuracy range slightly, in a magnitude smaller than each of major uncertainties from the drifts at least in one order; however, it eliminates the need for $\rho_{H_2O}$ in the $CO_2$ accuracy equation and for $\rho_{CO_2}$ in the $H_2O$ accuracy equation, which makes the equations easily applicable.

Precision uncertainty is statistically formulated as Eq. (4) for $CO_2$ and Eq. (16) for $H_2O$. This formulation is a common practice based on statistical methods (Hoel, 1984).

### 7.3 Use of relative accuracy for infrared analyzer specifications

Relative accuracy is often used concurrently with accuracy to specify sensor measurement performance. The accuracy is the numerator of relative accuracy whose denominator is the true value of a measured variable. When evaluated for the applications of OPEC systems in ecosystems, $CO_2$ accuracy in magnitude is small in a range within one order (0.39 ~ 1.22 mg$CO_2$ m$^{-3}$, data for Fig. 2a), and so is $H_2O$ accuracy (0.04 ~ 0.10 g$H_2O$ m$^{-3}$, data for Fig. 3a). In ecosystems, $CO_2$ is naturally high, as compared to its accuracy magnitude, and does not change much in terms of a magnitude order (e.g., no more than one order from 600 to 1,600 g$H_2O$ m$^{-3}$, assumed in this study). However, unlike $CO_2$, $H_2O$ naturally changes in its amount dramatically across at least three orders in magnitude (e.g., at 101.325 kPa, from 0.03 g$H_2O$ m$^{-3}$ when RH is 10% at –30 °C to 40 g$H_2O$ m$^{-3}$ when dew point temperature is 35 °C at the highest as reported by National Weather Service (2022); under drier conditions, the $H_2O$ amount could be even lower). Because, in ecosystems, $CO_2$ changes differently from $H_2O$ in amount across magnitude orders, the relative accuracy behaviors in $CO_2$ differ from $H_2O$ (Figs. 2b and 3b).

### 7.3.1 $CO_2$ relative accuracy

Because of the small $CO_2$ accuracy magnitude relative to the natural $CO_2$ amount in ecosystems, the $CO_2$ relative accuracy magnitude varies within a narrow range of ±0.07 to ±0.19% (Sect. 4.5.2). If the relative accuracy is used, either a range of ±0.07 − ±0.19% or an inequality of ≤ 0.19% can be specified as the $CO_2$ relative accuracy magnitude for field $CO_2$ measurements. Both range and inequality would be equivalently perceived by users to be a fair performance of OPEC systems. For simplicity, our study with the OPEC systems can be specified for their $CO_2$ relative accuracy to be ±0.19%.

### 7.3.2 $H_2O$ relative accuracy

Although the $H_2O$ accuracy magnitude is also small, the "relatively" great change in natural air $H_2O$ across several magnitude orders in ecosystems results in a much wider range of the $H_2O$ relative accuracy magnitude, from ±0.23% at maximum air moisture to ±96% when RH is 20% at –30 °C (Fig. 3b and Sect. 5.4.2). $H_2O$ relative accuracy can be much greater under dry conditions at low $T_a$ (e.g., ±192% for air when RH is 10% at –30 °C). Accordingly, if the relative accuracy is used, either a range of ±0.23 − ±192% or an inequality of ≤ 192% can be specified as the $H_2O$ relative accuracy magnitude for field $H_2O$ measurements. Either ±0.23 − ±192% or ≤ 192% could be perceived by users intrinsically as poor measurement performance of the infrared analyzers, although either specification is conditionally right for fair $H_2O$ measurement.

Apparently, the relative accuracy for $H_2O$ measurements in ecosystems is not intrinsically interpretable by users to correctly perceive the performance of the infrared analyzers in OPEC systems. Instead, if $H_2O$ relative accuracy is unconditionally specified just in an inequality of ≤ 192%, it could easily mislead users to wrongly assess the performance as unacceptable for $H_2O$ measurements, although this performance of the infrared analyzers in OPEC systems is fair for air when RH is 10% at –30 °C. Therefore, $H_2O$ relative accuracy is not recommended to be used for specification of infrared

analyzers for $H_2O$ measurement performance. If this descriptor is used, the $H_2O$ relative accuracy under a standard condition should be specified. This condition may be defined as saturated air at 35 ℃ (i.e., the highest natural dew point (National Weather Service, 2022)) under normal $P$ of 101.325 kPa (Wright et al., 2003). For our study case, under such a standard condition, the $H_2O$ relative accuracy can be specified within ±0.18% after manufacturing calibration (data for Fig. 3b).

## 8 Conclusions

The accuracy of field $CO_2/H_2O$ measurements from the infrared analyzers in OPEC systems can be defined as a maximum range of composited measurement uncertainty (Eqs. 14 and 22) from the specified sources: zero drift, gain drift, sensitivity-to-$CO_2/H_2O$, and precision variability (Table 1), all of which are included in the system specifications for infrared $CO_2$-$H_2O$ analyzers currently used in field OPEC systems. The specified uncertainties interactionally or independently contribute to the overall uncertainty. Fortunately, the interactions between component uncertainties in each pair is three orders smaller than either component individually (Appendix A). Therefore, these specified uncertainties can be simply added together as the accuracy range in a general $CO_2/H_2O$ accuracy model for OPEC systems (Model 2). Based on statistics, bio-environment, and approximation, the specification descriptors of the infrared analyzers in OPEC systems are incorporated into the model terms to formulate the $CO_2$ accuracy equation (14) and the $H_2O$ accuracy equation (22), both of which are computable to evaluate corresponding $CO_2$ and $H_2O$ accuracies. For the EC150 infrared analyzers used in the OPEC systems over their operational range in $T_a$ at the standard $P$ of 101.325 kPa (Figs. 2 and 3 and Table 2), the $CO_2$ accuracy can be specified as ±1.22 mg$CO_2$ m$^{-3}$ (relatively within ±0.19%, Fig. 2) and $H_2O$ accuracy as ±0.10 g$H_2O$ m$^{-3}$ (relatively within ±0.18% for saturated air at 35 ℃ at the standard $P$, Fig. 3).

Both accuracy equations are not only applicable for further uncertainty estimation for $CO_2$ and $H_2O$ fluxes due to $CO_2$ and $H_2O$ measurement uncertainties (Eqs. 27 and 28) and the error/uncertainty analyses in $CO_2$ and $H_2O$ data applications (e.g., Eq. 29), but they also may be used as a rationale to assess and guide field maintenance on infrared analyzers. Equation (14) as shown in Fig. 2a, along with Eqs. (7) and (11) as shown in Fig. 4a, guides users to adjust the $CO_2$ zero and $CO_2$ gain drifts, through the corresponding zero and span procedures, near a $T_a$ value that minimizes the $T_a$ departures, on average, during the period of interest if this period were not under extreme and hazard conditions (Fratini et al., 2014). As assessed on atmospheric $CO_2$ background, the procedures can narrow the maximum $CO_2$ accuracy range by 40%, from ±1.22 to ±0.72 mg$CO_2$ m$^{-3}$, and thereby greatly improve the $CO_2$ measurement accuracies with these regular zero and span procedures for $CO_2$.

Equation (22) as shown in Fig. 3a, along with Eqs. (18) and (19) as shown in Fig. 4b, presents users with a rationale to adjust the $H_2O$ zero drift of analyzers in the same technique as for $CO_2$, but the $H_2O$ gain drift under hot and humid environments needs more attention (see the right portion above $T_c$ in Figs. 3a and 4b); under cold and/or dry environments, it needs no further concern (see the left portion below 0 ℃ in Fig. 4b). In a $T_a$ range above 5 ℃, the maximum $H_2O$ accuracy range of ±0.10 g$H_2O$ m$^{-3}$ can be narrowed by 30% to ±0.07 g$H_2O$ m$^{-3}$ if both zero and span procedures for $H_2O$ are

performed as necessary. In a $T_a$ range below 5 ℃, the $H_2O$ zero procedure alone can narrow the maximum $H_2O$ accuracy range of $\pm 0.066$ g$H_2O$ m$^{-3}$ by 22%, to $\pm 0.051$ g$H_2O$ m$^{-3}$. Under cold environmental conditions, the $H_2O$ span procedure is found to be unnecessary (Fig. 5), and the $H_2O$ zero procedure is proposed as the only, and prominently efficient, option to minimize $H_2O$ measurement uncertainty in OPEC systems. This procedure plays the same role under dry conditions. Under cold and/or dry environments, the zero procedure for $CO_2$ and $H_2O$ together would be a practical and efficient option not only to warrant, but also to improve, measurement accuracy. In a cold environment, adjusting the $H_2O$ gain drift is impractical because of the failure of a dew point generator under freezing conditions.

Additionally, as a specification descriptor for OPEC systems used in ecosystems, relative accuracy is applicable for $CO_2$ instead of $H_2O$ measurements. A small range in the $CO_2$ relative accuracy can be perceived intuitively by users as normal. In contrast, without specifying the condition of air moisture, a large range in $H_2O$ relative accuracy under cold and/or dry conditions (e.g., 100%) can easily mislead users to an incorrect conclusion in interpretation of $H_2O$ measurement reliability, although, it is the best achievement of the modern infrared analyzers under such conditions. If the $H_2O$ relative accuracy is used, the authors suggest to conditionally define it for saturated air at 35 ℃ (i.e., 39.66 g$H_2O$ m$^{-3}$ at 101.352 kPa). Ultimately, this study provides some scientific bases for the flux community to specify the accuracy of $CO_2$–$H_2O$ measurements from infrared analyzers in OPEC systems although only one model of infrared analyzers (i.e., EC150) is used for this study.

**Appendix A: Derivation of the accuracy model for infrared $CO_2$–$H_2O$ analyzers**

As defined in the Introduction, the measurement accuracy of infrared $CO_2$–$H_2O$ analyzers is a range of the difference between the true $\alpha$ density ($\rho_{\alpha T}$, where $\alpha$ can be either $H_2O$ or $CO_2$) and measured $\alpha$ density ($\rho_\alpha$) by the analyzer. The difference is denoted by $\Delta\rho_\alpha$, given by Eq. (1) in Sect. 3. The range of this difference is contributed from the analyzer performance uncertainties, as specified by use of the four descriptors: zero drift, gain drift, cross-sensitivity, and precision (LI–COR Biosciences, 2021c; Campbell Scientific Inc., 2021b).

According to the definitions in Sect. 2, zero drift uncertainty ($\Delta\rho_\alpha^z$) is independent of $\rho_{\alpha T}$ value and gain trend related to analyzer response; so, too, is cross-sensitivity uncertainty ($\Delta\rho_\alpha^s$), which depends upon the amount of background $H_2O$ in the measured air if $\alpha$ is $CO_2$, and upon the amount of background $CO_2$ in the measured air if $\alpha$ is $H_2O$. In the case that both gain drift and precision uncertainties are zero, $\Delta\rho_\alpha^z$ and $\Delta\rho_\alpha^s$ are simply additive to any true value as a measured value, including zero drift and cross-sensitivity uncertainties ($\rho_{\alpha\_zs}$)

$$\rho_{\alpha\_zs} = \rho_{\alpha T} + \Delta\rho_\alpha^z + \Delta\rho_\alpha^s, \tag{A1}$$

where subscript $z$ indicates zero drift uncertainty included in the measured value, and subscript $s$ indicates cross-sensitivity uncertainty included in the measured value. During the measurement process, while zero is drifting and cross-sensitivity is active, if gain also drifts, then the gain drift interacts with the zero drift and the cross-sensitivity. This is because $\rho_{\alpha\_zs}$ is a

790 linear factor for this gain drift (see the cells along the gain drift row in the value columns in Table 1) that is added to $\rho_{\alpha\_zs}$ as a measured value additionally including gain drift uncertainty ($\rho_{\alpha\_zsg}$, where subscript $g$ indicates gain drift uncertainty included in the measured value), given by

$$\rho_{a\_zsg} = \rho_{\alpha\_zs} + \delta_{\alpha\_g}\rho_{\alpha\_zs}, \tag{A2}$$

where $\delta_{\alpha\_g}$ is gain drift percentage ($\delta_{CO2\_g} = 0.10\%$ and $\delta_{H2O\_g} = 0.30\%$, Table 1). Substituting $\rho_{\alpha\_zs}$, as expressed in Eq. (A1),

into this equation leads to

$$\rho_{a\_zsg} = \rho_{\alpha T} + \Delta\rho_{\alpha}^{z} + \Delta\rho_{\alpha}^{s} + \delta_{\alpha\_g}\rho_{\alpha T} + \delta_{\alpha\_g}\Delta\rho_{\alpha}^{z} + \delta_{\alpha\_g}\Delta\rho_{\alpha}^{s}. \tag{A3}$$

In this equation, $\delta_{\alpha\_g}\Delta\rho_{\alpha}^{z}$ is the zero-gain interaction, and $\delta_{\alpha\_g}\Delta\rho_{\alpha}^{s}$ is the cross-sensitivity-gain interaction. In magnitude,

the former is three orders smaller than either zero drift uncertainty ($\Delta\rho_{\alpha}^{z}$) or gain drift uncertainty ($\delta_{\alpha\_g}\rho_{\alpha T}$) and the latter is

three orders smaller than either cross-sensitivity uncertainty ($\Delta\rho_{\alpha}^{s}$) or gain drift uncertainty. Therefore, both interactions are

800 relatively small and can be reasonably dropped. As a result, Eq. (A3) can be approximated and rearranged as:

$$\begin{aligned}\rho_{a\_zsg} &\approx \rho_{\alpha T} + \Delta\rho_{\alpha}^{z} + \delta_{\alpha\_g}\rho_{\alpha T} + \Delta\rho_{\alpha}^{s}\\ &= \rho_{\alpha T} + \Delta\rho_{\alpha}^{z} + \Delta\rho_{\alpha}^{g} + \Delta\rho_{\alpha}^{s}\end{aligned}, \tag{A4}$$

where $\Delta\rho_{\alpha}^{g}$ is gain drift uncertainty (i.e., $\delta_{\alpha\_g}\Delta\rho_{\alpha}^{z}$). Any measured value has random error (i.e., precision uncertainty)

independent of $\rho_{\alpha T}$ in value (ISO, 2012). Therefore, $\rho_{\alpha\_zsg}$ plus precision uncertainty ($\Delta\rho_{\alpha}^{p}$) is the measured value including

all uncertainties ($\rho_{\alpha}$), given by

805 $$\rho_{a} = \rho_{a\_zsg} + \Delta\rho_{\alpha}^{p}. \tag{A5}$$

The insertion of Eq. (A4) into this equation leads to

$$\rho_{a} - \rho_{\alpha T} = \Delta\rho_{\alpha}^{z} + \Delta\rho_{\alpha}^{g} + \Delta\rho_{\alpha}^{s} + \Delta\rho_{\alpha}^{p}. \tag{A6}$$

This equation holds

$$\Delta\rho_{a} \leq \left|\Delta\rho_{\alpha}^{z}\right| + \left|\Delta\rho_{\alpha}^{g}\right| + \left|\Delta\rho_{\alpha}^{s}\right| + \left|\Delta\rho_{\alpha}^{p}\right|. \tag{A7}$$

The range of the right side of this equation is wider than the measurement uncertainty from all measurement uncertainty sources, as shown on the right side of Eq. (A6), and the difference of $\rho_{\alpha}$ minus $\rho_{\alpha T}$ (i.e., $\Delta\rho_{\alpha}$). Using this range, the measurement accuracy is defined in Model (2) in Sect. 3.

**Appendix B: Water vapor density from ambient air temperature, relative humidity, and atmospheric pressure**

Given ambient air temperature ($T_a$ in °C) and atmospheric pressure ($P$ in kPa), air has a limited capacity to hold an amount

of water vapor (Wallace and Hobbs, 2006). This limited capacity is described in terms of saturation water vapor density ($\rho_s$ in gH$_2$O m$^{-3}$) for moist air, given through the Clausius−Clapeyron equation (Sonntag, 1990; Wallace and Hobbs, 2006)

$$\rho_s\left(T_a,P\right)=\frac{0.6112\,f\left(P\right)}{R_v\left(273.15+T_a\right)}\begin{cases}\exp\left(\dfrac{17.62T_a}{T_a+243.12}\right) & T_a\geq0\\[3mm]\exp\left(\dfrac{22.46T_a}{T_a+272.62}\right) & T_a<0\end{cases},\tag{B1}$$

where $R_v$ is the gas constant for water vapor ($4.61495\times10^{-4}$ kPa m$^3$ K$^{-1}$ gH$_2$O$^{-1}$), and $f(P)$ is an enhancement factor for moist air, being a function of $P$: $f(P)=1.0016+3.15\times10^{-5}P-0.0074P^{-1}$. At relative humidity (RH in %), the water vapor density [ $\rho_{H_2O}^{RH}\left(T_a,P\right)$ in gH$_2$O m$^{-3}$] is

$$\rho_{H_2O}^{RH}\left(T_a,P\right)=\text{RH}\rho_s\left(T_a,P\right).\tag{B2}$$

This equation, along with Eq. (B1), is used to calculate $\rho_{H_2O}^{RH}$ used in Fig. 3 in Sect. 5.4 and Figs. 4b and 5 in Sect. 6.3.

**Appendix C: The relationship of measured to "true" covariance of vertical wind speed with CO$_2$, H$_2$O, or air temperature**

For open-path eddy-covariance systems, the computation of CO$_2$/H$_2$O flux between ecosystems and the atmosphere starts from covariance of an individual 3-D wind component with a CO$_2$/H$_2$O density. To express the covariance, as similarly used in Eqs. (1), $\alpha$ is used as a subscript of $\rho$ to represent either CO$_2$ or H$_2$O and subscript $T$ is used to indicate a measurement free of uncertainty as if it were "true". According to Eq. (1), a measured $\alpha$ density ($\rho_\alpha$) with a measurement uncertainty ($\Delta\rho_\alpha$) can be expressed as

$$\rho_\alpha=\rho_{\alpha T}+\Delta\rho_\alpha,\tag{C1}$$

where $\rho_{\alpha T}$ is an assumed $\alpha$ density free of measurement uncertainty as if measured by an accurate sensor with the same frequency response as the one measuring $\rho_\alpha$. This assumed $\alpha$ density ($\rho_{\alpha T}$) is also referred to as "true $\alpha$ density" although not. The covariance of vertical wind speed ($w$) with $\rho_\alpha$ is given by

$$\overline{w'\rho_\alpha'}=\frac{1}{n}\sum_{i=1}^{n}\left(w_i-\bar{w}\right)\left(\rho_{\alpha i}-\overline{\rho_\alpha}\right),\tag{C2}$$

where $n$ is the sample number over an averaging interval (e.g., 36,000 over an hour interval if $w_i$ and $\rho_{\alpha i}$ are measured at 10 Hz), subscript $i$ indexes the sequential numbers for $w_i$ and $\rho_{\alpha i}$, the overbar is the Reynolds' averaging operator, and prime denotes the fluctuation of a variable away from its mean (e.g., $w_i'=w_i-\overline{w}$). Without considering the measurement error of $w$ for this study topic, submitting Eq. (C1) into (C2) leads to

$$\overline{w'\rho_\alpha'} = \frac{1}{n}\sum_{i=1}^{n}(w_i - \overline{w})\left[\rho_{\alpha Ti} + \Delta\rho_{\alpha i} - \overline{(\rho_{\alpha T} + \Delta\rho_\alpha)}\right]$$

$$= \frac{1}{n}\sum_{i=1}^{n}(w_i - \overline{w})(\rho_{\alpha Ti} - \overline{\rho}_{\alpha T}) + \frac{1}{n}\sum_{i=1}^{n}(w_i - \overline{w})(\Delta\rho_{\alpha i} - \Delta\overline{\rho}_\alpha) \tag{C3}$$

Within an averaging interval (e.g., an hour), the systematic error components inside terms $\Delta\rho_{\alpha i}$ and $\Delta\overline{\rho}_\alpha$ are not only constant, but also equal. Accordingly, the systematic errors inside the term $\Delta\rho_{\alpha i} - \Delta\overline{\rho}_\alpha$ are cancelled out (Richardson et al., 2012). In essence, this term is a random error whose statistical distribution generally is assumed to be normal with a zero mean (i.e., $\Delta\rho_{\alpha i} - \Delta\overline{\rho}_\alpha$ is expected to be zero. Hoel, (1984)). The correlation of $w$ with a random variable normally distributed with an expected zero mean tends to be zero, particularly for a large sample of 36,000 under discussion, even 18,000 for half hours

(Snedecor and Cochran, 1989), which is the shortest period commonly used for flux computations. Accordingly, the second term in the second line of Eq. (C3) can be considered as zero. Therefore, the covariance of $w$ with measured $\alpha$ density is equal to the covariance of $w$ with the true $\alpha$ density, given by

$$\overline{w'\rho_\alpha'} = \overline{w'\rho_{\alpha T}'} \,. \tag{C4}$$

If $w$ from a sonic anemometer and $\rho_\alpha$ from an infrared analyzer are not measured through spatial and temporal

synchronization, the values of covariance of $w$ with $\rho_\alpha$ in the different lags of measurement (hereafter referred to as the lagged covariance) are computed for use in the lag maximization to find their maximum covariance as if $w$ and $\rho_\alpha$ were measured at the same time in the same space (Moncrieff et al., 1997; Ibrom et al., 2007). Each lagged covariance from field measurements can be expressed as $\overline{w'\rho_{\alpha l}'}$, where subscript $l$ is the index for a lag number. If $l = i$, $w_i$ and $\rho_{\alpha l}$ were measured at the same time. If $l = i - 1$, $w_i$ was measured one measurement interval (i.e., 100 ms for 10-Hz measurements) later than $\rho_{\alpha l}$

whereas $w_i$ was measured one measurement interval earlier than $\rho_{\alpha l}$ if $l = i+1$. The index $l$ can be $-k$ to $k$ where $k$ is a positive integer, including 0, to represent the maximum number of the lags that is optional to users. Therefore, given $l$ from $-k$ to $k$, the number of $\overline{w'\rho_{\alpha l}'}$ values is $2k+1$. Using the same approach to Eq. (C4), $\overline{w'\rho_{\alpha l}'} = \overline{w'\rho_{\alpha Tl}'}$ can be proved.

The lagged covariance values for $\overline{u'\rho_{\alpha l}'}$ and $\overline{v'\rho_{\alpha l}'}$ ($l$ is $-k$, $-k+1$, …,0, …, or $k$) are also computed for each lag where, in the sonic anemometer coordinate system, $u$ is the wind speed in the $x$ direction and $v$ is the wind speed in the $y$ direction.

Both $\overline{u'\rho_{\alpha l}'} = \overline{u'\rho_{\alpha Tl}'}$ and $\overline{v'\rho_{\alpha l}'} = \overline{v'\rho_{\alpha Tl}'}$ are also can be proved in the same way for Eq. (C4). Given the rotation angles from $\overline{u}, \overline{v}, \overline{w}, \overline{u^2}, \overline{v^2}, \overline{w^2}, \overline{u'v'}, \overline{u'w'}$, and $\overline{v'w'}$ (Tanner and Thurtell, 1960), each set of $\overline{u'\rho_{\alpha l}'}, \overline{v'\rho_{\alpha l}'}$, and $\overline{w'\rho_{\alpha l}'}$ are rotated to be $\left(\overline{u'\rho_{\alpha l}'}\right)_r, \left(\overline{v'\rho_{\alpha l}'}\right)_r$, and $\left(\overline{w'\rho_{\alpha l}'}\right)_r$, respectively, where $u$, $v$, and $w$ through the rotations are transformed into the natural wind coordinate system correspondingly as stream-wise, lateral, and vertical wind speeds. In the rotation process, $\rho_\alpha$ is not additionally involved. Because $\rho_{\alpha l}'$ inside the covariance is a scalar rather than a vector variable, the rotation would not be

influenced by $\bar{\rho}_{al}$ and $\overline{\rho_{al}^2}$ as by the three means and three variance values of 3-D wind components (Tanner and Thurtell, 1960). Because the same set of rotation angles are also used for the rotations of $\overline{u'\rho_{aTl}'}, \overline{v'\rho_{aTl}'}$, and $\overline{w'\rho_{aTl}'}$, the covariance values rotated from these three covariance values are correspondingly equal to those rotated from $\overline{u'\rho_{al}'}, \overline{v'\rho_{al}'}$, and $\overline{w'\rho_{al}'}$, given by

$$\left(\overline{w'\rho_{al}'}\right)_r = \left(\overline{w'\rho_{aTl}'}\right)_r. \tag{C5}$$

Therefore, from the lag maximization (Moncrieff et al., 1997; Ibrom et al., 2007), the maximum covariance in magnitude among $\left(\overline{w'\rho_{al}'}\right)_r$ ($l$ from $-k$ to $k$) is equal to the maximum in magnitude among $\left(\overline{w'\rho_{aTl}'}\right)_r$. Denoting the former maximum covariance by $\left(\overline{w'\rho_{a}'}\right)_{rm}$, where subscript $m$ indicates the maximum, and the latter one by $\left(\overline{w'\rho_{aT}'}\right)_{rm}$, this equality leads to

$$\left(\overline{w'\rho_{a}'}\right)_{rm} = \left(\overline{w'\rho_{aT}'}\right)_{rm}. \tag{C6}$$

For flux computations, both covariance values in this equation need further corrections for their low- and high-frequency
loss (Moore, 1986). The correction factor for $\left(\overline{w'\rho_{a}'}\right)_{rm}$ can be denoted by $f_{ca}$ and for $\left(\overline{w'\rho_{aT}'}\right)_{rm}$ can be denoted by $f_{caT}$. Both $f_{ca}$ and $f_{caT}$ are integrated in the same way from the cospectrum of $w$ with a scalar as represented by $T_a$ (air temperature) and the transfer functions of high-frequency loss separately for $w$ and $\alpha$ density (Moore, 1986; van Dijk, 2002), and low-frequency loss for Reynolds' averaging $\overline{w'\rho_{a}'}$ (Massman, 2000). Although depending on the structure of boundary-layer turbulent flows (Kaimal and Finnigan, 1994), under the same boundary-layer turbulent flows, the cospectrum for $w$ with $\rho_a$ is the same as for
$w$ with $\rho_{aT}$. Because the sensor for $\rho_{aT}$ is assumed to have the same frequency response as the sensor for $\rho_a$, both sensors have the same high-frequency loss, sharing the same transfer function (Moore, 1986). The transfer function for low-frequency loss due to Reynolds' averaging either side of Eq. (C6) is also used for its other side (Massman, 2000). Therefore, $f_{ca}$ is equal to $f_{caT}$, which, from Eq. (C6), leads to

$$f_{ca}\left(\overline{w'\rho_{a}'}\right)_{rm} = f_{caT}\left(\overline{w'\rho_{aT}'}\right)_{rm}. \tag{C7}$$

In this equation, the left term is the frequency-corrected $\left(\overline{w'\rho_{a}'}\right)_{rm}$, which can be denoted by $\left(\overline{w'\rho_{a}'}\right)_{rmf}$ where subscript $f$ indicates this covariance to be corrected for frequency loss, and the right term is the frequency-corrected $\left(\overline{w'\rho_{aT}'}\right)_{rm}$, which can be denoted by $\left(\overline{w'\rho_{aT}'}\right)_{rmf}$ (Moore, 1986; Massman, 2000; van Dijk, 2002). Accordingly, Eq. (C7) becomes

$$\left(\overline{w'\rho_{a}'}\right)_{rmf} = \left(\overline{w'\rho_{aT}'}\right)_{rmf}, \tag{C8}$$

where subscript *rmf* indicates the covariance was corrected through coordinate rotations (*r*), lag maximization (*m*), and low-
and high-frequency corrections (*f*). Equation (C8) shows the covariance of *w* with measured $\rho_\alpha$ is equal to its counterpart of *w* with true $\rho_\alpha$ even after a series of corrections before used to calculate α flux through Webb-Pearman-Leuning (WPL) corrections (Webb et al., 1982).

For the covariance of *w* with $T_a$, the same conclusion can be derived, given by

$$\overline{\left(w'T_a'\right)}_{rmf} = \overline{\left(w'T_{aT}'\right)}_{rmf} \tag{C9}$$

Assume *w* to be an accurate value for this study topic, through WPL corrections, $\overline{\left(w'\rho_\alpha'\right)}_{rmf}$ and $\overline{\left(w'T_\alpha'\right)}_{rmf}$ can be used to derive an analytical equation for α flux from $\rho_\alpha$ with an error as ranged by its accuracy and $T_a$ with its error specified for the air temperature sensor whereas $\overline{\left(w'\rho_{\alpha T}'\right)}_{rmf}$ and $\overline{\left(w'T_{\alpha T}'\right)}_{rmf}$ can be used to derive an analytical equation for α flux from $\rho_{\alpha T}$ and $T_{aT}$, each of which is assumed not to include an error. The comparison of both analytical equations derived after the WPL corrections can demonstrate the partial effects of $\Delta\rho_\alpha$ on the uncertainty of α flux data (see Sect. 6.2).

**Author Contributions**

XZ, TG, NZ, and BY contributed equally to this this work; YL, FY, and TA discussed the points of this study topic and made comments on the manuscript; JZ led the team.

**Competing interest**

XZ, BY, and YL have affiliation with Campbell Scientific Incorporation, which is the manufacturer of the example model EC150 of infrared $CO_2-H_2O$ analyzers. The other authors declare that they have no conflict of interest.

**Acknowledgments**

Authors thank anonymous reviewers for their rigorous and dedicated review, understanding of our study topic, and constructive comments on the manuscript for significant improvement, Brittney Smart for her dedicated revision, Linda Worlton-Jones for her prelimary professional proofreading, and Kati Kovacs for her final professional proofreading.

**Financial support**

This research has been supported by the Strategic Priority Research Program of the Chinese Academy of Sciences (grant no. XDA19030204), Campbell Scientific Research and Development, Campbell Scientific Inc. (project no. 14433), National

Key Research and Development Program of China (grant no. 2016YFD0600206), and Long-Term Agroecosystem Research,
USDA (award no. 58-3042-9-014).

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
