# Peer review of "Accuracies of field CO2–H2O data from open-path eddy-covariance flux systems: Assessment based on atmospheric physics and biological environment"

_Geoscientific Instrumentation, Methods and Data Systems, 2022_

## Author Comment (AC1)

Ker Research and Development
Institute of Applied Ecology
Chinese Academy of Sciences
72 Wenhua Road, Shenyang
Liaoning, 110016, China

April 9, 2022

RE: Responses to reviewer's comments on manuscript gi-2022-1

Dr. Grimaldi, Associate Editor
Geoscientific Instrumentation, Methods and Data Systems

Dear Dr. Grimaldi,

We have been really appreciated with reviewer #1's strong positive comments on the significance of this manuscript as a completion of systematic study of overall accuracies of field $CO_2$ and $H_2O$ data from infrared gas analyzers in both closed-path and open-path eddy covariance flux systems.

We are happy with thoroughly address the technical and editorial comments from reviewer #1 in the final revision while addressing upcoming comments from reviewer #2. Here, we are briefly responding the major and minor comments from reviewer #1 below.

**Major comments**

I have two open-path analyzers, i.e., EC150 and LI-COR 7500. In practice, when I perform a zero calibration, I always found a positive zero drift about 10 $\mu$mol mol$^{-1}$ for LI-COR 7500 at ambient temperature, slightly higher in the unit of mg $CO_2$ m$^{-3}$ and much higher than the upper of the values in the manuscript, but a much smaller accuracy due to gain drift when tubing the $CO_2$ span gas of 500-$\mu$mol mol$^{-1}$ after a zeroing operation. I speculate that this was caused a non-negligible housing $CO_2$/$H_2O$ accumulation, although the chemicals in the internal cell needs no replacement of new ones, i.e., after a zero calibration the analyzer works well for months. This is the same for $H_2O$ density. Therefore, in practice, I recommend the author give a short discussion of the possibility of field drift of zero and gain using the big data of analyzer-supplier, for example, that from EC150 in the lab of CSI, in the 6.3 section. These data may be helpful for providing suggestions for new users.

**Response**

Yes, an individual infrared $CO_2$−$H_2O$ gas analyzer may behaviors differently due to unexpected reasons. For this study, we must use the specifications of analyzers from their manufacturer. Our assessment must be based on the official specifications from manufacturer. We are not sure whether the data from field individual analyzers are valid because no benchmark data are available to assess the field data, which is the reason we assess the overall accuracies for field $CO_2$ and $H_2O$ data based on atmospheric physics and ecological background.

**Minor comments**

1.      Title: "$CO_2$−$H_2O$" (and in the text). I understand the authors wanted to identify both gas types using "−" from one of the two gas types using "/". In my opinion, however,

"$CO_2/H_2O$" may be better, just the same as they are in the profile system. The same for other parts of the manuscript.

**Response**
We also preferred "$CO_2/H_2O$", but "$CO_2/H_2O$" means "$CO_2$ or $H_2O$" and "$CO_2-H_2O$" means "$CO_2$ and $H_2O$". "$CO_2-H_2O$" is the editorial choice for this expression.

2.      L24: For a background concentration of atmospheric $CO_2$?

**Response**
The background concentration of atmospheric $CO_2$ is reported by Global Monitoring Laboratory and is used globally. The details about this background concentration are given in the paragraph of lines 95 to 100. In abstract, there is no room to describe what is a background concentration of atmospheric $CO_2$.

3.      L27-29: I recommend deleting "Under freezing conditions, an $H_2O$ span is both impractical and unnecessary, but the zero procedure becomes imperative to minimize $H_2O$ measurement uncertainty.", because there was some overlap of this sentence with the next one "In cold/dry conditions, the zero procedure for $H_2O$, along with $CO_2$, is an operational and efficient option to ensure and improve $H_2O$ accuracy".

**Response**:
We discuss two issues:
a.   $H_2O$ span
"Under freezing conditions, an $H_2O$ span is both impractical and unnecessary, but the zero procedure becomes imperative to minimize $H_2O$ measurement uncertainty."

b.   $H_2O$ zero
"In cold/dry conditions, the zero procedure for $H_2O$, along with $CO_2$, is an operational and efficient option to ensure and improve $H_2O$ accuracy".

Both sentences are not overlap each other

4.      *L36: delete "fluctuations", for consistency with "3-D wind and sonic temperature".*
Response:
Yes, this word can be removed. It may be redundant although the word can reflect the nature of turbulence measurements.

5.      *L75: "$CO_2/H_2O$ molar mixing ratio" or "$CO_2/H_2O$ dry molar fraction" is better.*
Response:
The former is more popularly use. $CO_2/H_2O$ molar mixing ratio is used in manual of close-path eddy-covariance systems and in AmeriFlux variable names.

6.      *L108: "in practice"?*
Response:
"In practice" can be used to replace "in applications".

7.      *L170: Possibly, use "the analyzer often gradually reports that this zero $\rho CO_2$ value, when exposed to a zero gas, is different from zero".*
Response:
This recommendation will be adopted in final revision.

8.    *L190: housing CO2/H2O accumulation.*

    Response
    See response to minor comment 1.

9.    *L209: housing CO2/H2O accumulation.*

    Response
    See response to minor comment 1.

10.    *L224: remove "calibration/", "span" is clear enough.*

    Response
    "Calibration" is a full process to construct the H2O and CO2 working equations in production process. "Span" is a user operation to adjust H2O/CO2 span coefficients. We clarified the difference in use of two terms in the manuscript. We will further check the clarity.

11.    L233-234: "that is smaller in magnitude by at least two orders" may be more concise.
    Response
    Yes, the word of "reasonably" ahead of "smaller" can be removed.

12.    L283: "microbial respiration" is more commonly used.
    Response
    The word of "microorganism" can be replaced with "microbial".

*13.*    Figure 2: For simplicity, I recommend using only absolute value of accuracy and relative accuracy.

    Response
    Accuracy is defined as a range. One positive value may mislead readers.

*14.    Table 2: These numbers are very detailed, and thus are somewhat a repeat of Figures 2 and 3. I recommend only show the temperature points in a coarse resolution, for example, -30, -20, -10, 0, 10, 20, 30, 40, 50 °C.*

    Response
    Yes, in final revision, this table can be simplified as reviewer suggested.

    Again, we really appreciate reviewer's positive comments in the significance of our study.

Sincerely,

Ning Zheng, Ph.D.
Application Scientist

---

## Author Response (AR1)

**INSTITUTE OF APPLIED ECOLOGY, CHINESE ACADEMY OF SCIENCES**

**72 Wenhua Road, Shenyang, Liaoning, 110016, China**

May 30, 2022

RE: Final author response to referees' comments on GI-2022-1

Dr. Salvatore Grimaldi
Dept. for Innovation in Bio., Agro-food, & For. Systems
University of Tuscia, Viterbo, Italy

Dear Dr. Grimaldi,

Thank you for allowing us to submit a revised version of our manuscript, "Accuracies of field $CO_2-H_2O$ measurements from open-path eddy-covariance systems: Assessment based on atmospheric physics and biological environment," for further consideration of publication in *Geoscientific Instrumentation, Methods and Data Systems (GI)*. Through the interactive discussion, two journal referees reviewed our manuscript and, as indicated by the metrics on the website https://gi.copernicus.org/preprints/gi-2022-1/#discussion, 612 international public reviewers viewed and/or downloaded the preprinting of this manuscript.

Both referees found this manuscript to be innovative in different aspects. Referee #1 commented, "The analysis methodology based on atmospheric physics and ecosystem background is truly innovative." Referee #2 commented, "This manuscript is innovative in trying to quantify the overall uncertainties in the measurements of $CO_2$ and $H_2O$ amounts by open-path eddy-covariance (OPEC) gas analysers due to their different sources." Also, as stated by Referee #1, "This manuscript along with Zhou et al. (2021) is the completion of systematic study on the overall accuracy of $CO_2/H_2O$ measurements from eddy-covariance systems." We particularly appreciated Referee #2's strong positive general comments and detailed constructive suggestions on how to substantially improve this manuscript. We appreciate the two anonymous journal referees for their feedback, and our appreciation to both was added to the acknowledgement.

The authors carefully discussed every comment from the two journal referees for this revision. Our discussions and proposed revisions in response to the corresponding comments are given below.

The line numbers used below correlate with manuscript GI-2022-1 instead of GI-2022-1R.

We appreciate your favorable consideration for publication of this manuscript in *GI*.

Sincerely,

Ning Zheng, Ph.D., Application Scientist
Eddy-Covariance Flux Instrumentation

**Response to Referees' comments on "Accuracies of field $CO_2$−$H_2O$ measurements from open-path eddy-covariance systems: Assessment based on atmospheric physics and biological environment"**

X.H. Zhou, B. Yang, T. Gao, Ning Zheng, Yanlei, Li, Fengyuan Yu, T. Awada, J.J. Zhu
https://doi.org/10.5194/gi-2022-1

**Response to Referee #1**
(https://doi.org/10/5194/gi-2022-1-RC1 and -RC2)

**General comments**

This study focuses on a practical subject needed to quantify the overall accuracy of $CO_2$/$H_2O$ measurements from open-path eddy-covariance (OPEC) systems. The OPEC is more popular than CPEC because of, for example, their lower power consumption and maintain demanding, in the flux community. While I am analyzing my data, I always concern the overall accuracy in $CO_2$ measurements from my infrared gas analyzers in OPEC systems, but the method how to estimate the overall accuracy were unavailable from published literature. Indeed, this manuscript along with Zhou et al (2021) is the completion of systematic study on the overall accuracy of $CO_2$/$H_2O$ measurements from EC systems. What is estimable is that the authors showed the accuracies of $CO_2$/$H_2O$ densities based on biologically meaningful data in the field and solid physical principles. Clearly, this study provided valuable results for scientists like me to reference. The analysis methodology based on atmospheric physics and ecosystem background is truly innovative and the equation development is logical in theory and practical in applications. Although the authors only used an old version of OPEC, equations (14) and (22) were easily used to calculate the accuracies of $CO_2$/$H_2O$ densities for other types of open-path analyzers, e.g., IRGASON and LI-COR 7500 series, having potentials in applications to analyzers for other gas species like $CH_4$ and $N_2O$ in the areas of geosciences. Additionally, the structure was well organized and the language were also very well written. Therefore, I would highly recommend this manuscript to be accepted for publication on Geoscientific Instrumentation, Methods and Data Systems after a minor revision.

> *Author response*
> We sincerely thank Referee #1 for his/her recommendation for publication of this manuscript, his/her comments on our approach based on atmospheric physics and the biological environment as innovative in estimating the overall accuracies of $CO_2$−$H_2O$ measurements from open-path eddy-covariance (OPEC) gas analyzers, and his/her awareness of this manuscript as the completion of systematic studies on the overall accuracy of $CO_2$−$H_2O$ measurements from eddy-covariance gas analyzers.

**Major comments**

I have two open-path analyzers, i.e., EC150 and LI-COR 7500. In practice, when I perform a zero calibration, I always found a positive zero drift about 10 μmol $mol^{-1}$ for LI-COR 7500 at ambient temperature, slightly higher in the unit of mg $CO_2$ $m^{-3}$ and much higher than the upper of the values in the manuscript, but a much smaller accuracy due to gain drift when tubing the $CO_2$ span gas of 500-μmol $mol^{-1}$ after a zeroing operation. I speculate that this was caused a non-negligible housing $CO_2$/$H_2O$ accumulation, although the chemicals in the internal cell needs no replacement of new ones, i.e., after a zero calibration the analyzer works well for months. This is

the same for $H_2O$ density. Therefore, in practice, I recommend the author give a short discussion of the possibility of field drift of zero and gain using the big data of analyzer-supplier, for example, that from EC150 in the lab of CSI, in the 6.3 section. These data may be helpful for providing suggestions for new users.

> *Author response*
> Yes, an individual infrared $CO_2-H_2O$ gas analyzer may behave differently in the field than in the lab due to various reasons. As Referee #2 commented, one of the more important strengths in this study is the use of sensor specifications in determining the range of uncertainties. Instead of using field data from individual infrared analyzers, our assessment is based on official manufacturer specifications for a given model of analyzers. The validity of data from individual analyzers in the field is unsure because no benchmark data are available to assess the field data, which is why we assess the overall accuracies for field $CO_2-H_2O$ data using sensor specifications based on atmospheric physics and the ecological background. This approach is recognized by Referee #2 as "a valid approach to visualize the uncertainty in a straightforward way."

**Minor comments**

**1.** Title: "$CO_2-H_2O$" (and in the text). I understand the authors wanted to identify both gas types using "−" from one of the two gas types using "/". In my opinion, however, "$CO_2/H_2O$" may be better, just the same as they are in the profile system. The same for other parts of the manuscript.

> *Author response*
> We also preferred "$CO_2/H_2O$," but that designation is isolating, technically meaning "$CO_2$ or $H_2O$." Alternatively, "$CO_2-H_2O$" is inclusive, meaning "$CO_2$ and $H_2O$." Based on our experience, we have learned that "$CO_2-H_2O$" is the editorial preference of Copernicus Publications (the publisher for *GI*) for this expression.

**2.** Line 24: For a background concentration of atmospheric $CO_2$?

> *Author response*
> In our interactive discussion, we misunderstood this comment. Referee #1 clarified this comment in http://doi.org.10.5194/gi-2022-1-RC2. Based on that clarification, we adopted this comment.

> *Author proposed revision*
> Line 24: "In an atmospheric $CO_2$ background" was replaced with "For a background concentration of atmospheric $CO_2$."

**3.** Lines 27-29: I recommend deleting "Under freezing conditions, an $H_2O$ span is both impractical and unnecessary, but the zero procedure becomes imperative to minimize $H_2O$ measurement uncertainty.", because there was some overlap of this sentence with the next one "In cold/dry conditions, the zero procedure for $H_2O$, along with $CO_2$, is an operational and efficient option to ensure and improve $H_2O$ accuracy".

> *Author response*
> Reminded by comment #2 in http://doi.org.10.5194/gi-2022-1-RC2, we reread the two sentences from lines 27 to 29. Both sentences read awkwardly, in large part due to the double mentions of the role of the zero procedure. We revised the two sentences.

> *Author proposed revision*

Lines 27−30, the two relevant sentences are revised to be:

"The $H_2O$ span procedure is impractical under freezing conditions and unnecessary under cold/dry conditions. However, the zero procedure for $H_2O$, along with $CO_2$, is imperative as an operational and efficient option under these conditions to minimize $H_2O$ measurement uncertainty."

**4.** Line 36: delete "fluctuations", for consistency with "3-D wind and sonic temperature".

*Author response*
This word can be removed. It may be redundant, although the word can reflect the nature of turbulence measurements.

*Author proposed revision*
Line 36:
The word "fluctuations" was removed.

**5.** Line 75: "$CO_2$/$H_2O$ molar mixing ratio" or "$CO_2$/$H_2O$ dry molar fraction" is better.

*Author response*
If "$CO_2$/$H_2O$ molar mixing ratio" is used here, then "$CO_2$/$H_2O$ molar or mass density" should be used as well. That wording becomes more cumbersome than clear. For simplicity inside parentheses, "$CO_2$/$H_2O$ mixing ratio vs. $CO_2$/$H_2O$ density" is sufficient.

**6.** Line 108: "in practice"?

*Author response*
"In practice" can be used to replace "in applications."

*Author proposed revision*
Line 108:
The word "applications" was replaced with "practice."

**7.** Line 170: Possibly, use "the analyzer often gradually reports that this zero $\rho CO_2$ value, when exposed to a zero gas, is different from zero".

*Author response*
This recommendation is adopted with slight revision.

*Author proposed revision*
Lines 169−170:
"However, during use of the analyzer in measurement environments that are different from calibration conditions, the analyzer often reports this zero $\rho_{CO2}$ value, while exposed to zero air, gradually away from zero and possibly beyond $\pm\Delta\rho_{CO_2}^{p}$, which is known as $CO_2$ zero drift."

**8.** Line 190: housing $CO_2$/$H_2O$ accumulation.

Line 209: housing $CO_2$/$H_2O$ accumulation.

*Author response*
Addressed in response to comment #1, above.

**9.** Line 224: remove "calibration/", "span" is clear enough.

*Author response*

In this manuscript, "calibration" involves the full process of constructing the $H_2O$ and $CO_2$ working equations in the production process; "span" means a user operation to adjust $H_2O/CO_2$ span coefficients. We clarified this distinction in the manuscript (see lines 160−63).

**10.** Lines 233-234: "that is smaller in magnitude by at least two orders" may be more concise.

*Author response*
Yes, the word "reasonably" ahead of "smaller" can be removed.

*Author proposed revision*
Line 233:
The word "reasonably" was removed.

**11.** Line 283: "microbial respiration" is more commonly used.

*Author response*
The word "microorganism" can be replaced with "microbial."

*Author proposed revision*

Line 283:

The word "microbial" replaced "microorganism."

**12.** Figure 2: For simplicity, I recommend using only absolute value of accuracy and relative accuracy.

*Author response*
Accuracy is defined as a range. The use of just one positive value to represent this range may mislead readers.

**13.** Table 2: These numbers are very detailed, and thus are somewhat a repeat of Figures 2 and 3. I recommend only show the temperature points in a coarse resolution, for example, -30, -20, -10, 0, 10, 20, 30, 40, 50 °C.

*Author response*
We adopted this comment.

*Author proposed revision*
Table 2:
The rows for the ambient temperatures of –22, –18, –12, –7, –2, 2, 7, 13, 18, 22, 28, and 32 ºC were removed.

**Response to Referee #2**
(https://doi.org/10/5194/gi-2022-1-RC3)

**General comments**
The manuscript is innovative in trying to quantify the overall uncertainties in the measurements of $CO_2$ and $H_2O$ amounts by Open Path Eddy Covariance (OPEC) gas analysers due to their different sources. The aim is pursued by means of a Campbell Scientific IRGASON, and then generalised. A simple model is developed to combine the different sources of errors, and the resulting uncertainties are plotted under different conditions of temperature and gas concentrations. In addition, some applications of the results are reported, together with some suggestions for the users during field calibration. This preprint follows the same approach of a

paper published last year by the same main Author relative to the Closed Path Eddy Covariance (CPEC) sensors.

The study has some points of strength and some points of weakness. Among the strenghts it is the fact that the study addresses relevant scientific questions within the scope of GI, using in part novel ideas, and using a proper language. One of the more important strenghts is that it uses the specs of the sensor to define its uncertainties, and it defines the uncertainty in terms of range: the worst case scenario is depicted for each source of error as the limits of the range, and then combined with the others. This is a valid approach to visualising the uncertainty in a straightforward way.

The more important weaknesses are in my opinion: #1. the poor link with the eddy covariance method, despite this is mentioned since the beginning; #2. the generalisation from the IRGASON/EC150 to all the Open Path sensors is not robust enough; #3. Applications and calibration suggestions are only partly relevant; #4. more references are needed, as the most cited are not peer-reviewed papers but sensors manual; #5. the discussion section is more dedicated to other things (recap of what done, applications), but the real discussion is limited; #6. conclusions should be strenghten as well.

I'll recall these points in the comments below when relevant.

*Author response*
We sincerely appreciate Referee #2's strong positive general comments above and his/her detailed constructive suggestions below. The suggestions guided us in addressing the related issues more clearly and eventually led to a substantially improved manuscript. We also appreciate his/her high expectation.

The specific comments from Referee #2 are reorganized into two sections:

A. Suggestions to fix the major issues pointed out in General Comments.
B. Technical and editorial comments to improve the manuscript expressions.

Section A is organized into six sub-sections corresponding to the six major issues. In each sub-section, all comments related to this major issue are addressed and the revisons proposed.

**Suggestions to fix the major issues pointed out in General Comments**

**#1.** The EC method includes a very long chain of steps from field measurements to calculation of the fluxes. In this chain, the specs of the sensor are in general considered less important in terms of final uncertainties. Also, the uncertainty are more relevant to the EC method in terms of fluxes (as the result of covariance between sonic and IRGA signals), not concentrations: this is clearly out of the scope of the manuscript, but should be mentioned and maybe discussed a bit. Also, an OPEC system is made of two main sensors: the IRGA and the sonic. But the latter is almost not considered in the study: this could be reconsidered, or at least the reasons for excluding this sensor should be given. A possible alternative could be to reconsider the link with EC: is that really needed? The study may focus on the Open path IRGA (so the EC150, not the IRGASON), just mentioning that it is often used for eddy covariance measurements, but clearly state since the beginning that the study will not focus on EC. (please note this will clearly impact the title as well).

*Author response*

The overall accuracies of $CO_2$ and $H_2O$ density measurements by infrared gas analyzers (IRGAs) are often questioned by field scientists in their data observations and requested by scientists in their purchasing processes. These accuracies are also needed for field auto $CO_2$ and $H_2O$ zero/span procedures to instrumentally adjust the drifts of IRGA in $CO_2$ and $H_2O$ zero and/or gain drifts (e.g., EasyFlux series, Campbell Scientific Inc., UT, US). In these procedures, the overall accuracies are used to judge the degree of the drifts. Although flux uncertainties are related to several factors (Richardson et al., 2012), the measurement accuracy in $CO_2/H_2O$ density is a fundamental question of interest to scientists (Fratini et al., 2014). Because the overall accuracy would have multiple applications, we limited our scope in this study to be within the model, estimation, and assessment of the overall accuracy of the $CO_2/H_2O$ density measured by IRGAs and used for $CO_2$ and $H_2O$ fluxes. As commented above, linking this study to the fluxes is clearly out of study scope. However, the IRGA is a major component in open-path eddy-covariance (OPEC) systems for $CO_2$ and $H_2O$ fluxes. Linking the context of this manuscript to related flux topics would make this manuscript more significant. To this end, in our revision, we:

a. Followed the comments below to link manuscript contexts to flux topics.
b. Checked the manuscript throughout for these linking opportunities.
c. Discussed how the overall accuracies in $CO_2$ and $H_2O$ density are analytically related to the flux errors in computations (see Appendix C and section 6.2).

**1.1.** Lines 13-14: as the IRGAs can be used for several scopes, if the link with the EC is maintained it is preferrable to mention "fluxes" (#1).

Line 14: As the focus of the manuscript seems to be only the IRGA, and not the sonic, this should be clearly stated (#1).

*Author response*
Yes, this manuscript focuses on the accuracies of field $CO_2-H_2O$ data measured by IRGAs used in OPEC systems. The first sentence specifies this link.

*Author proposed revision*
Lines 14–15:
"Ecosystem $CO_2-H_2O$ data measured by infrared gas analyzers in open-path eddy-covariance (OPEC) systems have numerous applications, such as estimations of $CO_2$ and $H_2O$ fluxes in the atmospheric boundary layer. To assess the applicability of these estimations, data uncertainties from infrared gas analyzer measurements are needed."

**1.2.** Lines 38-39: If the link with EC is maintained, it may be relevant in my opinion to mention that the exactness of EC measurements depends also on this, but not only. There is a long way to get to the fluxes after the field measurements, and each step sources uncertainty. This should be mentioned in my opinion, also referencing the papers dealing with other sources of uncertainties. (#1).

*Author response*
Our dilemma has been whether to link our manuscript context either more or less to flux computations. If more, the manuscript becomes lengthy, loses focus, and dims other applications of the overall accuracies. If less, it feels misleading because the infrared gas analyzers studied are, in fact, used in OPEC systems for $CO_2$ and $H_2O$ fluxes. In this revision, we adopted this comment to overcome weakness #1 while still linking the manuscript context to flux topics as much as possible.

*Author proposed revision*
Lines 38–42: "The degree of …………Hill et al., 2017)." was replaced with:

"Given that the measurement conditions, which are spatially homogenous in flux sources/sinks and temporally steady in turbulent flows without advection, satisfy the underlying theory for eddy-covariance flux techniques (Katul et al., 2004; Finnigan, 2008), the quality of each flux data primarily depends on the field measurement exactness of variables, such as $CO_2$, $H_2O$, $T_s$, and 3-D wind, at the sensor sensing scales (Foken et al., 2012; Richardson et al., 2012), although this quality can also be degraded by other biases if not fully corrected. In an OPEC system, other biases are commonly sourced from the tilt of vertical axis of the sonic anemometer away from the natural wind (Kaimal and Haugen, 1969), the spatial separation between the anemometer and the analyzer (Laubach and McNaughton, 1998), the line and/or volume averaging in measurements (Wyngaard, 1971; Andreas, 1981), the response delay of sensors to fluctuations in measured variables (Horst, 2000), the air density fluctuations due to heat and water fluxes (Webb et al., 1980), and the filtering in data processing (Rannik and Vesala, 1999). These biases are correctable through coordinate rotation corrections for the tilt (Tanner and Thurtell, 1960; Wilczak, 2001), covariance maximization for the separation (Moncrieff et al., 1997; Ibrom et al., 2007), low- and high-frequency corrections for the data filtering, line and/or volume averaging, and response delay (Moore, 1986; Lenschow et al., 1994; Massman, 2000; van Dijk, 2002), and WPL corrections for the air density fluctuations (Webb et al., 1980). Even though these corrections are thorough for corresponding biases, errors in the ultimate flux data still exist due to uncertainties related to measurement exactness of the sensor sensing scales (Fratini et al., 2014; Zhou et al., 2018). These uncertainties are not only unavoidable because of actual or apparent instrumental drifts due to the thermal sensitivity of sensor path lengths, long-term aging of sensor detection components, and unexpected factors in field operations (Fratini et al., 2014), but they are also not mathematically correctable because their sign and magnitude are unknown (Richardson et al., 2012). The overall measurement exactness related to these uncertainties would be a valuable addition to flux data analysis (Goulden et al., 1996; Anthoni et al., 2004).

Beyond flux computations, the data for individual variables from these field measurements have numerous applications. Knowledge of measurement exactness is also required for accurate assessment of data applicability (Csavina et al., 2017; Hill et al., 2017)."

**1.3.** Line 473: As you are considering eddy covariance applications, mentioning only $T_a$ is a bit reductive in my opinion (no user will buy the IRGASON to calculate $T_a$...). Also, $T_a$ is more related to sonic temperature $T_s$, and here you are only considering the IRGA uncertainties, not the sonic ones: $\Delta T_s$ is reported in the sensor's specs, right, as it is $\Delta\rho_{CO_2}$ and $\Delta\rho_{H_2O}$. $T_s$ is probably less sensitive (e.g., not cross-sensitivity present), but still can drift with temperature for example (see for example Mauder et al. 2007 https://doi.org/10.1007/s10546-006-9139-4). (#1, #3).

*Author response*
See our response to 1.2, above. A discussion of the applications of $\Delta\rho_{CO_2}$ and $\Delta\rho_{H_2O}$ for flux uncertainty was added to the manuscript as section 6.2 and section 6.3. The original sub-section numbers following 6.3 in section 6 were revised accordingly.

a. "A bit reductive in my opinion"

Computing $T_a$ from $T_s$ and a moisture-related variable (e.g., $\rho_{H2O}$), including atmospheric pressure ($P$), has been of interest to scientists since 1932 (Ishii, 1932; Barrett and Suomi, 1949; Kaimal and Businger, 1963; Schotanus et al., 1983; Kaimal and Gaynor, 1991; van Dijk, 2002; Swiatek, 2018; Zhou et al., 2022), although an exact equation for finding $T_a$ from $T_s$, $\rho_{H2O}$, and $P$ has not yet been reached (Zhou et al., 2022). Because this $T_a$ is a high-frequency signal that is insensitive to the solar contamination suffered by conventional $T$ sensor measurements inside a radiation shield (Lin et al., 2001; Blonquist and Bugbee, 2018) this air temperature is increasingly needed in the measurement space of flux for correction of the spectroscopic effect on $CO_2$ (Bogoev et al., 2015; Helbig et al., 2016; Wang et al., 2016) and on $CH_4$ (Burba et al., 2019). To correct a flux error due to bias in a gas concentration measurement, Fratini et al. (2014) requires the measurement of air temperature in the optical path of the infrared analyzer. $T_a$ is the best proxy of this temperature, although Fratini et al. (2014) might not use this high-frequency $T_a$ then.

b.  "No user will buy the IRGASON to calculate $T_a$"
All IRGASON sensors internally calculate this $T_a$ to correct the spectroscopic effect on $CO_2$ (Bogoev et al., 2015) using an approximation equation of $T_a$ (Swiatek, 2018). The option of a corrected $CO_2$ was incorporated into EasyFlux-DL-CR6OP by Dr. Zhou, who is the first author of this manuscript.

c.  "$T_a$ is more related to sonic temperature $T_s$"
In dry air, $T_a$ is equal to $T_s$; in this case, the bias of $T_a$ totally depends on the accuracy of $T_s$, although this accuracy has not been available yet (Zhou et al., 2022). In the case of moist air, if $T_s$ is accurately measured, the bias of $T_a$ depends on the errors in $\rho_{H2O}$ and $P$. Under humid conditions at a high air temperature (e.g., 35 ºC), the bias can reach a couple of Kelvins if $\rho_{H2O}$ and $P$ have larger errors.

d.  "$T_s$ is probably less sensitive (e.g., not cross-sensitivity present), but still can drift with temperature"
The ability to accurately measure $T_s$ with an overall accuracy of <0.5 ºC is a challenge. So far, no methodology is available to quantify the overall accuracy on a solid base (personal communications in 2022 with Larry Jacobsen [CSAT authority] and Richard McKay [Product Manager with Gill sonic anemometer for the last 10 years and now with Campbell Scientific Ltd.]). Regardless, this topic is out of the scope of this manuscript.

*Author proposed revision*
Line 667:
After Line 667, we inserted Appendix C: Appendix C: The relationship of measured to true covariance to of vertical wind speed with $CO_2$, $H_2O$, and air temperature

Line 393:
After Line 393, we inserted section 6.1
6.1 Partial effects of $\Delta\rho_{CO2}$ and $\Delta\rho_{H2O}$ on uncertainty of hourly $CO_2$/$H_2O$ flux

**#2.** It should be shown that the specs used are all necessary and sufficient, and provide guidance to the reader in case some of them are missing on a different sensor specs (better if also considering additional specs that may be found). In some occasions the authors refer to "OPEC systems" while dealing with the specs of the IRGASON - which may be not the case.

*Author response*
We clarified this issue in the revision in response to the following comments.

**2.1** Lines 20-22: please specify that it refers to IRGASON/EC150 only: it seems to be a generic statement for OPEC systems (#2).

*Author response*
IRGASON and EC150 infrared $CO_2-H_2O$ analyzers have the same specifications. This study uses EC150 as an example.

*Author proposed revision*
Lines 20−22:
"Based on atmospheric physics and the biological environment, for EC150 infrared $CO_2-H_2O$ analyzers, these equations are used to evaluate $CO_2$ accuracy ($\pm1.21$ mg$CO_2$ m$^{-3}$, relatively $\pm0.19\%$) and $H_2O$ accuracy ($\pm0.10$ g$H_2O$ m$^{-3}$, relatively $\pm0.18\%$ in saturated air at 35 °C and 101.325 kPa)."

**2.2.** Line 91 (tab1): if you want to make it more general, you should specify whether or not this list is sufficient and necessary: what if a different sensor is missing some info? And what if there are more sources of uncertainties listed for a sensor? This should be reported (here and/or in Appendix A) (#2)

*Author response*
Your comment 2.11 addresses this issue. See our response to comment 2.11 and corresponding author proposed revision.

**2.3.** Line 148 (eq 5): while I think this equation is general, as it is proposed in a sensor's manual (i.e., not peer-reviewed) in my opinion it is not very robust to include it in a scientific paper without an indepth analysis. As sources are present in LICOR's manual, I would prefer to see it derived from there. Otherwise, in addition to not being scientifically robust, this may also be felt as ambiguous in terms of at which sensors can be generalised: its applicability at sensors other than the one the manual is referring to should be shown (IRGASON and beyond). In alternative, if some other publications exist that already "validated" LICOR's equation, they could be referenced here. Then, the parameters in the equation can guide the reader in understanding its applicability, e.g., all the IRGAs using a 5th order polynomial for $CO_2$, etc. (#2, #4).

*Author response*
Globally, all fast-response infrared $CO_2-H_2O$ analyzers currently used for field OPEC and CPEC systems are made by two manufacturers: LI−COR Biosciences and Campbell Scientific. Either is trustworthy in the flux community. Their manuals and programs for release as documents and products are rigorously, but internally, reviewed within professional standards; they are not necessarily externally reviewed in the same way as journals. Particularly, all proprietary techniques related to most advanced technologies are reviewed by a small group of internal experts instead of external referees. However, these technologies in related areas are technically and commercially valuable. If journal reviews were the only data sources deemed valid, then all current $CO_2$ and $H_2O$ flux data from field OPEC and CPEC systems would be invalid because many technical details inside infrared $CO_2-H_2O$ analyzers are not published by either LI−COR Biosciences or Campbell Scientific. We believe the manuals from industry-trusted manufacturers have equal

credibility to journal publications. In fact, our first author is affiliated with Campbell Scientific; therefore, we strongly feel that citing LI−COR Biosciences documents is credible.

a.  Derivation of Models (5) and (17)

Both models have been in the different versions of manuals for the LI–7500 series over last 20 years (LI−COR Biosciences 2001, 2021a, 2021b). The derivations of both models are solid in sciences and understandable. For detailed derivations, see Eqs. 2-1 to 2-17 in the Theory and operation section on pages 2-1 to 2-12 in LI−COR Biosciences (2001). In this revision, the derivations of both models are referred to in the Theory and operation section in the literature over the past 20 years.

b.  Working model inside EC150, IRGASON, and EC155

The working models inside EC150, IRGASON, and EC155 infrared analyzers were not published in public domains. We are not allowed to disclose the related information, although the field auto and manual $CO_2/H_2O$ zero/span procedures for the three models are implemented by the first author of this manuscript and are used globally. Following LI−7500, these three models of infrared analyzers were developed after 2005. There was no reason not to use Models (5) and (17) for EC150/IRGASON consistency with LI−7500 data in precision, zero drift, and gain drift uncertainties. Of the three models, EC150 was the first released, in 2011. At that time, it would have been unacceptable for EC150 data uncertainties to be inconsistent with L−I7500 uncertainties.

c.  Robust of Models (5) and (17)

Figure R1 below shows two Calibration Certificates for LI−7500. One was issued in 2002 and the other in 2021. The parameters in the two certificates indicate that Models (5) and (17) have been consistently used for 20 years. Both models are robust. We are continuously looking for ways to improve the infrared $CO_2/H_2O$ analyzers and would be glad to know of models that work better than Models (5) and (17).

**LI-7500 CO₂/H₂O Analyzer**

Calibration Certificate

Serial Number 75H-0150

Date: 23 Apr 2002          Technician ________

**CO₂ Calibration Values**
A = 1.3184E2
B = 1.59059E4
C = 2.21287E7
D = -5.26988E9
E = 6.28028E11
XS = 0.0075
Z = 0.0015

**H₂O Calibration Values**      **Pressure Calibration\***
A = 5.15909E3                   A0 = 10.697
B = 3.44417E6                   A1 = 26.036
C = 2.35922E7
XS = -0.0038                    \* Ver 3.0.1 and above
Z = 0.0202

**Zero/Span set on 24 Apr 2002**
CO2 Zero = 0.8900
CO2 Span = 0.9980 (at 999 ppm)
H2O Zero = 0.5890
H2O Span = 0.9840 (at 15 C)

**LI-7500 CO₂/H₂O Analyzer**

Calibration Certificate

Serial Number 75D-4427

Date: 30 Mar 2021          Technician ________
      Code:34829                       Jerry F.

**CO₂ Calibration Values**   **H₂O Calibration Values**
A = 1.16284E2                A = 5.81603E3
B = 9.99702E3                B = 6.55766E6
C = 1.75717E7                C = -7.08744E8
D = -4.12443E9               XS = 0.0001
E = 4.16348E11               Z = -7.40000E-3
XS = -0.0011
Z = 6.50000E-3              ⁱSD1 = 0.0204
⁺SD1 = 0.0147               SD2 = -0.1081
SD2 = -0.0414              SD3 = 2.1820
SD3 = 1.9450
                           \* **Signal Strength**
\* Ver 6.5 and above           B = 1.6
⁺ Ver 7.6 and above           C = 3.2

**Zero/Span set on 31 Mar 2021**
CO2 Zero = 1.1271
CO2 Span = 1.0010 (at 752 ppm)    **Pressure Calibration**
CO2 Span2 = 0.0000               A0 = 10.416
H2O Zero = 0.9815                A1 = 26.036
H2O Span = 1.0051 (at 12 C)
H2O Span2 = 0.0000               <serial number>

Figure R1. Two Calibration Certificates were issued in 2002 and 2021 for two LI–7500 CO₂/H₂O analyzers. For $CO_2$; see model (5); A is $a_{c1}$; B, $a_{c2}$; C, $a_{c3}$; D, $a_{c4}$; E, $a_{c5}$; XS, $S_w$, and Z, $Z_c$. For $H_2O$; see model (17); A is $a_{w1}$; B, $a_{w2}$; C, $a_{w3}$; XS, $S_c$, and Z, $Z_w$.

*Author proposed revision*
Lines 146−147:
"According to LI−COR Biosciences (2021b)" was revised to be:

 "From the derivations in the Theory and operation section in LI−COR Biosciences (2001, 2021a, 2021b)."

**2.4.** Line 195: For EC150, not for OPEC in general (#2).

*Author proposed revision*
Line 195:

Between "of" and "OPEC," we inserted the words "EC150 infrared analyzers used for."

**2.5.** Lines 203-204: These values are again for the EC150 only. (#2) Please also note that "rh" may be misunderstood for relative humidity.

*Author response*
Revised in the same way as above. The subscript "rh" written in lowercase should be fine. To avoid the risk of introducing errors with revision, we decided to keep this subscript in its current form.

*Author proposed revision*
Line 203:

Between "of" and "OPEC," we inserted the words "EC150 infrared analyzers used for."

**2.6.** Line 211: Again, if it has to be generic, sentences from LICOR manuals shouldn't be used

alone, as 1. they are not peer-reviewed and 2. things could be different for different models (#2, #4).

> *Author response*
> Richardson et al. (2012) and Fratini et al. (2014) also support this statement. Both references were added.

> *Author proposed revision*
> Line 211:

> Between "(" and "LI−COR," we inserted "Richardson et al., 2012; Fratini et al., 2014;"

**2.7.** Line 271: Again, it should be noted that these specs, and then the results below, are relative to the EC150, including the operational range: are you sure you can generalise to all the OPEC systems? (e.g., LICOR LI7500DS has a range of -25 to 50°C) (#2)

> *Author response*
> Yes, both LI-COR Biosciences and Campbell Scientific used the same specification term, although the specification values in some terms are slightly different (e.g., measurement range for $CO_2$). Because some authors of this manuscript have an affiliation with Campbell Scientific, it seems appropriate to give more credit to LI-COR and less generalization about LI-COR products.

> *Author proposed revision*
> Line 270:

> Between "which" and "OPEC," we inserted the words "EC150 infrared analyzers used for."

**2.8.** Line 323: see the comments in section 4, in particular at line 148 (#2, #4).

> *Author response*
> See response to comment 2.3.

> *Author proposed revision*
> Lines 322−323:

> The words "by (LI−COR Biosciences 2001b)" was revised to be "also by the derivations in the Theory and operation section in LI−COR Biosciences (2001, 2021a, 2021b)"

**2.9.** Line 360: Here could be a good candidate to mention the generalisation point (#2).

> *Author response*
> For the generalization, it is better to be specific in the last sentence in this paragraph and then mention the application of Eq. (12) for other models of infrared analyzers. We sincerely appreciate Referee's thinking.

> *Author proposed revision*
> Lines 359−360:
> The last sentence in these two lines was replaced with:
> "Using this equation and the specification values as in Table 1 for EC150 infrared analyzers, the accuracy of field $H_2O$ measurements can be evaluated as a range for OPEC systems with such anlyzers. For an OPEC system with another model of open-path infrared anlyzer, such as the LI−7500 series (LI−COR Biosciences, NE, USA) or IRGASON (Campbell Scientific Inc., UT, USA), its corresponding specification values are used."

**2.10.** Line 367: please consider rephrasing: this is a plausibility range, and the calibration range

of both EC150 and LI7500. It is likely the same for most analysers, but again I think it can't be generalised in absolute terms. (#2).

*Author response*
Yes, any manufacturer should calibrate their gas analyzers for $H_2O$ measurement for $H_2O$ density around or below the highest dew point temperature of 35 ºC ever recorded under natural conditions on the earth (National Weather Service, 2021). The top limit of 44 $gH_2O$ $m^{-3}$ is equivalent to 37 °C at dew point in EC150 production conditions. This dew point is 2 °C higher than the highest one under natural conditions. Accordingly, this sentence can be rephrased.

*Author proposed revision*
Line 367:

The first sentence was replaced with the two sentences below:
"The EC150 analyzers were calibrated for $H_2O$ density from 0 to 44 $gH_2O$ $m^{-3}$ due to the reason addressed in Sect. 2. The highest limit of measurement range for $H_2O$ density by other models of analyzers also should be near 44 $gH_2O$ $m^{-3}$."

**2.11.** Line 407: Here it is a good candidate to discuss the fact that any other uncertainties are lacking in the model (#2).

*Author response*
Over last 20 years, the measurement uncertainties of infrared $CO_2$−$H_2O$ analyzers for OPEC systems have been defined consistently using precision, cross-sensitivities, zero drifts, and gain drifts for both $H_2O$ and $CO_2$ (LI−COR Biosciences, 2001, 2021a, 2021b; Campbell Scientific Inc., 2021). With the development of optical technologies, more measurement uncertainties are not expected to be added to analyzer specifications, and current measurement uncertainties could be removed from the current specification list. However, this removal would not happen in the near future unless low-cost laser heads for such measurements become available for field applications in performance stability and power saving. As you suggested, we added more discussion in the Discussion section.

*Author proposed revision*
Line 413:

After Line 413, the following paragraph was added.

"Additionally, included in the accuracy model, the four types of measurement uncertainties (zero drift, gain drift, sensitivity-to-$CO_2$/$H_2O$, and precision variability) to specify the performance of infrared $CO_2$−$H_2O$ analyzers for OPEC systems have been consistently used over last 20 years (LI−COR Biosciences, 2001, 2021a, 2021b; Campbell Scientific Inc., 2021). With the advancement of optical technologies, the measurement uncertainties for analyzer specifications are not expected to increase; rather, some current measurement uncertainties could be removed from the current specification list, even if not in the near future. If removed, the corresponding terms in the model could be easily removed, at which point, this model would be adapted to the new set of specifications for infrared $CO_2$−$H_2O$ analyzers."

**#3.** The suggestion of calibrating on an "average" temperature (Ta) to basically avoid to be in the worst case scenario (Ta and Tc at the extremes) is not robust as this is what normally happens, also because the range of Ta between two calibrations can be very large. In addition, it is based

on the assumption of linear relationship between the difference Ta-Tc and the drift magnitude, which derives from a simplification not so deeply documented. Also the applications proposed are not very impactful: if the EC method is kept (see #1) many more interesting applications could be thought of (but again, probably out of the scope). Even without that, I would use this idea of "applications" to improve the point above: the first and more relevant application should be "how to calculate the uncertainty for a generic IRGA".

*Author response*
We adopted this idea to revise our manuscript. We offer the following discussion in response to several points in this comment:

a. "on an "average" temperature"
Our first author, Dr. Zhou, was trained to use LI−7500 series for eight OPEC stations at University of Nebraska–Lincoln for over the course of 10 years. Now, Dr. Zhou advises global Campbell Scientific users daily. It is a popular recommendation to zero and/or span infrared $CO_2$−$H_2O$ analyzers around the average air temperature for analyzer operations rather than perform the zero and/or span procedure at extreme conditions or lab conditions. Fratini et al. (2014) recommended the zero and/or span procedure "at the temperature that minimizes the temperature departure, on average, during the period of interest." This recommendation is better in wording, but it is hard for users to digest. We adopt this wording in our revision.

b. "on the assumption of linear relationship"
This assumption is not the first principle, but it is a way to describe the behavior of zero and gain drifts with ambient air temperature ($T_a$). As seen in Figure 4b, the $H_2O$ accuracy as influenced by the linear trend of zero and gain drifts with $T_a$ is more shadowed by the exponential trend of saturated $H_2O$ density with $T_a$. In Figure 2a, the $CO_2$ accuracy as influenced by the linear trend of zero and gain drifts with $T_a$ is dominated by the $CO_2$ density of the ecosystem background with $T_a$. The merits of our methodology are the uses of atmospheric physics and biological environment principles for field data.

c. "not very impactful"
According to the metric of interactive discussion (https://gi.copernicus.org/preprints/gi-2022-1/#discussion), so far, 612 viewers have viewed and/or downloaded this manuscript. This high number of viewers indicates that this manuscript is already impactful.

Our finding, based on atmospheric physics, that $H_2O$ gain drift insignificantly contributes the accuracy range for $H_2O$ measurement at low temperatures (e.g., <5 ºC) and/or under dry conditions, has been successfully applied to the field auto adjustment of $H_2O$ zero coefficient instrumentally for IRGASON and EC150 + CSAT3A OPEC systems. This technique has been used in remote areas in Tibet and Qinghai in China and Logan in the US, although it has not been published yet. When published, this technique will benefit a large number of users who operate OPEC system in cold and/or dry regions.

Our finding, based on atmospheric physics, that the trend of $H_2O$ relative accuracy can answer the question of the $H_2O$ relative accuracy of infrared analyzers. This question is often asked by scientists worldwide for their selection in purchase processes.

**3.1.** Line 335: I think an important point should be taken into consideration here: Tc must be significantly lower than Ta at the moment of field calibration for $H_2O$ span to avoid condensation

(3-5°C, as reported in the LICOR manual) (#3).

*Author response*
$T_c$ is the environmental temperature at which an infrared analyzer is zeroed or spanned. What Referee #2 talked about is the dew point temperature set for an LI−610 Dew Point Generator to perform an $H_2O$ span for an infrared analyzer (LI−COR Biosciences, 2004). A common recommendation is to set the dew point temperature for LI−610 at 5 ºC, at least, below $T_c$. The correct use of LI-610 also needs the consideration of the difference between the pressures inside and outside the compressor. In our opinion, this manuscript does not have room to train users on such details (LI−COR Biosciences, 2004).

*Author proposed revision*
N/A

**3.2.** Line 439: Applications should probably go in a dedicated section. However, the first two suggested applications of relative accuracy is just a way to define sensors' specs, then in my opinion they should be just mentioned, not reported in such details. (#3, #5).

*Author response*
The $H_2O$ relative accuracy of infrared analyzers has not been clearly addressed before this study, even though many users do care about the relative accuracy in their instrument selection processes. We are often asked this question by scientists, which is the motivation for us to write this paragraph. The $CO_2$ relative accuracy of infrared analyzers is not an issue; however, if we only address the $H_2O$ relative accuracy, the question of why the $CO_2$ relative accuracy is not discussed will ultimately arise. We prefer to detail the use of relative accuracies here.

*Author proposed revision*
N/A

**3.3.** Lines 512-513: it is also true that the widest possible range would apply only if calibrating in extreme conditions far from the daily average (#3, #5).

*Author response*
We do not suggest that users zero and/or span the infrared analyzers under extreme conditions (e.g., $T_c$ below –15 ºC or above 30 ºC), although it is possible for analyzers to run in extreme conditions. Therefore, our discussion gives the $H_2O$ limit $T_c$ within a range of 5 to 20 ºC (see Figure 5) for analyzer use in a range of $T_a$ from –30 to 50 ºC (see Figure 5), over which the EC150/IRGASON infrared analyzers can run.

*Author proposed revision*
N/A

**3.4.** Line 524: See comment at line 335: the span procedure with a dew point generator MUST be performed at a much lower temperature than ambient to avoid condensation in the tubes and a bad calibration. This should be mentioned (also, does it worth it to "risk" to perform a bad calibration for correcting this? This is probably out of topic for the manuscript, but a short note could be beneficial to the reader). This risk is also reported in the LICOR manual (a note on "Checking the span" section) (#3, #5).

Line 568: again, the worst case scenario is also less likely... (#3, #5)

*Author response*

See our author response to comment 3.1.

*Author proposed revision*
N/A

3.5. Lines 617-618: this suggestion is mostly for sensors producers (#3, #6)

*Author response*
Not necessarily, because producers and users should share an understanding of instrument specifications. Producers need the knowledge gained from research community feedback, including demands and desires. To be clear, the developers and manufacturers of infrared $CO_2-H_2O$ analyzers, including sonic anemometers that are used for OPEC systems, are scientists, and users who use the OPEC systems are mostly for scientific projects.

*Author proposed revision*
N/A

**#4.** I think the paper from Fratini et al. 2014 (Fratini, G., McDermitt, D. K., and Papale, D.: Eddy-covariance flux errors due to biases in gas concentration measurements: origins, quantification and correction, Biogeosciences, 11, 1037–1051, https://doi.org/10.5194/bg-11-1037-2014, 2014) should definitely be included in the discussion, as it develops a correction of EC fluxes based on the drift of the IRGA as measured during field calibrations. It is different from what presented in this preprint, strongly bounded to the EC method; however it cannot be omitted in a paper dealing with the drift of the IRGA. Some publications on the theory beyond the IRGA working principles, from which the working equations presented are derived, should also be included, in addition (and in support) to the ones in the LICOR manual (not peer-reviewed). Also, publications dealing with uncertainties in EC method should be present, if the link with EC (#1) is maintained.

*Author response*
We regret missing this publication for our discussion in our study. Following Referee #2's advice, we thoroughly reviewed Fratini et al. (2014) and Richardson et al. (2012). Both studies supported our understanding of drifts of infrared analyzers with ambient air temperature for measurements. Both were valuable additions to our discussion in this manuscript and were cited several times in our revision.

**4.1**. Line 55-56: This is likely the case. However, several publications exist trying to quantify the uncertainties of EC measurements: this should be mentioned and the difference between this study discussed (here we are dealing with the exactness of the measurements of the IRGA only, there they are considering the EC flux. In some work the instrumentation uncertainty is included in the overall uncertainty). (#4).

*Author response*
The topic of flux uncertainties is broad. Additional information may make readers feel the introduction is too lengthy. To follow this comment, we added one paragraph to mention related studies.

*Author proposed revision*
Line 52:
After Line 52, we inserted the following:
"As comprehensively reviewed by Richardson et al. (2012), numerous previous studies including Goulden et al. (1996), Lee et al. (1999), Anthoni et al. (1999, 2004), and Flanagan

and Johonson (2005) have quantified various sources of flux measurement uncertainties and have attempted to attach confidence intervals to the annual sums of net ecosystem exchange. These sources include measurement methods (e.g., sensor separation and site homogeneity (Munger et al., 2012)), data processing algorithms (e.g., data filtering (Rannik and Vesala, 1999) and data gap filling (Richardson and Hollinger, 2007)), measurement conditions (e.g., advection (Finnigan, 2008), energy closure (Foken, 2008), and sensor body heating effect (Burba et al., 2008). Instead of quantifying the flux uncertainties, Foken et al. (2004, 2012) assessed the flux data into nine grades (1 to 9) based on steady state, turbulence conditions, and wind direction in the sonic anemometer coordinate system. The lower the grade, the less uncertainty; the higher the grade, the more uncertainty. The grade matrix for flux data uncertainty (e.g., quality) has been adopted by AmeriFlux (2018). In other aspects to correct the measurement bias from infrared analyzers, Burba et al. (2008) developed the correction for a sensor body heating effect on $CO_2$ and $H_2O$ fluxes, wheras Fratini et al. (2014) developed a method for correcting the raw high-frequency $CO_2$ and $H_2O$ data using the zero and span coeffcients of an infrared gas analyzer that were acquired from the same conditions, but at the begnning and ending of a time period. The corrected data were used to re-estimate the fluxes. To the best of our knowledge, no study has addressed the uncorrectable, although preventable to some degree, overall uncertainties in $CO_2$ and $H_2O$ data from infrared anlyzers, even though both overall uncertainties are fundamental for data analysis in applications (Richardson et al., 2012)."

**4.2**. Line 173: ref needed. Indeed, other reasons for the drifts are: dirt contamination, ageing of the IRGA's components, errors in pressure correction (absorptances are normalised to P), and errors in field calibration. If only Ta has to be considered, all of the other sources should be assumed to be zero - which should be at least mentioned. See also Fratini et al. 2014 (#4)

*Author response*

Dirt contaminations in the field and field zero/span errors from users cannot be considered by any manufacturer while specifying the drifts. No open-path analyzers for eddy-covariance measurements can perform reasonably well under heavy pollution, fog, rain, snow, ice, and/or sandstorm conditions. The measurement uncertainties from these events are unpredictable, which is out of the scope of the manufacturer specifications. The drifts as influenced by the aging of infrared analyzers within some age range (e.g., 10 years), a little $CO_2/H_2O$ accumulation inside a light house under normal maintenance, and thermal expansion/contraction of instrument components are in the scope of manufacturer specifications. Fratini et al. (2014) had the most excellent analysis for the dependence of drift on ambient air temperature, which we included to revise Lines 173 to 176.

*Author proposed revision*

Lines 171–176:

We revised Lines 171 to 176 to connect the context to the following inserted paragraph:

"Firstly, the dependency of analyzer $CO_2$ zero drift on ambient air temperature arises due to a thermal expansion/contraction of analyzer components that slightly changes the analyzer geometry (Fratini et al., 2014). This change in geometry can deviate the light path length for measurement a little away from the length under manufacturer calibration, contributing to the drift. Additionally, inside an analyzer, the performance of the light source and absorption detector for measurement, as well as the electronic components for measurement control,

can vary slightly with temperature. In production, an analyzer is calibrated to compensate for the ensemble of such dependencies as assessed in a calibration chamber. The compensation algorithms are implemented in the analyzer operating system, which is kept as proprietary by the analyzer manufacturer. However, the response of an analyzer to a temperature varies as conditions change over time (Fratini et al., 2014). Therefore, manufacturers typically specify an expected range of typical or maximal drift per ºC (see Table 1). Secondly, the housing $CO_2-H_2O$ accumulation is caused by unavoidable little leaks in the light housing of an infrared analyzer. The housing is technically sealed to keep housing air close to zero air by implementing scrubber chemicals into the housing to absorb any $CO_2$ and $H_2O$ that may sneak into the housing through an exchange with any ambient air (LI−COR Biosciences, 2021b). Over time, the scrubber chemicals may be saturated by $CO_2$ and/or $H_2O$ or lose their active absorbing effectiveness, which can result in housing $CO_2-H_2O$ accumulations. Thirdly, as optical components, the light source may gradually become dim, and the absorption detector may gradually become less sensitive. The accumulation and aging develop slowly in the relative long term (e.g., months or longer), whereas the dependencies of drift on ambient air temperature occur as soon as an analyzer is deployed in the field (Richardson et al., 2012). Apparently, the drift with ambient air temperature is a major concern if an analyzer is maintained as scheduled (Campbell Scientific Inc., 2021b)."

**4.3.** Line 347-348: I understand the logic behind this, however some evidence should be provided that this is the case, against the case, for example, that the lowest cross sensitivity to $CO_2$ unc. is with $CO_2$-free air - or vice versa in the $CO_2$ case, that the lowest cross sensitivity to $H_2O$ is with dry air and not with a "standard" water vapour concentration (somehow related to #4, and to #5 as it could be matter of discussion).

*Author response*
We appreciate Referee #2's understanding of the rationale behind Eqs. (13) and (20). It is our innovation to use atmospheric physics and the biological environment as a base to assess the overall accuracies of $CO_2$ and $H_2O$ data from infrared analyzers deployed in ecosystems for flux measurement. In ecosystems, the minimum $H_2O$ density is close to zero (e.g., cold and/or dry), and the minimum $CO_2$ density is close to 760 mgCO$_2$ m$^{-3}$, which is the atmospheric background $CO_2$ density. This represents a conservative way to estimate sensitivity-to-$CO_2$/$H_2O$ uncertainty. Additionally, the magnitude of this term is small (see Table 1).

*Author proposed revision*
N/A

**4.4.** Lines 500-502: this is also a good place to discuss Fratini paper, which is based on field calibration data (#4, #5).

*Author response*
We added an extensive discussion about Fratini et al. (2014) in response to comments 4.1 and 4.2. Here, their work should be mentioned.

*Author proposed revision*
Line 501:

Before the word "this," we inserted:

"Fratini et al. (2014) provided a technique implemented into the EddyPro program to correct

the drift bias from a raw time series of $CO_2$ and $H_2O$ data through post-processing."

**4.5.** Lines 571-572: Ref. needed (#4, #5)

*Author response*
Fratini et al. (2014) supports this statement.

*Author proposed revision*
Line 571:

After the word "environment," we inserted the citation "Fratini et al. (2014)."

**#5.** if some more applications are described, I would opt for a separate section of the paper, and for enriching the discussion section with discussion, citing different papers and going more details on what the results suggest.

*Author response*
We appreciate this idea from Referee #2. Separation of Discussion into Application and Discussion will improve the structure of manuscript.

*Author proposed revision*
The discussion section was separated section 6 into Application and Discussion. The section number for Conclusion was revised from 7 to 8.

**5.1.** Line 393: In general, this section is often more a (even useful) recap of what has been done and an application study (also useful) than a discussion of what done, also against other studies (#5).

*Author response*
See author response to comment #5.

**5.2.** Line 424-426: This is quite a critical point: I agree that such a relationship is not modeled yet, and that considering the maximum range is what the users may want and understand; however, I think assuming a linear scaling of the uncertainty and including it in the computation is a bit risky. At least, how far from the actual uncertainty is that one? Some more discussion needed, also checking Fratini et al. 2014 (#5, #4).

*Author response*
See Author response b. to comment #3; the discussion can be enhanced here. For comparison of this study to Fratini et al. (2014), due to the distinction between the two studies (see Author proposed revision in response to comment #4.1 and Author response b. to comment #4), the authors decided not to compare.

*Author proposed revision*
Line 427:
After this line, we added:
"In fact, the $H_2O$ accuracy as influenced by the linear trend of zero and gain drifts with the difference between $T_a$ and $T_c$ is more shadowed by the exponential trend of saturated $H_2O$ density with $T_a$ (Fig. 4b). Similarly, the $CO_2$ accuracy as influenced by the linear trend of zero and gain drifts with this difference is dominated by the $CO_2$ density of the ecosystem background with $T_a$, particularly in the low temperature range. Ultimately, the assumed linear trend does not play a dominant role in the accuracy trends of $CO_2$ and $H_2O$, which

shows the merits of our methodology in the uses of atmospheric physics and biological environment principles for the field data."

**5.3.** Lines 490-491: this is correct and probably the most relevant part of this section. However, this is strongly related to the drift uncertainty that is re-scalded to the difference Tc-Ta, and this is said above to be not exact (@425), and is also based on the assumptions that only the Ta dependency impacts the drifts. As no other demonstrations are given, this is also not very robust in my understanding (#5).

*Author response*
According to the definition of analyzer drifts by unit (see the first paragraph in section 3.3.1 of Fratini et al. (2014) and Table 1 in this manuscript), it is robust. As both Referee #2 and authors agree, it is not. This term would not be considered an uncertainty if any mathematical description could describe this trend exactly based on the first principles of physics. We believe, based on the best of our knowledge, we adequately addressed this issue in discussion and in Author proposed revision in response to comment 5.2.

**#6.** With the improvements above, the conclusion section will become more robust.

**6.1** Line 577: Some of the comments above clearly applies to this section as well (all the points, #6).

*Author response*
See author response to comment #5. Based on revisions suggested by both referees, the conclusion was revised.

*Author proposed revision*
See conclusion section.

**Technical and editorial comments to improve the manuscript expressions**

**1.** Line 25: "narrow the accuracy" is improving it? Please consider rephrasing.

*Author proposed revision*
Line 25:

"$CO_2$ accuracy" was revised to be "$CO_2$ accuracy range."

**2.** Line 48-49: Such an example at the beginning of the intro is misplaced in my opinion. Also, Ts accuracy is not under discussion. I would keep it for later.

*Author response*
Following to other comments, the introduction section was revised.

*Author proposed revision*
N/A

**3.** Line 64: Also CPEC.

*Author response*
"CPEC" was added, and the context related to this abbreviation was also revised.

*Author proposed revision*
Line 64:

" … in OPEC systems" was revised to be " … in OPEC and CPEC systems, where CPEC is an

acronym for closed-path eddy-covariance."

Line 71:

"For closed-path eddy-covariance (CPEC) systems" was revised to be "For CPEC systems."

**4.** Line 72: To be more clear: density measurements. It is probably worth it to state that in the manuscript $CO_2/H_2O$ measurements always refer to density, not flux (as EC technique estimates $CO_2/H_2O$ fluxes).

*Author response*
Yes, the wording needed precision. We clarified the same expression throughout the manuscript.

*Author proposed revision*
Line 72:

"$CO_2/H_2O$ measurements" was specified to be "$CO_2/H_2O$ mixing ratio measurements."

Line 73:

"$CO_2$ and $H_2O$ data" was specified to be "$CO_2$ and $H_2O$ mixing ratio data."

**5.** Line 83: a (typo)

*Author proposed revision*
Line 83:

The word "an" was revised to be "a."

**6.** Line 88: amount.

*Author proposed revision*
Line 88:

We inserted "amounts" between "$H_2O$" and "(Fig. 1)."

**7.** Line 128: This is mathematically shown in Appendix A: Please clearly refer to it (not only later).

*Author response*
A reader would benefit from an early reference to the mathematical derivations in Appendix A.

*Author proposed revision*
Line 125:
We inserted "as mathematically shown in Appendix A" between "analyzers" and "this."

**8.** Lines 133-134: Not clear: please consider rephrasing.

*Author proposed revision*
Lines 133–134:

We inserted ", by the use of known and/or estimable variables," between "is" and "to."

**9.** Line 143 (eq 4): Under the assumption that the errors are normally distributed? Please also specify that $\sigma_{CO_2}$ is the std. dev of...

*Author response*

$\sigma_{CO_2}$ was specified again.

*Author proposed revision*
Line 140:

The term "($\sigma_{CO_2}$)" was inserted between "$CO_2$ precision" and "is."

Line 14:

Behind "… 2008)", we inserted "The random errors generally have a normal distribution in statistics (Hoel, 1984)."

**10.** Line 166: Please consider expliciting here which parameters of eq. 5 are defined at the factory, and which ones can be corrected by field (or lab) calibration, even if reported in details later.

*Author proposed revision*
Line 160:

The first sentence was revised to be:

"The analyzer-specific working equation is deemed to be accurate immediately after calibration through the estimation of $a_{ci}$, $Z_c$, $G_c$, and $S_w$ in production, while $Z_c$ and $G_c$ can be re-estimated in the field (LI−COR Biosciences, 2021b)."

**11.** Line 168: is that part of the experiment? not very clear how it relates to the rest

*Author proposed revision*
Line 168:

"In production" was added at the beginning of this line.

**12.** Lines 170-171: bad wording

*Author response*
This was revised as Referee #1 suggested. See our response to comment #7 by Referee #1.

**14.** Line 180-182: I feel it as a "manual-like" text. I suggest avoiding expressions like "must be simple", "indeed", and be more descriptive.

*Author proposed revision*
Lines 180–182:

Two sentences were combined to be:

"In the field, the zero procedure must be feasibly operational, using one air/gas benchmark to re-estimate one parameter in the working equation."

**15.** Line 197: What do you mean? Almost?

*Author response*
"Even without zero drift" was not an appropriate phrase.

*Author proposed revision*
Line 197: "even without zero drift," was revised to be "almost without zero drift."

**16.** Line 199: Respect to...?

*Author response*
Respect to $CO_2$ zero.

*Author proposed revision*
Line 199:

After "… drifts." we inserted "the $CO_2$ zero."

**17.** Line 207: See comment at line 168. I think you are referring here to what is done at the factory during production and/or recalibration. If so, please explain better.

*Author response*
The term "calibration" in this manuscript was defined particularly as the production calibration (see lines 154–156).

**18.** Lines 214-215: This is correct, this is what is done in 80-90% of the cases. However, there exist the possibility to perform more than one span calibration, e.g., one slightly below the ambient CO2 concentration and one at a much higher value, to have a better reconstruction of the sensor behaviour: this should be mentioned in my opinion.

*Author response*
All options available should be mentioned to readers.

*Author proposed revision*
Line 214:
The following text was inserted:
"This procedure can be performed through use of either one or two span gases (LI−COR Biosciences, 2021b). If two are used, one span gas is slightly below the ambient $CO_2$ density and the other is at a much higher density to fully cover the $CO_2$ density range by the working equation. However, commonly…"

**19.** Line 216: Yes, but you can adjust it twice in the case of two span calibration (in LICOR IRGA, actually this parameter is a linear function relating absorptance to density, and what is set by the software is the offset, as the slope is fix and determined at the factory).

*Author response*
We perform the zero and span procedures iteratively two or three times.

**20.** Line 255 (eq. 12): Please recall to the reader that 44 g$H_2O$ m$^{-3}$ is a threshold for $H_2O$ concentration in air based on dew point values.

*Author proposed revision*
Line 256:

Ahead of "Accordingly …", we inserted "where 44 g$H_2O$ m$^{-3}$, as addressed in section 2, is a threshold for $H_2O$ density measurements."

**21.** Line 318 (eq. 16): Where $\sigma_{H_2O}$ is the standard deviation of the random errors...

*Author proposed revision*
Line 319:

Ahead of "The other …", we inserted "where $\sigma_{H_2O}$, as defined in Table 1, is the precision of EC150 analyzers for $H_2O$ measurements."

**22.** Line 324 (eq. 17): Even if defined earlier in the $CO_2$ section, it is probably worth it to report again what $A_c$, $A_w$ etc are. Probably a symbols list would help the reader.

*Author response*
It would be lengthy to redefine $A_w$, $A_{ws}$, $A_c$, and $A_{cs}$. Alternatively, we decided to direct readers to revisit the definitions for these parameters in Model (5).

*Author proposed revision*
Line 328:

After "… coefficient)" we inserted "; and $A_w$, $A_{ws}$, $A_c$, and $A_{cs}$ represent the same as in Model (5)."

**23.** Line 333: I would also mention the same assumptions as above (i.e., $\rho_{H_2O}$ is the closer proxy for true $\rho_{H_2O}$).

*Author proposed revision*
Line 331:

After "… $\rho_{H_2O}$", we inserted "as the closest proxy for true $\rho_{H_2O}$"

**24.** Line 348: Typo, subscript should be $H_2O$.
Line 351: Typo: $\Delta\rho_{H_2O}^s$ range.

*Author response*
Both are the same typos.

*Author proposed revision*
Lines 348 and 351:

"$\Delta\rho_{CO_2}^s$" was corrected to be "$\Delta\rho_{H_2O}^s$."

**25.** Lines 385-386: At which Ta?

*Author response*
The relative accuracy of infrared analyzers for $H_2O$ density measurements relies on the base amount of air moisture. When a temperature is high but dry (e.g., $\rho_{H_2O}$ is close to zero), the relative accuracy would be very poor.

**References**

Barrett, E. W. and Suomi, V. E.: Preliminary report on temperature measurement by sonic means, J. Atmos. Sci., 6, 273–276, 1949.

Blonquist, J. M. and Bugbee, B.: Air temperature, in: Agroclimatology: Linking Agriculture to Climate, Agronomy Monographs, edited by: Hatfield, J., Sivakumar, M., and Prueger, J., American Society of Agronomy, Crop Science Society of America, and Soil Science Society of America, Inc., Madison, WI, USA, https//doi.org/10.2134/agronmonogr60.2016.0012, 2018.

Bogoev, I., Helbig, M., and Sonnentag, O.: On the importance of high-frequency air-temperature fluctuations for spectroscopic corrections of open-path carbon dioxide flux measurements. EGU General Assembly, Apr 12-17, 2015, Vienna, Austria, ID. 2333, https://ui.adsabs.harvard.edu/abs/2015EGUGA..17.2333B/abstract, 2015.

Burba, G., Anderson, T., and Komissarov, A.: Accounting for spectroscopic effects in laser-based open-path eddy covariance flux measurements, Global Change Biology, 25, 2189-2202, DOI: 10.1111/gcb.14614, 2019.

Campbell Scientific Inc.: EC150 CO2/H2O Open-Path Gas Analyzer, Revision 09/21, Logan, UT, USA, 41 p., 2021.

Fratini, G., McDermitt, D. K., and Papale, D.: Eddy-covariance flux errors due to biases in gas concentration measurements: Origins, quantification and correction, Biogeisciences, 11: 1037-1051, 2014.

Helbig, M, Wischnewski, K., Gosselin, G.H., Biraud, S.C., Bogoev, I., Chan, W.S., Euskirchen, E.S., Glenn, A.J., Marsh, P.M., Quinton, W.L., and Sonnentag, O.:. Addressing a systematic bias in carbon dioxide flux measurements with the EC150 and the IRGASON open-path gas analyzers, Agricultural and Forest Meteorology, 228-229, 349-359, https://doi.org/10.1016/j.agrformet.2016.07.018, 2016.

Ishii, C.: Supersonic velocity in gases: especially in dry and humid air. Scientific Papers of the Institute of Physical and Chemical Research, Tokyo, 26, 201–207 pp., 1932.

Kaimal, J. C., and Gaynor, J. E.: Another look to sonic thermometry, Boundary-Layer Meteorol., 56, 401-410, https://doi.org/10.1007/BF00119215, 1991.

LI−COR Biosciences: LI-610 Portable dew point generator: Instruction manual (pp. 3-1~20), Lincoln, NE, US, 2004.

LI−COR Biosciences: LI−7500 $CO_2$/$H_2O$ Analyzer: Instruction Manual, p. 1−1 ~ D35., Lincoln, NE, USA, 2001.

LI−COR Biosciences: LI−7200RS Closed CO2/H2O Gas Analyzer: Instruction Manual, 318 p., Lincoln, NE, USA, 2021a.

LI−COR Biosciences: Using the LI−7500DS Open Path CO2/H2O Gas Analyzer and the SmartFlux 3 Systems: Instruction Manual, 224 p., Lincoln, NE, USA, 2021b.

Lin, X., Hubbard, K. G., Walter-Shea, E. A., Brandle, J. R., and Meyer, G. E.: Some perspectives on recent in situ air temperature observations: modeling the microclimate inside the radiation shields, J. Atmos. Ocean. Tech., 18, 1470–1484, https://doi.org/10.1175/1520-0426(2001)018<1470:SPORIS>2.0.CO;2, 2001.

Mauder, M., and Zeeman, M. J.: Field intercomparison of prevailing sonic anemometers, Atmos. Meas. Tech., 11, 249-263, https://doi.org/10.5194/amt-11-249-2018, 2018.

National Weather Service: Fast Facts, National Oceanic and Atmospheric Administration, Accessed October 01, 2021, https://www.weather.gov, 2021.

Richardson, A.D., Aubinet, M., Barr, A. G., Hollinger, D. Y., Ibrom, A, Lasslop, G., and Reichstein, M.: Uncetryaity quantification, In: Eddy Covariance: A Practice Guide to Measurement and Data Analysis, edited by Aubinet, M., Vesala, T., and Papale D., 85–131, Springer, New York, https://doi.org/10.1007/978-94-007-2351-1_7, 2012.

Schotanus, P., Nieuwstadt, F. T. M., and DeBruin, H. A. R.: Temperature measurement with a sonic anemometer and its application to heat and moisture fluctuations, Boundary-Layer Meteorol., 26, 81-93, 1983.

Swiatek, E.: Derivation of Temperature (Tc) from the Sonic Virtual Temperature (Ts), vapor density (ρv)/vapor pressure (e) and pressure (P). Campbell Scientific Inc. Logan, UT, 1-5 pp., 2018.

van Dijk, A.: The principles of surface flux physics. Department of Meteorology and Air Quality, Agriculture University Wageningen, 40–41 pp., 2002.

Wang, W., J.P. Xu, Y.Q. Gao, I. Bogoev, J. Cui, L.C. Deng, C. Hu, C. Liu, S.D. Liu, J. Shen, X.M. Sun, W. Xiao, G.F. Yuan, X.H. Lee. 2016. Performance evaluation of an integrated open-path eddy covariance system in a cold desert environment. J Atmos Oceanic Techn 33: 2385 2399.

Zhou, X., Gao, T., Pang, Y., Manhan, H., Li, X., Zheng, N., Suyker, A. E., Awada, T., and Zhu., J.: Based on atmospheric physics and ecological principle to assess the accuracies of field $CO_2/H_2O$ measurements from infrared gas analyzers in closed-path eddy-covariance systems, Earth and Space Science, 8, e2021EA001763, https://doi.org/10.1029/2021EA001763, 2021.

Zhou, X., Gao, T., Takle, E.S., Zhen, X., Suyker, A.E., Awada, T., Okalebo, J., Zhu, J.: Air temperature equation derived from sonic temperature and water vapor mixing ration for turbulent air flow sampled through closed-path eddy-covariance systems, Atmos. Meas. Tech., 15, 95–115, https://doi.org/10.5194/amt-15-95-2012, 2022.

Zhou, X., Yang, Q., Zhen, X., Li, Y., Hao, G., Shen, H., Gao, T., Sun, Y., and Zheng, N.: Recovery of the three-dimensional wind and sonic temperature data from a physically deformed sonic anemometer, Atmos. Meas. Tech., 11, 5981–6002, doi:10.5194/amt-11-5981-2018, 2018.

---

## Editor Decision (ED1)

- General comments

The authors carefully considered the comments provided by the reviewers, implementing the suggested modifications when deemed necessary and useful, and in my opinion significantly improving the quality of the manuscript. I thank the authors for this meticulous act of revision of their work, and for the detailed explanations and comments to my suggestions.

All in all, my final indication to the editor is to accept this manuscript for publication after a few minor and technical corrections.
For what stated above, I am not going to reply to each single Authors' response, but only focus on the very few points that still deserve attention, or some explanations, and on a few technical points to be addressed before publication. It is intended that all of the other comments fully answer my previous points. In doing so I will follow the system used by the Authors to index the sections and subsections in their "Response to Referee #2" (https://doi.org/10/5194/gi-2022-1-RC3) for what concerns the Specific comments, while for the Technical comments it is more straightforward to refer to the line numbers as in gi-2022-1-ATC1 file.

- Specific comments

1.3: I thank the Authors for such a detailed and convincing explanation. I agree with the Authors that the spectroscopic effect on the IRGA's precision is a "hot" topic, and that for that reason the temperature measurements are of great importance. What the Authors added on section 6, and on Appendix C, fully addresses my comments. I just wish to highlight that by writing "No user will buy the IRGASON to calculate Ta" I meant to say "only" for that: omitting this word may lead to misunderstanding, and I apologise with the Authors if that was the case.

2.3: this is the more tricky aspect in my opinion. I fully agree with the Authors on the reliability of the equations used: I am fully aware of the Scientific teams that work at LICOR and Campbell Scientific (even though I am not sure that 100% of EC systems in the World use sensors from only these two brands), and for sure their work is consolidated by tenth of years of expertise in the field. However, my point was not on the actual reliability of the Equations, but more on the fact that, in a scientific publication, using peer-reviewed references, when available, is the basis of the Scientific approach (I have to say that on that I tend to respectfully disagree with the Authors: "We believe the manuals from industry-trusted manufacturers have equal credibility to journal publications"). The new wording in part addresses this point: to definitely fix it, my suggestion to the Authors is to strengthen the link between using the manuals as "starting point" for the development of the method, and the fact that the approach proposed is based on the sensors' specifications: in that way the starting equations and the specs are both found in the same source.

**3: I think there were a few misunderstanding on this generic point, my apologies to the Authors for the not-clear-enough wording. First, the Authors claim their method can narrow the widest possible range of uncertainty to a significant degree, which is correct, in particular by calibrating far from the extremes of the temperature range. I just wanted to point out that the common user will rarely calibrate in extreme conditions, and so the widest range of uncertainty will rarely be the actual case ("this is what normally happens" referred to**

calibrating in mild climatic conditions, not the opposite). However, I do agree that there exist several users working in harsh conditions, for who this recommendation is precious. Secondly, "the applications proposed are not very impactful" was likely a fully-unintended but still improper selection of words. I wish to apologise again with the Authors, I should have selected a different wording. In any case I wasn't referring to the overall manuscript. My point was simply that there may have been several additional applications in the EC framework deriving from this interesting analysis. The authors addressed this point within the new sections (6.1 and the Appendix C), and also I underrated the importance of the applications proposed for the users working in very cold climates. I am certain the manuscript is impactful - would have I thought the opposite, I wouldn't have accepted to review it.

- Technical comments:

line 81: overall (typo)
line 83: available (typo)
line 86: Lee et al. (1999) (typo, a 9 missing)
line 480 (and elsewhere): "hourly" fluxes may also be half-hourly, or calculated over other time scales. Probably better to use a different term, like "calculated and temporally aggregated fluxes" or similar; or to report earlier that you use "hourly" to refer to these fluxes as it is a common time scale, but all would remain valid for half-hourly fluxes or fluxes calculated over different time scales
line 491: "with an error as ranged by its accuracy and Ta with an error": please consider rephrasing for more clarity
line 513: is added only by (typo, "by" is missing)
line 544: this is the first time you mention EddyPro, probably you wish to consider explaining what it is. Or maybe reconsider including it at all
line 645: "measurement uncertainties" may be misleading: please consider selecting a different wording
line 664: please consider removing the word "more"
line 805: I think there is a typo in the title of Appendix C: "The relationship of measured to true covariance to of vertical wind speed with CO2, H2O, and air temperature" should be instead "The relationship of measured to true covariance of vertical wind speed with CO2, H2O, and air temperature"
line 808: please consider rephrasing to something like "from the covariance between each of the three components of the 3-D wind field and the density of CO2/H2O"
line 814 (Eq. C2): subscript "i" missing for rho-alpha
line 818 (Eq. C3): I think you are implicitly using Reynolds rule to derive the final equation (that is the average of the sum of two terms is the sum of their averages): please consider making it explicit.
line 838: an equation is missing after "and" (I think covariance between v and rho measured = covariance between v and rho true), while "are also" should be deleted in my opinion
line 839: $v^2$ mean repeated (second one should be $w^2$ mean)
line 841: scaler should be scalar instead (typo)
line 842: this means that this would not be valid for momentum flux (covariance between vectors)? (out of the scope of the manuscript, just came to my mind as a matter of curiosity)
line 845-846: please state that what is between square brackets is the notation for the maximisation of covariance (otherwise the reader may think you are multiplying things)

line 851-854: is this also valid for other spectral correction methods, like Ibrom et al. 2007 and Fratini et al. 2012 methods?

line 858: please consider rephrasing

line 862: please consider rephrasing

line 918: Biogeosciences (typo)

---

## Author Response (AR2)

**INSTITUTE OF APPLIED ECOLOGY, CHINESE ACADEMY OF SCIENCES**

**72 Wenhua Road, Shenyang, Liaoning, 110016, China**

September 06, 2022

RE: Response to referee's comments on GI-2022-1R

Dr. Salvatore Grimaldi
Dept. for Innovation in Bio., Agro-food, & For. Systems
University of Tuscia, Viterbo, Italy

Dear Dr. Grimaldi,

Thank you for inviting us to revise our manuscript, "Accuracies of field $CO_2-H_2O$ measurements from open-path eddy-covariance systems: Assessment based on atmospheric physics and biological environment," to address the comments included in the referee's note for publication in *Geoscientific Instrumentation, Methods and Data Systems (GI)*. Through the review process, we have been enjoyed our professional discussions with the referee about his/her professional and constructive comments, which significantly improves the quality of our manuscript. We have felt lucky to meet this referee over the review process. We appreciate referee's knowledge and dedication. Many thanks to the referee for his/her two rounds of reviews.

The authors carefully checked every comment from the referee for this revision. Our discussions and proposed revisions in response to the corresponding comments are given below. While responding the comments, the atmosphere $CO_2$ background value of 415 ppm (760 $mgCO_2 \ m^{-2}$) in 2021 was brought into current as 419 ppm (767 $mgCO_2 \ m^{-2}$). Accordingly, Table 2 and Figure 2 were updated. After our revision, proofreading was requested from the communication team of Campbell Scientific. Ms. Kati Kovacs proofread the manuscript. The corrections from her proofreading are indicated in this version (GI-2022-1RR) with trackers.

In our response to the comments, we directly follow the indexes of referee's comments for the sections and subsections as well as the line numbers. Because the line numbers were used by the Referee while reviewed GI-2022-1R, these line numbers are more correlated with manuscript GI-2022-1R than GI-2022-1RR.

We appreciate your favorable consideration for publication of this manuscript in *GI*.

Sincerely,

Ning Zheng, Ph.D., Application Scientist
Eddy-Covariance Flux Instrumentation

**Response to Referee's comments on "Accuracies of field $CO_2-H_2O$ measurements from open-path eddy-covariance systems: Assessment based on atmospheric physics and biological environment"**

X.H. Zhou, T. Gao, N. Zheng, B. Yang, Y. Li, F.Y. Yu, T. Awada, J.J. Zhu
https://doi.org/10.5194/gi-2022-1R

**Response to Referee's Comments**
(*gi-2022-1-referee-report.pdf*)

**General comments**

The authors carefully considered the comments provided by the reviewers, implementing the suggested modifications when deemed necessary and useful, and in my opinion significantly improving the quality of the manuscript. I thank the authors for this meticulous act of revision of their work, and for the detailed explanations and comments to my suggestions.

All in all, my final indication to the editor is to accept this manuscript for publication after a few minor and technical corrections.

For what stated above, I am not going to reply to each single Authors' response, but only focus on the very few points that still deserve attention, or some explanations, and on a few technical points to be addressed before publication. It is intended that all of the other comments fully answer my previous points. In doing so I will follow the system used by the Authors to index the sections and subsections in their "Response to Referee #2" (https://doi.org/10/5194/gi-2022-1-RC3) for what concerns the Specific comments, while for the Technical comments it is more straightforward to refer to the line numbers as in gi-2022-1-ATC1 file.

*Author response*
Again, we have been so enjoyed our professional discussions with you about your professional and constructive comments. We have felt lucky to meet you over this review process. We appreciate your knowledge and dedication. Many thanks.

**Specific comments**

1.3: I thank the Authors for such a detailed and convincing explanation. I agree with the Authors that the spectroscopic effect on the IRGA's precision is a "hot" topic, and that for that reason the temperature measurements are of great importance. What the Authors added on section 6, and on Appendix C, fully addresses my comments. I just wish to highlight that by writing "No user will buy the IRGASON to calculate Ta" I meant to say "only" for that: omitting this word may lead to misunderstanding, and I apologise with the Authors if that was the case.

*Author response*
Our review and response processes are truly academic and professional discussions. We truly enjoyed your comments. No apology necessary. I believe we are sharing the same goal to gradually improve the flux measurements in our community.

*Author proposed revision*
A revision is not suggested by the Referee.

2.3: this is the more tricky aspect in my opinion. I fully agree with the Authors on the reliability of the equations used: I am fully aware of the Scientific teams that work at LICOR and Campbell Scientific (even though I am not sure that 100% of EC systems in the World use sensors from only these two brands), and for sure their work is consolidated by tenth of years of expertise in the field. However, my point was not on the actual reliability of the Equations, but more on the fact that, in a scientific publication, using peer-reviewed references, when available, is the basis of the Scientific approach (I have to say that on that I tend to respectfully disagree with the Authors: "We believe the manuals from industry-trusted manufacturers have equal credibility to journal publications"). The new wording in part addresses this point: to definitely fix it, my suggestion to the Authors is to strengthen the link between using the manuals as "starting point" for the development of the method, and the fact that the approach proposed is based on the sensors' specifications: in that way the starting equations and the specs are both found in the same source.

*Author response*

I agree with your "disagree". Our wording "equal credibility" might not be fully fair although we did emphasize on "industry-trusted manufacturers". Overall, for sure, journal publications have better credibility than manufacturer manuals. In our community, it is well-known that LI-COR Biosciences did publish credible manuals for almost 30 years, particularly related to gas analyzers. In this manuscript, the models in a gas analyzer manual from LI-COR Biosciences are used to explain the specifications from a gas analyzer manual by Campbell Scientific. Both manufacturers are competitors in $CO_2$ and $H_2O$ analyzers, which is the reason we try to balance the use of literature from both. To the best of our knowledge, we are not aware of any publication to explain the models of $CO_2$ and $H_2O$ measurements as clearly as LI−COR Biosciences (2001 ~2021) does although Fratini et. al (2014) principally used the models. Therefore, we feel that the manuals of LI-COR Biosciences are the best sources for our citations.

*Author proposed revision*
A revision is not suggested by the Referee.

**3: I think there were a few misunderstanding on this generic point, my apologies to the Authors for the not-clear-enough wording. First, the Authors claim their method can narrow the widest possible range of uncertainty to a significant degree, which is correct, in particular by calibrating far from the extremes of the temperature range. I just wanted to point out that the common user will rarely calibrate in extreme conditions, and so the widest range of uncertainty will rarely be the actual case ("this is what normally happens" referred to calibrating in mild climatic conditions, not the opposite). However, I do agree that there exist several users working in harsh conditions, for who this recommendation is precious. Secondly, "the applications proposed are not very impactful" was likely a fully-unintended but still improper selection of words. I wish to apologise again with the Authors, I should have elected a different wording. In any case I wasn't referring to the overall manuscript. My point was simply that there may have been several additional applications in the EC framework deriving from this interesting analysis. The authors addressed this point within the new sections (6.1 and the Appendix C), and also I underrated the importance of the applications proposed for the users working in very cold climates. I am certain the manuscript is impactful - would have I thought the opposite, I wouldn't have accepted to review it.**

*Author response*
Thank you so much for your positive comments. Again, our review and response processes are truly academic and professional discussions. We truly enjoyed your comments. No apology necessary.

Yes, common users rarely calibrate $CO_2$ and $H_2O$ infrared gas analyzer in extreme conditions. Similar to Fratini et al. (2014) did, more work is needed to ensure the quality of data from extreme conditions. Other approaches are needed to stabilize the performance of $CO_2$ and $H_2O$ infrared gas analyzers in extreme conditions. This study provides a pilot analysis.

*Author proposed revision*
We revised some wording while thoroughly reading and checking the manuscript. See the version of GI-2022-1RR with tracks

**Technical comments**

line 81: overall (typo).

*Author response*
Corrected

line 83: available (typo).

*Author response*
Corrected

line 86: Lee et al. (1999) (typo, a 9 missing).

*Author response*
Corrected

line 480 (and elsewhere): "hourly" fluxes may also be half-hourly, or calculated over other time scales. Probably better to use a different term, like "calculated and temporally aggregated fluxes" or similar; or to report earlier that you use "hourly" to refer to these fluxes as it is a common time scale, but all would remain valid for half-hourly fluxes or fluxes calculated over different time scales.

*Author response*
We understood your concern. We are always struggling with the use of "hourly" for this context. We are trying a new approach to this expression.

*Author proposed revision*
Throughout the manuscript
"Hourly $CO_2$/$H_2O$ flux" was revised to "$CO_2$/$H_2O$ flux data"

line 491: "with an error as ranged by its accuracy and Ta with an error": please consider rephrasing for more clarity.

*Author response*
Rephrased.

line 513: is added only by (typo, "by" is missing)

> *Author response*
> Corrected.

line 544: this is the first time you mention EddyPro, probably you wish to consider explaining what it is. Or maybe reconsider including it at all.

> *Author response*
> The full official name plus the EddyPro reference would be helpful to readers who are not familiar with EddyPro.

> *Author proposed revision*
> Line 544: Revise the EddyPro program to "EddyPro® Eddy Covariance Software (LI−COR Biosciences, 2021a).
>
> LI−COR Biosciences (2021a) is added to References and the references are reordered.

line 645: "measurement uncertainties" may be misleading: please consider selecting a different wording.

> *Author response*
> The sentence was rephrased.

> *Author proposed revision*
> Original
> "…, the measurement uncertainties for analyzer specifications are not expected to increase rather some current terms could be removed from the current specification list, …"
> Revised
> "…, the number of these uncertainty sources for analyzer specifications is not expected to increase, rather some current uncertainty sources could be eliminated from the current specification list, …".
>
> Additionally, the wording related to this revision in this paragraph also is accordingly revised.

line 664: please consider removing the word "more".

> *Author response*
> The word of "more" was removed.

line 805: I think there is a typo in the title of Appendix C: "The relationship of measured to true covariance to of vertical wind speed with $CO_2$, $H_2O$, and air temperature" should be instead "The relationship of measured to true covariance of vertical wind speed with $CO_2$, $H_2O$, and air temperature"

> *Author response*
> The extra word of "to" was removed from "to of".

line 808: please consider rephrasing to something like "from the covariance between each of the three components of the 3-D wind field and the density of $CO_2$/$H_2O$"

*Author response*
Our expression is not precise.

*Author proposed revision*
Before revision
"from covariance of 3-D wind with a $CO_2$/$H_2O$ density."
After revision
"from covariance of an 3-D wind component with a $CO_2$/$H_2O$ density."

line 814 (Eq. C2): subscript "i" missing for rho-alpha

*Author response*
Subscript "$i$" is added to $\Delta\rho_\alpha$.

line 818 (Eq. C3): I think you are implicitly using Reynolds rule to derive the final equation (that is the average of the sum of two terms is the sum of their averages): please consider making it explicit.

*Author response*
Yes, we used the Reynolds rule.

*Author proposed revision*
Original
"the over bar is an averaging operator,"
Revised
"the over bar is the Reynolds' averaging operator,"

line 838: an equation is missing after "and" (I think covariance between v and rho measured = covariance between v and rho true), while "are also" should be deleted in my opinion line 839: v^2 mean repeated (second one should be w^2 mean).

*Author response*
Yes, right. Apparently, our newly added appendix in last revision needs more attentions. Thank you so much for your such dedication. We will read through again and ask a professional to proofread this version although revisions are minor from last version.

*Author proposed revision*
a. "Both $\overline{u'\rho_{\alpha l}'} = \overline{u'\rho_{\alpha Tl}'}$ and are" is corrected to be "Both $\overline{u'\rho_{\alpha l}'} = \overline{u'\rho_{\alpha Tl}'}$ and $\overline{v'\rho_{\alpha l}'} = \overline{v'\rho_{\alpha Tl}'}$ are".
b. "$...,\overline{u^2},\overline{v^2},\overline{v^2},...$" is corrected to be "$...,\overline{u^2},\overline{v^2},\overline{w^2},...$".

line 841: scaler should be scalar instead (typo).

*Author response*
Corrected.

line 842: this means that this would not be valid for momentum flux (covariance between vectors)? (out of the scope of the manuscript, just came to my mind as a matter of curiosity).

*Author response*
In coordination rotation process, unlike a mean of scalar unchangeable, the mean of vectors related to momentum flux is changed. Carefully examine the rotation matrices, the mean of scalar is not involved. Instead, all means of 3-D vectors are included in the matrices. We did not numerically analyze deeper.
*Author proposed revision*
No revision is suggested by the Referee.

line 845-846: please state that what is between square brackets is the notation for the maximisation of covariance (otherwise the reader may think you are multiplying things).

*Author response*
The expression is hard for readers although the expression is correct.

*Author proposed revision*
Before revision
"Therefore, the maximum covariance in magnitude among $\overline{(w'\rho'_{\alpha l})}_r$ ($l$ from $-k$ to $k$) $[\overline{(w'\rho'_{\alpha})}_{rm}]$ is equal to the maximum in magnitude among $\overline{(w'\rho'_{\alpha Tl})}_r$ $[\overline{(w'\rho'_{\alpha T})}_{rm}]$ (Moncrieff et al., 1997; Ibrom et al., 2007), given by"

After revision
"Therefore, the maximum covariance in magnitude among $\overline{(w'\rho'_{\alpha l})}_r$ ($l$ from $-k$ to $k$) is equal to the maximum in magnitude among $\overline{(w'\rho'_{\alpha Tl})}_r$ (Moncrieff et al., 1997; Ibrom et al., 2007). Denoting the former maximum covariance by $\overline{(w'\rho'_{\alpha})}_{rm}$, where subscript $m$ indicates the maximum, and the latter one by $\overline{(w'\rho'_{\alpha T})}_{rm}$, this equality leads to".

line 851-854: is this also valid for other spectral correction methods, like Ibrom et al. 2007 and Fratini et al. 2012 methods?

*Author response*
Yes, in terms of a correction factor in the context, both methods are valid, but Ibrom et al. (2007) and Fratini et al. (2012) methods are particularly for close-path eddy-covariance systems. This study is for open-path eddy-covariance systems. Therefore, both are not included in discussion for this context.

line 858: please consider rephrasing.

*Author response*
Rephrased.

line 862: please consider rephrasing.

*Author response*
Rephrased.

line 918: Biogeosciences (typo).

*Author response*
Corrected

**References**

Fratini, G., McDermitt, D. K., and Papale, D.: Eddy-covariance flux errors due to biases in gas concentration measurements: Origins, quantification and correction, Biogeisciences, 11: 1037-1051, 2014.

Fratini, G., Ibrome, A. Burba, G.G., Arriga, N. and Papale, D.: Relative humidity effects on water vapour fluxes measured with closed-path eddy-covariance systems with short sampling lines, Agricultural and Forest Meteorology, 165: 53-63, 2012. https://doi.org/10.1016/j.agrformet.2012.05.018

Ibrom, A., Dellwik, E., Flyvbjerg, H., Jensen, N. O., and Pilegaard, K.: Strong low-pass filtering effects on water vapour flux measurements with closed-path eddy correlation systems, Agr. Forest Meteorol., 147: 140–156, https://doi.org/10.1016/j.agrformet.2007.07.007, 2007.

LI−COR Biosciences: EddyPro® Software Instruction, Version 7, p. 1−1~6−74, Lincoln, NE, USA, 2021a.

LI−COR Biosciences: LI−7500 $CO_2$/$H_2O$ Analyzer: Instruction Manual, p. 1−1 ~ D35., Lincoln, NE, USA, 2001.

LI−COR Biosciences: LI−7200RS Closed CO2/H2O Gas Analyzer: Instruction Manual, 318 p., Lincoln, NE, USA, 2021b.

LI−COR Biosciences: Using the LI−7500DS Open Path CO2/H2O Gas Analyzer and the SmartFlux 3 Systems: Instruction Manual, 224 p., Lincoln, NE, USA, 2021c.

Moncrieff, J. B., Massheder, J. M., de Bruin, H., Elbers, J., Friborg, T., Heusinkveld, B., Kabat, P., Scott, S., Soegaard, H., and Verhoef, A.: A system to measure surface fluxes of momentum, sensible heat, water vapour and carbon dioxide, J. Hydrol., 188-189, 589–611, https://doi.org/doi:10.1016/S0022-1694(96)03194-0, 1997.